

# Diffusion in generalized hydrodynamics and quasiparticle scattering

**Jacopo De Nardis[1], Denis Bernard[2] and Benjamin Doyon[3]**

**1** Department of Physics and Astronomy, University of Ghent,
Krijgslaan 281, 9000 Gent, Belgium
**2** Laboratoire de Physique Théorique de l'Ecole Normale Supérieure, CNRS, ENS,
PSL University & Sorbonne Université, 75005 Paris, France
**3** Department of Mathematics, King's College London, Strand WC2R 2LS, London, U.K.

## Abstract

We extend beyond the Euler scales the hydrodynamic theory for quantum and classical integrable models developed in recent years, accounting for diffusive dynamics and local entropy production. We review how the diffusive scale can be reached via a gradient expansion of the expectation values of the conserved fields and how the coefficients of the expansion can be computed via integrated steady-state two-point correlation functions, emphasising that $\mathcal{PT}$-symmetry can fully fix the inherent ambiguity in the definition of conserved fields at the diffusive scale. We develop a form factor expansion to compute such correlation functions and we show that, while the dynamics at the Euler scale is completely determined by the density of single quasiparticle excitations on top of the local steady state, diffusion is due to scattering processes among quasiparticles, which are only present in truly interacting systems. We then show that only two-quasiparticle scattering processes contribute to the diffusive dynamics. Finally we employ the theory to compute the exact spin diffusion constant of a gapped XXZ spin$-1/2$ chain at finite temperature and half-filling, where we show that spin transport is purely diffusive.



# 1 Introduction

Exactly solvable models in statistical physics constitute an ideal outpost to study a whole range of emerging physical phenomena [1]. The Ising model in two dimensions [2] or the Heisenberg spin chain [3] played a fundamental role in the understanding of classical and quantum phase transitions, in the theory of magnetism, and in general in deciphering the ground state properties of many-body quantum systems. While these quotations are mostly associated to the physics observed in systems at equilibrium, during the past years the study of non-equilibrium quantum and classical physics has become gradually one of the main research topics in high-energy and condensed matter physics. In particular a large research community has devoted itself to the study of the non-equilibrium dynamics of isolated quantum systems, with the exactly solvable models playing a pivotal role. One of the most studied protocol to put an isolated system out of equilibrium is the so-called quantum quench [4–9], where an initial many-body state is let to unitarly evolve under a many-body Hamiltonian. The initial state can be chosen to be homogeneous (invariant under space translations) [4, 5], or inhomogeneous [10–15], as it is the case in many experimental settings [16–21]. The theoretical study of the latter has started in earnest only very recently, in particular for the case of the so-called bi-partite quench, where two macroscopic many-body systems with different thermodynamic quantities (temperature or chemical potential for example), are joined together [10–12, 12–15, 22–43] or with an initial non-homogeneous profile of densities [44–48]. The intrinsic difficulty of such non-equilibrium dynamics has motivated the formulation of a hydrodynamic description for the dynamics at large space and time scales, based on the local conserved quantities [49, 50]. In cases where the only local conserved operators are the Hamiltonian of the system, the momentum and, for example, the total particle number, the system is indeed expected to be described at large scales by an Euler hydrodynamics for the energy, momentum and mass densities. Such hydrodynamic theories were extensively used also for cold atomic gases, see for example [51–55]. However for systems with a large number of conserved quantities, such as the Lieb-Liniger model describing quasi-one-dimensional cold atomic gases, there are many more emerging hydrodynamic degrees of freedom, one for each conserved quantity. The hydrodynamics, reproducing the large time and scale dynamics, must therefore be enlarged in order to account for this. In a quasiparticle picture that is natural in Bethe ansatz and other integrable models, the Euler-scale hydrodynamic theory is formulated for the density of stable quasiparticle excitations. Nowadays such a theory is referred to as Generalized Hydrodynamics (GHD). It consists of a non-linear differential equation describing quasiparticles propagating with effective velocities which are functional of the local density, due to the microscopic interactions among the elementary constituents. The net effect of such interactions is the so-called dressing of thermodynamic functions, which can be exactly expressed as functionals of the local density using the underlying integrable structure. The GHD equations were successfully applied to cold atomic systems [56,57] and verified experimentally, [58], to spin chains [50,59–62], classical gases and fields [63–65] and Floquet dynamics [66].

In the Euler hydrodynamic theory, the length scales considered are of the same order as that of the time scales, $x \sim t$ with $t$ much greater than any other available scales, and the scattering processes among quasiparticles are neglected. However they become important at smaller length scales, giving rise to different physical phenomena. One of them

is dissipation and diffusive spreading [67–71], which are relevant phenomena at diffusive space-time scales, $x \sim t^{1/2}$. This follows from the expectation [72] that generic many-body interacting systems, like normal fluids, display a coexistence of ballistic (convective) and diffusive dynamics [69,73–77], as classical systems of particles indeed show [78,79]. Given that the hydrodynamics of the classical gas of hard rods, which has an extensive amount of conserved quantities, contains diffusion [80], it is a relevant question to understand if the simple interactions that characterize quantum integrable models can lead to similar generic properties at large scales. In [81] this was indeed shown to be the case: there is diffusion dynamics in isolated integrable quantum (and classical) systems and the diffusion matrix can be exactly computed as a functional of the thermodynamic quantities of the local stationary state. Its form is similar to that found for the hard rods, but with important differences encoding the interaction and statistics of the quasiparticle excitations. In a following publication [66], the diagonal part of the diffusion matrix was shown to be obtainable from quasiparticle spreading, and it was shown that diffusion has consequences for the dynamics of the out-of-time correlators (OTOC) [82] in interacting integrable systems, which displays the generic behaviour observed in random circuit models [83–85].

In this paper we fully develop the results of [81]. In particular we clearly present the connection between the hydrodynamic theory and local correlation functions of the densities of conserved quantities. Using a form factor expansion, we put in evidence the presence of a hierarchy of quasiparticle excitations, to be considered in order to reconstruct the large space and time expansion of the non-equilibrium dynamics. The so-called one-particle-hole excitations are the simplest excitations on top of the local equilibrium states. They correspond to single quasiparticles moving with an effective velocity, and they are present in interacting and non-interacting systems. They are responsible for the Euler-scale GHD, the equations of motion for the densities of conserved charges at the largest scales. In order to go beyond the Euler scale dynamics, one needs to include scattering events among quasiparticles, which amount to considering the two- or higher-particle-hole excitations. Quite remarkably, as we show in this paper, the entire physics of diffusion is fixed by the two-particle-hole excitations, which can be interpreted as accounting for two-body scattering events among quasiparticles: the infinite tower of particle-hole excitations truncates. Two-particle-hole excitations are only present in truly interacting systems, therefore confirming the general intuition that there is no diffusive dynamics in non-interacting systems [69, 86] (except, potentially, with external disorder [87–90]).

Our results are based on two main assumptions:

(a) First, we make the standard *hydrodynamic assumptions*. That is, on one hand, the assumption that at large times, the relevant degrees of freedom for describing all averages on a time slice are reduced to the local mean charge densities on that time slice, which are then identified as the hydrodynamic variables; and on the other hand, the assumption that the derivative expansions, up to second order, of local averages in terms of local mean charge densities captures in a consistent way the effective dynamics. Proving these assumptions is of course one of the most challenging problems of mathematical physics, beyond the scope of this paper

(b) Second, we assume some analytical properties of *finite-density (thermodynamic) form factors* (matrix elements) of conserved densities and currents. These are

generalisations of analytical properties calculated for certain conserved densities in some integrable models, are in analogy with well established bootstrap framework of relativistic quantum field theory, and are in agreement with recent proposals [137].

The paper is organized as follows:

- In section 2 we review the general, standard theory of hydrodynamics, based on the *gradient expansion* for the expectation values of the currents of conserved densities, which allows one to go beyond the Euler scale. We connect the coefficients of such expansions to the steady-state correlation functions of conserved densities and their currents. Conserved densities have an inherent ambiguity at the Euler scale: they are not fully fixed by the conserved quantities themselves. We emphasize that the presence of $\mathcal{PT}$-symmetry – found in a wide family of integrable models including those most studied in the literature – allows one to lift this ambiguity.

- In section 3 we show how to properly define the local stationary states as macrostates of quasiparticles, and how to evaluate the necessary correlation functions using the concept of excitations above a given macrostate: the so called *particle-hole excitations*.

- In section 4 we show how to compute the *diffusion matrix* in generic quantum integrable models via the calculation of the integrated current-current correlation functions and the sum over of two-particle-hole excitations as the only contributing intermediate states. In order to obtain this correlation function, we *conjecture* a generic form of the poles of the form factor of the current operators, by generalising the so-called form factor axioms. We then extend the formula we find for the diffusion matrix, following the principles already developed at the Euler scale, to arbitrary quantum and classical integrable models that have an Euler-scale GHD description, thus obtaining a completely general expression for the diffusion matrix in a wide family of integrable models. We also derive some of the properties of the diffusion matrix and diffusion kernel, and we present a number of important applications as for example to the solution of the bi-partite quench and the increase of entropy.

- Finally in section 6 we use the previus result to exactly compute the *spin diffusion constant* of an XXZ spin$-1/2$ chain at finite temperature and half-filling, a regime where there is no ballistic spin transport and, quite remarkably for an integrable models, spin transport is purely diffusive. We provide *numerical predictions* for the diffusion constant at infinite temperature and we discuss its comparison with the numerical predictions from tDMRG obtained in [35, 91, 92].

## 2 Hydrodynamic theory and Navier-Stokes equation

Hydrodynamics is a very general theory for emerging degrees of freedom at long wavelengths and low frequencies. We here review a few basic principles underlying the hydrodynamic approximation. These principles are well known, but it is convenient to review them, and express them in the context of an arbitrary number of conservation laws.

We note that the basic principles of hydrodynamics, the construction of the hydrodynamic equations, and their relations to correlation and response functions, are largely independent of the microscopic nature of the system, which may be quantum or classical, a lattice of spins, a field theory, or a gas of interacting particles, etc. It is also important to realise that, although many works on the hydrodynamic theory of quantum systems is based on studying the analytic structure of Green's functions, this is in fact not necessary; in models with a large amount of conserved densities, the approach we review here, in particular for the diffusion matrix, appears to be more powerful.

## 2.1 Hydrodynamic expansion

The physical idea at the basis of hydrodynamics is that, after an appropriate relaxation time, an inhomogeneous, non-stationary system approaches, locally, states which have maximised entropy with respect to the conservation laws afforded by the dynamics. Let $\mathfrak{q}_i(x, t)$ and $\mathfrak{j}_i(x, t)$ be conserved densities and currents. Then the conservation laws are

$$\partial_t \mathfrak{q}_i(x, t) + \partial_x \mathfrak{j}_i(x, t) = 0, \qquad i \in I \tag{2.1}$$

with associated conserved quantities

$$Q_i = \int \mathrm{d}x \, \mathfrak{q}_i(x, t) \tag{2.2}$$

($I$ is the index set indexing the conserved quantities). A homogeneous, stationary, maximal entropy state has density matrix formally written as

$$Z^{-1} e^{-\sum_i \beta_i Q_i}. \tag{2.3}$$

In inhomogeneous, non-stationary situations, relaxation occurs within fluid cells which are large enough with respect to microscopic scales, and small enough with respect to the variation lengths and times, the latter therefore assumed to be large. Since a maximal entropy state is completely characterised by the averages of the local (or quasi-local) conserved densities within it, according to this idea, a state at a time slice $t$ is completely determined by the profiles of conserved densities $\{\bar{\mathfrak{q}}_i(x, t) := \langle \mathfrak{q}_i(x, t) \rangle : x \in \mathbb{R}, i \in I\}$. This means that the state of the system on the time slice $t$ – that is, the set of all averages of all local observables at $t$ – can be described in terms of a reduced number of degrees of freedom, the set $\{\bar{\mathfrak{q}}_i(x, t) : x \in \mathbb{R}, i \in I\}$, instead of the exact density matrix, or many-body distribution, at $t$. That is, for every observable $\mathfrak{o}(x, t)$, there exists a functional $\mathfrak{O}[\bar{\mathfrak{q}}.(\cdot, t)](x, t)$ such that

$$\langle \mathfrak{o}(x, t) \rangle = \mathfrak{O}[\bar{\mathfrak{q}}.(\cdot, t)](x, t). \tag{2.4}$$

The main point is that the dynamical variables of hydro are the conserved densities evaluated on a given time slice. The choice of reference time slice is arbitrary. These dynamical variables evolve in time according to the hydrodynamic equations.

This reduction of the number of degrees of freedom is the main postulate of hydrodynamics. It is expected to provide a good approximation to the evolution (in an asymptotic sense) when variations in space and time occur on lengths which are large enough.

Consider the continuity equation for the conserved densities and currents (an *operatorial equation*, direct consequence of the dynamics of the system),

$$\partial_t \mathfrak{q}_i(x, t) + \partial_x \mathfrak{j}_i(x, t) = 0. \tag{2.5}$$

Hydrodynamics is a theory for the evolution of the *mean values* of these operators

$$\bar{\mathsf{q}}_i(x,t) = \langle \mathsf{q}_i(x,t) \rangle, \qquad \bar{\mathsf{j}}_i(x,t) = \langle \mathsf{j}_i(x,t) \rangle, \tag{2.6}$$

provided by the continuity equation

$$\partial_t \bar{\mathsf{q}}_i(x,t) + \partial_x \bar{\mathsf{j}}_i(x,t) = 0. \tag{2.7}$$

By the main postulate of hydrodynamics described above, the average currents $\bar{\mathsf{j}}_i(x,t)$ may depend on the densities $\bar{\mathsf{q}}_j(y,t)$ at all points $y$ and index $j$, but at identical time,

$$\bar{\mathsf{j}}_i(x,t) =: \mathfrak{J}_i[\bar{\mathsf{q}}.(\cdot,t)](x,t). \tag{2.8}$$

Since entropy maximisation is supposed to occur within local fluid cells when variation lengths are large, it is natural to assume that the functional $\mathfrak{J}_i[\bar{\mathsf{q}}.(\cdot,t)]$ depends on the values $\bar{\mathsf{q}}_j(y,t)$ for all $j$ but only for $y$ near to $x$. We thus express it in a *derivative expansion*,

$$\mathfrak{J}_i[\bar{\mathsf{q}}.(\cdot,t)](x,t) = \mathcal{F}_i(\bar{\mathsf{q}}.(x,t)) - \frac{1}{2}\sum_{j\in I}\mathfrak{D}_i^{\ j}(\bar{\mathsf{q}}.(x,t))\partial_x\bar{\mathsf{q}}_j(x,t) + O\big(\partial_x^2\bar{\mathsf{q}}.(x,t)\big), \tag{2.9}$$

where both $\mathcal{F}_i(\bar{\mathsf{q}}.(x,t))$ and $\mathfrak{D}_i^{\ j}(\bar{\mathsf{q}}.(x,t))$ are *functions* of the charge densities at position $x,t$ only. As consequence from eqs.(2.7,2.9), by neglecting higher order in derivatives we have (with implicit summation over repeated indices)

$$\partial_t\bar{\mathsf{q}}_i(x,t) + \partial_x\mathcal{F}_i(\bar{\mathsf{q}}.(x,t)) - \frac{1}{2}\partial_x\big(\mathfrak{D}_i^{\ j}(\bar{\mathsf{q}}.(x,t))\,\partial_x\bar{\mathsf{q}}_j(x,t)\big) = 0. \tag{2.10}$$

The first two terms correspond to the Euler equation and the last one to the Navier-Stokes correction.

In ordinary hydrodynamics, the derivative expansion is usually expected to be meaningful (at least if there is no sub/super-diffusion, namely when the matrix $\mathfrak{D}_i^{\ j} = 0$ or $\mathfrak{D}_i^{\ j} = \infty$) only up to the order written. Higher order terms in the derivative expansion are usually not predictive, because at that order, the assumption of the reduction of the number of degrees of freedom is incorrect.

The form of the first term in (2.9), $\mathcal{F}_i(\bar{\mathsf{q}}.(x,t))$, is a direct consequence of the thermodynamics of the model: indeed, it can be obtained by assuming the conserved densities (hence the state) to be homogeneous. The function $\mathcal{F}_i(\bar{\mathsf{q}}.)$ expresses the conserved currents as functions of conserved densities in homogeneous, stationary, maximal entropy state: these are the *equations of state*. The second term, $\mathfrak{D}_i^{\ j}(\bar{\mathsf{q}}.(x,t))$, encodes what is referred to as the *constitutive relations*, and its form is not a property of the homogeneous, stationary, maximal entropy states; it must be calculated in a different way.

## 2.2 Two-point function sum rules

A convenient way to code for hydrodynamic diffusion is via the connected two-point functions for all the local conserved densities[1]

$$S_{ij}(x,t) := \langle \mathsf{q}_i(x,t)\mathsf{q}_j(0,0) \rangle^c \tag{2.11}$$

---

[1]The upper index $\langle \cdots \rangle^c$ indicates that this is the *connected* correlation function: $\langle AB \rangle^c = \langle AB \rangle - \langle A \rangle \langle B \rangle$.

in a generic homogeneous stationary state. By the conservation law, and assuming clustering property of correlation functions of local densities, the space integral of $S_{ij}(x,t)$ is constant in time. It defines the matrix of static susceptibilities

$$C_{ij} := \int dx\, S_{ij}(x,t) = \int dx\, S_{ij}(x,0). \tag{2.12}$$

The tensor $C_{ij}$ is symmetric by translation invariance, and hence it defines a *metric on the space of conserved densities*.

We introduce the variance $\frac{1}{2}\int dx\, x^2 \left(S_{ij}(x,t) + S_{ij}(x,-t)\right)$ to code for the spreading of the correlations between the local densities. As a consequence of the conservation laws and of space and time translation invariance, we have the following sum rule [69] (see appendix A):

$$\frac{1}{2}\int dx\, x^2 \left(S_{ij}(x,t) + S_{ij}(x,-t) - 2S_{ij}(x,0)\right) = \int_0^t ds \int_0^t ds' \int dx\, \langle j_i(x,s) j_j(0,s')\rangle^c. \tag{2.13}$$

Note that by stationarity of the state, the current-current correlation function on the right-hand side only depends on $s-s'$.

Under appropriate simple conditions that we are going to spell out below, the spreading of these correlation functions is governed by separate ballistic and diffusive contributions:

$$\frac{1}{2}\int dx\, x^2 \left(S_{ij}(x,t) + S_{ij}(x,-t)\right) = D_{ij} t^2 + \mathfrak{L}_{ij} t + o(t) \tag{2.14}$$

as $t \to +\infty$, for some finite coefficients $D_{ij}$ and $\mathfrak{L}_{ij}$, which are respectively related to the ballistic and diffusive expansions of the correlation functions. The coefficients $D_{ij}$ are the Drude weights, and the coefficients $\mathfrak{L}_{ij}$ form what is called the Onsager matrix.

Let us now explain (2.14). As it is clear from the sum rule (2.13), the large time behaviour of the variance $\frac{1}{2}\int dx\, x^2 \left(S_{ij}(x,t) + S_{ij}(x,-t)\right)$ is encoded in the large time behaviour of the space-integrated current-current connected correlation functions. If the latter is finite at large time, the former is going to grow quadratically in time. More precisely, assume that the coefficients $D_{ij}$, defined as

$$D_{ij} := \lim_{t\to\infty} \frac{1}{2t} \int_{-t}^t ds \int dx\, \langle j_i(x,s) j_j(0,0)\rangle^c, \tag{2.15}$$

are finite. Then $\frac{1}{2}\int dx\, x^2 \left(S_{ij}(x,t) + S_{ij}(x,-t)\right) = D_{ij} t^2 + O(t)$. The coefficients defined in (2.15) are exactly the Drude weights of the model [93–96].

The sub-leading behaviour of the variance $\frac{1}{2}\int dx\, x^2 \left(S_{ij}(x,t) + S_{ij}(x,-t)\right)$ then depends on the behaviour of the time integrated current-current correlator. Indeed, as it follows from the sum rule (2.13), if the Onsager coefficients $\mathfrak{L}_{ij}$, defined by

$$\mathfrak{L}_{ij} := \lim_{t\to\infty} \int_{-t}^t ds \left( \int dx\, \langle j_i(x,s) j_j(0,0)\rangle^c - D_{ij} \right), \tag{2.16}$$

are finite, then the expansion (2.14) holds. Although eq.(2.16) has a form slightly different from Kubo–Mori inner product formula for diffusion constant, the latter reduces (under certain mild assumptions [97]) to standards grand-canonical averaging and therefore to equation (2.16).

We also note that one can derive similar expressions for the Drude weight and the Onsager matrix, but involving a mix of conserved densities and currents:

$$D_{ij} = \lim_{t\to\infty} \frac{1}{2t} \int dx\, x \left( \langle \mathsf{q}_i(x,t)\mathsf{j}_j(0,0)\rangle^c - \langle \mathsf{q}_i(x,-t)\mathsf{j}_j(0,0)\rangle^c \right) \tag{2.17}$$

and

$$\mathfrak{L}_{ij} = \lim_{t\to\infty} \left[ \int dx\, x \left( \langle \mathsf{q}_i(x,t)\mathsf{j}_j(0,0)\rangle^c - \langle \mathsf{q}_i(x,-t)\mathsf{j}_j(0,0)\rangle^c \right) - D_{ij}t \right]. \tag{2.18}$$

If the coefficients $D_{ij}$ diverge (i.e. the limits do not exist), then the ballistic description breaks down. If the coefficients $\mathfrak{L}_{ij}$ diverge, the diffusive expansion breaks down and the model is expected to display super-diffusion [98, 99].

## 2.3 Hydrodynamics and two-point functions

The coefficients $\mathfrak{L}_{ij}$ are related to the diffusion matrix introduced in (2.9). This can be seen by looking at the equation of motion for the two point function $S_{ij}(x,t)$. Indeed, within the hydrodynamic approximation, the derivative expansion (2.9) of the current $\langle \mathsf{j}_i(x,t)\rangle$ implies that the two-point density correlation functions satisfy [64] (see Appendix B)

$$\begin{aligned}
\partial_t S_{ij}(x,t) + \left(A_i{}^k \partial_x - \frac{1}{2}\mathfrak{D}_i{}^k \partial_x^2\right) S_{kj}(x,t) = 0 \quad \text{for} \quad t > 0, \\
\partial_t S_{ij}(x,t) + \left(A_j{}^k \partial_x + \frac{1}{2}\mathfrak{D}_j{}^k \partial_x^2\right) S_{ik}(x,t) = 0 \quad \text{for} \quad t < 0,
\end{aligned} \tag{2.19}$$

with the flux Jacobian defined as

$$A_i{}^j := \frac{\partial \langle \mathsf{j}_i\rangle}{\partial \langle \mathsf{q}_j\rangle} = \frac{\partial \mathcal{F}_i(\bar{\mathsf{q}}.)}{\partial \bar{\mathsf{q}}_j}. \tag{2.20}$$

Eq.(2.19) is valid on a homogeneous stationary state only, with mean densities $\bar{\mathsf{q}}_j$ independent of space and time $x, t$. Of course both $A_i{}^k$ and $\mathfrak{D}_i{}^k$ depend on those stationary mean densities $\bar{\mathsf{q}}_j$.

Operating with $\int_0^t ds \int dx\, x$ and integrating by part, we obtain, for $t > 0$,

$$\int dx\, x\, S_{ij}(x,t) = A_i{}^k \int_0^t ds \int dx\, S_{kj}(x,s) + E_{ij} = (A_i{}^k C_{kj})\, t + E_{ij}, \tag{2.21}$$

where we used that $\int dx\, S_{kj}(x,s)$ is independent of $s$ by the conservation laws, and equals $C_{kj}$ by definition, and where

$$E_{ij} = \int dx\, x\, S_{ij}(x,0). \tag{2.22}$$

One can show that [64]

$$A_i{}^k C_{kj} = C_{ik}A_j{}^k, \tag{2.23}$$

and therefore (2.21) holds for both $t > 0$ and $t < 0$.

Operating with $\int_0^t ds \int dx\, x^2$ with $t > 0$ and integrating by part again, we get

$$\int dx\, x^2 \big(S_{ij}(x,t) - S_{ij}(x,0)\big) = 2A_i^{\ k} \int_0^t ds \int dx\, x S_{kj}(x,s) \tag{2.24}$$
$$+ \mathfrak{D}_i^{\ k} \int_0^t ds \int dx\, S_{kj}(x,s),$$

while operating with $\int_0^{-t} ds \int dx\, x^2$, we find

$$\int dx\, x^2 \big(S_{ij}(x,-t) - S_{ij}(x,0)\big) = 2A_j^{\ k} \int_0^{-t} ds \int dx\, x S_{ik}(x,s) \tag{2.25}$$
$$- \mathfrak{D}_j^{\ k} \int_0^{-t} ds \int dx\, S_{ik}(x,s).$$

By integrating eq.(2.21), the first term in (2.24) is evaluated using $\int_0^t ds \int dx\ xS_{kj}(x,s) = \frac{1}{2}(A_k^{\ l} C_{lj}) t^2 + E_{kj} t$, and in (2.25) using $\int_0^{-t} ds \int dx\, xS_{ik}(x,s) = \frac{1}{2}(A_i^{\ l} C_{lk}) t^2 - E_{ik} t$. By the same argument as above, the last term is proportional to time and in (2.24) and (2.25) equals $(\mathfrak{D}_i^{\ k} C_{kj}) t$ and $(C_{ik} \mathfrak{D}_j^{\ k}) t$, respectively. Adding (2.24) and (2.25) and using (2.23) again, the hydrodynamic equation (2.19) for the two-point function therefore implies that

$$\frac{1}{2} \int dx\, x^2 \big(S_{ij}(x,t) + S_{ij}(x,-t) - 2S_{ij}(x,0)\big)$$
$$= (A_i^{\ k} A_k^{\ l} C_{lj}) t^2 + \frac{1}{2}(\mathfrak{D}_i^{\ k} C_{kj} + C_{ik}\mathfrak{D}_j^{\ k} + A_i^{\ k} E_{kj} - E_{ik} A_j^{\ k}) t. \tag{2.26}$$

Of course sub-leading terms in $O(t)$ would have been included if we would have pushed the hydrodynamic expansion further to include higher order derivatives.

As a consequence, the Drude weights and the Onsager coefficients are related to the diffusion matrix via the matrix of susceptibilities $C_{ij}$, up to terms proportional to $E_{ij}$,

$$D_{ij} = A_i^{\ k} A_k^{\ l} C_{lj}, \quad \mathfrak{L}_{ij} = \frac{1}{2}\Big(\mathfrak{D}_i^{\ k} C_{kj} + C_{ik}\mathfrak{D}_j^{\ k} + A_i^{\ k} E_{kj} - E_{ik} A_j^{\ k}\Big). \tag{2.27}$$

## 2.4 Gauge fixing and $\mathcal{PT}$-symmetry

The derivations in the previous two subsections are completely general. However, there is an *ambiguity* in the definition of the quantities that describe the fluid beyond the Euler scale. This is because the conserved densities $\mathsf{q}_i(x,t)$ are only defined by their relation to the total conserved quantities $Q_i = \int dx\, \mathsf{q}_i(x,t)$, and thus are ambiguous under addition of total derivatives of local observables. Consider the "gauge transformation"

$$\mathsf{q}_i(x,t) \mapsto \mathsf{q}_i(x,t) + \partial_x \mathfrak{o}_i(x,t), \qquad \mathsf{j}_i(x,t) \mapsto \mathsf{j}_i(x,t) - \partial_t \mathfrak{o}_i(x,t). \tag{2.28}$$

It is clear from the definition of the static covariance matrix $C_{ij}$ and the flux Jacobian $A_i^{\ j}$ that these are invariant: they are properties of homogeneous, stationary states, which are unaffected by the transformation (2.28). As a consequence, by the first equation in (2.27), the Drude matrix is also invariant: all Euler scale quantities are invariant. By contrast, quantities defined at the diffusive scale may be affected. It is possible to show,

assuming the validity of the hydrodynamic projection [72, 101], that the Onsager coefficients $\mathcal{L}_{ij}$ are invariant under (2.28). This has a clear physical meaning: by (2.14), these coefficients represent the strength of the diffusive spreading of the microscopic correlations, something which is independent form the choice of local densities. However, the diffusion matrix $\mathfrak{D}_i^{\ j}$ and the matrix $E_{ij}$ are *covariant*: they transform nontrivially under (2.28), in such a way as to make the combination on the right-hand side of the second equation of (2.27) invariant. The hydrodynamic approximation of the currents (2.9) is explicitly dependent on the choice of densities. See Appendix C. One must therefore choose a gauge in order to fix the diffusion matrix itself.

It turns out that there is a symmetry that allows us to fix a gauge in a very natural (and universal) way: $\mathcal{PT}$-symmetry. In quantum mechanics, $\mathcal{PT}$-symmetry is an anti-unitary involution $\mathsf{T}$ that preserves the Hamiltonian and the momentum operators. As a consequence, it has the effect of simultaneously inverting the signs of the space and time coordinates. In classical systems, it is the requirement that simultaneously inverting the signs of the space and time coordinates preserves the dynamics, the total energy and momentum. Let us consider a stronger version of $\mathcal{PT}$-symmetry: we require that all conserved quantities $Q_i$ be invariant, and that the $\mathcal{PT}$-transform of a local observable be a local observable.

A consequence of this strong version of $\mathcal{PT}$-symmetry is that homogeneous, stationary, maximal entropy states are $\mathcal{PT}$-symmetric. Another consequence is that[2]

$$\mathsf{T}\mathfrak{q}_i(x,t)\mathsf{T}^{-1} = \mathfrak{q}_i(-x,-t) + \partial_x \mathfrak{a}_i(-x,-t) \tag{2.29}$$

for some local observables $\mathfrak{a}_i$. We show in Appendix C that *there exists a unique gauge choice (under the gauge transformation* (2.28)*) such that*

$$\mathsf{T}\mathfrak{q}_i(x,t)\mathsf{T}^{-1} = \mathfrak{q}_i(-x,-t), \tag{2.30}$$

and that in this gauge choice, it is possible to further choose $\mathfrak{j}_i(x,t)$ such that

$$\mathsf{T}\mathfrak{j}_i(x,t)\mathsf{T}^{-1} = \mathfrak{j}_i(-x,-t). \tag{2.31}$$

This gauge choice simplifies drastically the equations of the previous two subsections. Indeed, a direct consequence is that

$$\frac{1}{2}\int \mathrm{d}x\, x^2 \big(S_{ij}(x,t) + S_{ij}(x,-t) - 2S_{ij}(x,0)\big) = \int \mathrm{d}x\, x^2 \big(S_{ij}(x,t) - S_{ij}(x,0)\big), \tag{2.32}$$

thus simplifying the left-hand side (2.13). Another consequence is that $E_{ij}$, defined in (2.22), is equal to the negative of itself, hence must vanish,

$$E_{ij} = 0. \tag{2.33}$$

Finally, applying $\mathcal{PT}$-symmetry on the left-hand side of (2.24), we obtain the left-hand side of (2.25), and thus we conclude that

$$\mathfrak{D}_i^{\ k} C_{kj} = C_{ik} \mathfrak{D}_j^{\ k}. \tag{2.34}$$

---

[2]Here we use a notation from quantum mechanics for the symmetry transformation, but the same holds in classical systems as well.

This shows that (2.27) simplifies to

$$D_{ij} = A_i{}^k A_k{}^l C_{lj}, \quad \mathfrak{L}_{ij} = \mathfrak{D}_i{}^k C_{kj}. \tag{2.35}$$

The strong version of $\mathcal{PT}$-symmetry is in fact extremely natural, and is expected to hold in many Gibbs states and Galilean and relativistic boosts thereof, and many generalised Gibbs ensembles [3]. Below we assume that this symmetry holds, and that the above gauge choice has been made. Note that the expression for $\mathfrak{D}_i{}^j$ obtained by inverting the second equation in (2.35) is the most direct generalization of the usual Green-Kubo formula for the diffusion constant of a single conserved quantity [97] to systems with an infinite number of conserved quantities. We will use the formula $\mathfrak{L}_{ij} = \mathfrak{D}_i{}^k C_{kj}$ together with eq.(2.16) in order to compute the diffusion coefficients.

### 2.5 Quantities with vanishing diffusion

In models with Galilean invariance which preserve particle number, the current $\mathfrak{j}_0$ of the conserved mass density $\mathfrak{q}_0$ equals the momentum density $\mathfrak{q}_1$. In relativistic models, the current of the conserved energy equals the momentum density. That is, in both cases, with $\mathfrak{q}_0$ either the mass density or energy density, we have the relation

$$\mathfrak{j}_0 = \mathfrak{q}_1. \tag{2.36}$$

In general, whenever the current of a conserved quantity is itself a conserved density, then it is a straightforward consequence of the above discussion that the part of the diffusion matrix associated with this conserved quantity (e.g. the mass (energy) in Galilean (relativistic) model) vanishes:

$$\mathfrak{D}_0{}^i = 0. \tag{2.37}$$

Indeed, in (2.16) the integral

$$\int dx \, \langle \mathfrak{j}_0(x,s) \mathfrak{j}_j(0,0) \rangle^c = \int dx \, \langle \mathfrak{q}_0(x,s) \mathfrak{j}_j(0,0) \rangle^c \tag{2.38}$$

is independent of $s$ by conservation, and therefore by (2.15) equals the Drude weight $D_{0j}$. As a consequence, the Onsager matrix elements $\mathfrak{L}_{0j}$ vanish, and by (2.35) this implies (2.37).

As we discuss below, in a large family of integrable models, even those which are not Galilean or relativistic invariant, there exists such a conserved quantity which has zero diffusion. In particular, in the XXZ model, it is well known that the current of energy is itself one of the conserved densities in the infinite tower afforded by integrability, and thus does not diffuse.

## 3 Quasiparticles, stationary states, and thermodynamic form factors

As we have seen in section 2, the main ingredients in order to formulate a hydrodynamic theory at large scales are the large scale connected correlations of local charges and their

---

[3]The identification of the charge densities $\bar{\mathfrak{q}}_i(x,t)$ with the densities of quasiparticle, see eq. (3.25), requires $\mathcal{PT}$-symmetry for the charge densities since the quasiparticle densities are indeed $\mathcal{PT}$-symmetric. However we stress that the hydrodynamical description given by eq. (2.10) is valid for any chosen gauge.

associated current. In particular, according to (2.15), (2.16) and (2.35), we need to evaluate the two-point function

$$\Gamma_{ij}(x,t) = \langle \mathrm{j}_i(x,t)\mathrm{j}_j(0,0)\rangle^c \qquad (3.1)$$

on a generic homogeneous stationary state given by the set of expectation values of local densities $\{\bar{\mathrm{q}}_i\}$, and then only at the end we shall promote this function to space and time dependence.

The aim of this section is to introduce the main objects for describing such states in integrable models, and the techniques to compute correlation functions in such states using the excitation of the system in the thermodynamic limit. In a wide family of integrable systems indeed, homogeneous stationary states admit an efficient description in terms of "quasiparticles", based on the thermodynamic Bethe ansatz (TBA). They are often referred to as generalised Gibbs ensembles (GGEs), which we will understand as a TBA state characterised by quasiparticle density (denoted $\rho_{\mathrm{p}}(\theta)$ below). This is then used as a basis for developing the hydrodynamics of integrable systems, generalised hydrodynamics (GHD). The description is expected to be very general, encompassing both quantum and classical models, and including field theories and chains. We recall the main aspects in subsection 3.1, and the GHD built from this in subsection 3.2.

The techniques to compute correlation functions are introduced in subsection 3.3, and used, as a check, in subsection 3.4 in order to re-obtain known results at the Euler scale. These techniques are based on the concept of particle and hole excitations above finite-density, TBA states. Although the TBA description of stationary states is expected to apply to a wide range of integrable models, to our knowledge, the understanding of particle-hole excitations is restricted to quantum Bethe-ansatz models with fermionic excitations. For these two sections, we thus restrict ourselves to this case.

The main derivation presented in the next section, for the diffusion matrix, uses the particle-hole excitation techniques, and is thus restricted to quantum fermionic excitations. The result is generalised to other quantum and classical integrable models, namely to the full range of application of GHD, and verified by comparing with the results obtained independently in the hard rod gas [80, 100].

## 3.1 Quasiparticles and homogeneous stationary states

Let us consider first a generic homogeneous integrable quantum model at equilibrium on a ring of length $L$. In the Bethe ansatz description, any eigenstate is specified by a set of real or complex "rapidities" (or Bethe roots) $\{\theta_j\}_{j=1}^N$, interconnected through nonlinear equations, the Bethe ansatz equations. For simplicity we first here consider those cases where all the states are characterized by real rapidities. Cases with complex rapidities shall be treated in Appendix G. In the thermodynamic limit $L \to \infty$ at fixed density $N/L$, the rapidities $\theta_j$ become dense on the real line, and therefore eigenstates can be described by densities of rapidities, $\rho_{\mathrm{p}}(\theta) = \lim_{L\to\infty} L^{-1}\big(\theta_{\ell(\theta,L)+1} - \theta_{\ell(\theta,L)}\big)^{-1}$ with $\theta_{\ell(\theta,L)+1} < \theta < \theta_{\ell(\theta,L)}$ which we also denote as density of quasiparticle. Such a macroscopic description of the eigenstates, given only in terms of the density $\rho_{\mathrm{p}}(\theta)$, neglects an exponential amount of information: many states lead to the same density. Defining as usual, informally, the entropy density of the macrostate $s[\rho_{\mathrm{p}}]$ as the number of states, divided by $L$, in a shell surrounding the density $\rho_{\mathrm{p}}$, one may evaluate it in the

thermodynamic limit,

$$\lim_{L\to\infty} s[\rho_{\mathrm{p}}] = \int \mathrm{d}\theta\, \rho_{\mathrm{s}}(\theta)g(\theta). \tag{3.2}$$

In this expression, $g(\theta)$ a functional of the state that depends on the statistics of the quasiparticles and that we describe below, and the density of states $\rho_{\mathrm{s}}(\theta)$ quantifies the total number of modes with rapidities inside the interval $[\theta, \theta + \mathrm{d}\theta)$; in the Bethe ansatz, one also defines the density of holes $\rho_{\mathrm{h}} = \rho_{\mathrm{s}} - \rho_{\mathrm{p}}$.

The density of states is not independent of the quasiparticle density. Indeed, taking the derivative of the (normalised) scattering phase,

$$T(\theta, \alpha) = \frac{\mathrm{d}}{\mathrm{d}\theta} \frac{\log S(\theta, \alpha)}{2\pi \mathrm{i}}, \tag{3.3}$$

the density of state is given, as a consequence of the Bethe ansatz equations, by

$$\rho_{\mathrm{s}}(\theta) = \frac{p'(\theta)}{2\pi} + \int \mathrm{d}\alpha\, T(\theta, \alpha)\rho_{\mathrm{p}}(\alpha), \tag{3.4}$$

where $p(\theta)$ is the momentum of the quasiparticle $\theta$, and $p'(\theta)$ its rapidity derivative. Below we assume for simplicity that $p'(\theta) > 0$ and that the differential scattering phase is symmetric:

$$T(\theta, \alpha) = T(\alpha, \theta). \tag{3.5}$$

See Remark 2 below. One also defines the filling or occupation function by the ratio

$$n(\theta) = \frac{\rho_{\mathrm{p}}(\theta)}{\rho_{\mathrm{s}}(\theta)}. \tag{3.6}$$

It is a functional of the density $\rho_{\mathrm{p}}$, and provides a good description of the state as well. In particular, the state density is obtained from the filling function by solving

$$\rho_{\mathrm{s}}(\theta) = \frac{p'(\theta)}{2\pi} + \int \mathrm{d}\alpha\, T(\theta, \alpha)n(\alpha)\rho_{\mathrm{s}}(\alpha). \tag{3.7}$$

In an integral operator language, where $T$ is the integral operator with kernel $T(\theta, \alpha)$ and the function $n$ is seen as a diagonal operator, we have

$$2\pi\rho_{\mathrm{s}} = (1 - Tn)^{-1}p'. \tag{3.8}$$

The operator $(1 - Tn)^{-1}$ is the dressing of a function,

$$h^{\mathrm{dr}}(\theta) = ((1 - Tn)^{-1}h)(\theta). \tag{3.9}$$

In all models we are aware of, $\rho_{\mathrm{s}}(\theta) > 0$ is always defined to be a positive function.

Although the above description was based on the Bethe ansatz for quantum models, it has much wider generality, and applies also to classical integrable models [101]; equations (3.2) up to (3.9) are valid within this level of generality. In order to describe the functional $g(\theta)$ as well as many other quantities such as Euler-scale correlation functions and, as we will see, diffusion functionals, we need additional information about the quasiparticles: their statistics. For instance, in the Bethe ansatz description, they are usually fermions (where the hole density $\rho_{\mathrm{h}}$ makes sense), but they can also be bosons,

and in classical models they can be classical particles or classical fields. In the TBA formalism, the statistics enters into a free energy function $F(\epsilon)$; this is the free energy for "free-particle" modes of energy $\epsilon$, with the same statistics as that of the quasiparticles of the model. For instance, it is given by $-\log(1+e^{-\epsilon})$ for fermions, $\log(1-e^{-\epsilon})$ for bosons, $-e^{-\epsilon}$ for classical particles, $1/\epsilon$ for classical radiative modes; see [101] for a discussion.

The statistics enters the functional $g(\theta)$, determining the entropy density $s[\rho_p]$, as follows. First, $g(\theta)$ is in fact a function of $n(\theta)$ only. In order to determine it, construct the pseudoenergy $\epsilon(\theta) = \epsilon(n(\theta))$ as a function of $n(\theta)$ by inverting the relation $n = dF(\epsilon)/d\epsilon$. Then $g$ is given by

$$g = (\epsilon + c)n - F(\epsilon), \tag{3.10}$$

with some physically unimportant constant $c$. We note that, seen as a function of $n$, $g$ satisfies

$$\frac{dg}{dn} = \epsilon(n) + c. \tag{3.11}$$

A macrostate specified by a distribution $\rho_p(\theta)$ is in the microcanonical ensemble. By a slight abuse of notation, we will denote the macrostate using the quantum "ket" notation $|\rho_p\rangle$[4]. As usual in thermodynamics, one expects this to be equivalent to the (grand) canonical description. In integrable systems, this is the so-called generalised Gibbs ensembles (GGEs), formally with density matrix proportional to $e^{-\sum_i \beta_i Q_i}/Z$ exactly as in (2.3) but now with an infinite sum over all conserved quantities $Q_i$ [103, 104]:

$$e^{-\sum_i \beta^i Q_i}/Z \longleftrightarrow |\rho_p\rangle\langle\rho_p|. \tag{3.12}$$

That is, given any local operator $\mathfrak{o}$, in the thermodynamic limit,

$$\lim_{L\to\infty} Z^{-1}\text{Tr}\left(e^{-\sum_i \beta^i Q_i}\,\mathfrak{o}\right) = \langle\rho_p|\mathfrak{o}|\rho_p\rangle. \tag{3.13}$$

The Lagrange parameters $\beta^i$ fix the function $\rho_p(\theta)$ by means of a non-linear integral equations. Let $h_i(\theta)$ be the one-particle eigenvalues if the conserved charges $Q_i$, that is $Q_i|\theta\rangle = h_i(\theta)|\theta\rangle$. Then the pseudoenergy solves

$$\epsilon(\theta) = d(\theta) + \int d\alpha\, T(\theta, \alpha) F(\epsilon(\alpha)), \tag{3.14}$$

where $d(\theta) = \sum_i \beta^i h_i(\theta)$. The expression $e^{-\sum_i \beta^i Q_i}$ for a GGE is formal, as one would need to specify the set of charges $Q_i$ and the convergence properties. More accurately, one instead considers the function $d(\theta)$ for characterising the GGE, independently from any series expansion. The specific free energy takes the general form

$$\int \frac{d\theta}{2\pi} p'(\theta) F(\epsilon(\theta)), \tag{3.15}$$

---

[4]As mentioned, a macrostate embodies an averaging inside a small shell of microscopic states. By a generalisation of the eigenstate thermalisation hypothesis [102] to integrable systems [103,104], one would expect a single state within this shell to give rise to the same local averages as those evaluated from the macrostate, whence this notation.

and from it all averages of conserved densities can be evaluated by differentiation with respect to $\beta^i$, giving the standard TBA formula

$$\langle \rho_{\mathrm{p}} | \mathsf{q}_i | \rho_{\mathrm{p}} \rangle = \int \mathrm{d}\theta \, h_i(\theta) \rho_{\mathrm{p}}(\theta). \tag{3.16}$$

One can also show that the entropy density satisfies the correct thermodynamic equation relating it to the conserved densities and the specific free energy,

$$s[\rho_{\mathrm{p}}] = \int \mathrm{d}\theta \, \rho_{\mathrm{p}}(\theta) d(\theta) - \int \frac{\mathrm{d}\theta}{2\pi} p'(\theta) \mathsf{F}(\epsilon(\theta)). \tag{3.17}$$

The quasiparticle densities not only specify the values of the conserved quantities, but also the expectation values of all the local operators, as they fully specify the state. One set of examples are the currents associated to the charge densities, as in (2.5). The expectation value of the currents on a generic homogeneous stationary state are given by

$$\langle \rho_{\mathrm{p}} | \mathsf{j}_i | \rho_{\mathrm{p}} \rangle = \int \mathrm{d}\theta \, \rho_{\mathrm{p}}(\theta) v^{\mathrm{eff}}(\theta) h_i(\theta), \tag{3.18}$$

where the effective velocities of the quasiparticles solve the linear integral equations

$$v^{\mathrm{eff}}(\theta) = \frac{E'(\theta)}{p'(\theta)} - \int \mathrm{d}\alpha \, \frac{T(\theta, \alpha)}{p'(\theta)} \rho_{\mathrm{p}}(\alpha)(v^{\mathrm{eff}}(\theta) - v^{\mathrm{eff}}(\alpha)), \tag{3.19}$$

with $E(\theta)$ the single-particle energy. It can be shown that this expression is equivalent to

$$v^{\mathrm{eff}}(\theta) = \frac{(E')^{\mathrm{dr}}(\theta)}{(p')^{\mathrm{dr}}(\theta)}. \tag{3.20}$$

This formula was proven in the context of integrable field theories in [49] and more recently in [105] (see also [106]) and in Appendix D we also provide an alternative derivation based on the dressed form factors given in this paper. More generally, the expectation value of any local operators on a homogeneous stationary state $\langle \rho_{\mathrm{p}} | \mathfrak{o} | \rho_{\mathrm{p}} \rangle$ is given by some complicated functional of the root densities $\rho_{\mathrm{p}}$. These are however much harder to compute, and only few expressions are available. For example, in the Lieb-Liniger gas there has been recent developments [107–109], while in the XXZ chain only few observables (beyond conserved densities and currents) can be computed [110, 111].

**Remarks:**

1. **Many quasiparticle types.** Generically, TBA (and the related Euler-scale GHD recalled below) must take into account many quasiparticle types emerging in the thermodynamic description, either as "bound states" seen as string configurations (or modifications thereof) in quantum TBA, or simply from the various particle types present in the microscopic model itself (for instance, in the asymptotic states of a QFT). In all cases, the differential scattering kernel takes the form $T_{a,b}(\theta_1, \theta_2)$ for quasiparticle types $a$, $b$ at rapidities $\theta_1$, $\theta_2$, respectively. Likewise, the rapidity in every TBA object is replaced by a doublet $(\theta, a)$ composed of a rapidity and a quasiparticle type. One then simply replaces each rapidity integral by the combination of a rapidity integral and a sum over quasiparticle types,

$$\int \mathrm{d}\theta \mapsto \sum_a \int \mathrm{d}\theta. \tag{3.21}$$

That is, all formula stay valid with $\int d\theta$ understood as an integration on an appropriate manifold – the spectral manifold of the model.

2. **Reparametrisation.** In the above, we assumed that the differential scattering kernel $T(\theta, \alpha)$ was symmetric, and that $p'(\theta) > 0$. In fact, all equations of the thermodynamic Bethe ansatz reviewed here can be written in a way that is invariant under reparametrisation $\theta \mapsto u(\theta)$ with $u'(\theta) > 0$. From (3.3), it is clear that $T(\theta, \alpha)$ is a vector field in the first argument, and a scalar in its second [101], and thus it is generically not symmetric. Likewise, $p'(\theta)$ and $E'(\theta)$ are vector fields, and the quantities $h_i(\theta)$ in (3.16) and (3.18), as well as the effective velocity $v^{\text{eff}}(\theta)$, are scalar fields. If we also consider reparametrisations that do not necessarily preserve the direction – that is, either $u'(\theta) > 0$ for all $\theta$, or $u'(\theta) < 0$ for all $\theta$ –, then generically $p'(\theta)$ may be negative, although it always has a definite, $\theta$-independent sign. In such cases, the covariant (1-form) integration measure is $d\theta\sigma$, where $\sigma$ is a pseudoscalar, changing sign under direction-inverting reparametrisations (with many quasiparticle types, $\sigma$ is independent of $\theta$, but may depends on the quasiparticle type $a$). Conventionally, one always take $\rho_p$ and $\rho_s$ to be positive quantities, and thus these are pseudovector fields. With these rules, it is a simple matter to generalise all equations to the case of an arbitrary parametrisation of the spectral space. For instance

$$w^{\text{dr}} = (1 - Tn\sigma)^{-1}w, \qquad h^{\text{dr}} = (1 - T^{\text{T}}n\sigma)^{-1}h \tag{3.22}$$

if $w$ is a vector field and $h$ is a scalar field, where $T^{\text{T}}(\theta, \alpha) = T(\alpha, \theta)$ is the transposed kernel (a vector (scalar) field in its second (first) argument), and

$$(p')^{\text{dr}} = 2\pi\sigma\rho_s. \tag{3.23}$$

3. **Parity symmetry.** In models with parity symmetry, it is expected that it is possible to choose a parametrisation, not necessarily direction-preserving, such that the differential scattering phase becomes symmetric. This is the case in the XXZ chain, in the Lieb-Liniger model, in the hard rod gas, and in many other field theories. With a symmetric choice of $T$, in the gapped spin XXZ chain all $\sigma_a$ (for quasiparticle types $a$) are equal to 1, however nontrivial parities occur in the gapless regime at roots of unity [112] or in fermionic models like the Fermi-Hubbard chain [113] (these therefore also occur, under this symmetric parametrisation choice, in the description of the local stationary state at the Euler scale, see [50, 114]). Such signs are then interpreted as the parities of the quasiparticles involved. We emphasise, however, that it is always possible to choose a parametrisation where no such signs occur, at the price, in general, of a non-symmetric $T$.

4. **Quantities with vanishing diffusion.** Whenever there is a choice of parametrisation such that $T$ is symmetric, then one can argue that, in this choice of parametrisation, the quantity associated to the one-particle eigenvalue $h_i(\theta) = p'(\theta)$ has vanishing diffusion. Indeed, with this choice, the GGE average current (3.18), which can also be written in general as

$$\langle \rho_p|j_i|\rho_p\rangle = \int \frac{d\theta}{2\pi}(E')^{\text{dr}}(\theta)n(\theta)p'(\theta) = \int \frac{d\theta}{2\pi}E'(\theta)n(\theta)(p')^{\text{dr}}(\theta) \tag{3.24}$$

becomes equal to the GGE average density of the quantity associated with $h_j(\theta) = E'(\theta)$, as is clear from (3.16). If this equality holds as an operator identity beyond GGE averages, then the argument presented in subsection 2.5 shows that $\mathfrak{D}_i{}^k = 0$ for all $k$. We will show below, by explicitly calculating the diffusion operator, that these elements of the diffusion matrix indeed vanish.

## 3.2 Local stationary states and GHD

In the previous section we described homogenous stationary states. As explained in section 2, in the context of hydrodynamics we need to characterise local averages in inhomogeneous situations. In inhomogeneous states, the TBA approach above does not hold anymore. However, the hydrodynamic approximation postulates that the values of $\bar{q}_i(x, t) = \langle q_i(x, t) \rangle$, on a fixed time slice $t$, completely determine the state. The first step in the hydrodynamic theory for integrable systems is to use the form on the right-hand side of (3.16) as a definition for space-time dependent "densities" $\rho_p(\theta; x, t)$ determining the state:

$$\bar{q}_i(x, t) =: \int d\theta \, \rho_p(\theta; x, t) h_i(\theta). \tag{3.25}$$

The quantity $\rho_p(\theta; x, t)$, as a function of $\theta$, is in general *no longer* a Bethe ansatz root density in quantum models; it is instead a way of representing the averages of conserved densities in space-time, and relation (3.25) is assumed to be an invertible ($x, t$-dependent) map $\bar{q}.(x, t) \mapsto \rho_p(\cdot; x, t)$.

At the Euler scale, in the limit of infinitely large variation lengths, the local state can be understood as a GGE. In this case, $\rho_p(\theta; x, t)$, as a function of $\theta$, is interpreted as a space-time dependent Bethe ansatz root density, and therefore all local observables take their GGE form with respect to $\rho_p(\theta; x, t)$:

$$\langle \mathfrak{o}(x, t) \rangle \overset{\text{Euler}}{=} \langle \rho_p(x, t) | \mathfrak{o} | \rho_p(x, t) \rangle. \tag{3.26}$$

In particular, recalling $\bar{j}_i(x, t) = \langle j_i(x, t) \rangle$, we have

$$\bar{j}_i(x, t) \overset{\text{Euler}}{=} \mathcal{F}_i(\bar{q}.(x, t)) = \int d\theta \, \rho_p(\theta; x, t) v^{\text{eff}}(\theta) h_i(\theta). \tag{3.27}$$

However, going beyond the Euler scale, namely adding the Navier-Stokes (NS) corrections as in eq.(2.9), extra terms occur in averages of generic observables that depend on the space derivative of $\rho_p(\theta; x, t)$. At this scale, the local state described by $\rho_p(\theta; x, t)$ cannot be interpreted as a space-time dependent GGE. For the currents, we define the integral operator $\mathfrak{D}[\rho_p]$, with kernel $\mathfrak{D}[\rho_p](\theta, \alpha)$, via the expansion

$$\bar{j}_i(x, t) \overset{\text{NS}}{=} \int d\theta \, h_i(\theta) \left( \rho_p(\theta; x, t) v^{\text{eff}}(\theta) - \frac{1}{2} \int d\alpha \, \mathfrak{D}[\rho_p(\cdot; x, t)](\theta, \alpha) \, \partial_x \rho_p(\alpha; x, t) \right), \tag{3.28}$$

and a similar modification is expected for any local operator,

$$\langle \mathfrak{o}(x, t) \rangle \overset{\text{NS}}{=} \langle \rho_p(x, t) | \mathfrak{o} | \rho_p(x, t) \rangle + \int d\alpha \, \mathfrak{D}_{\mathfrak{o}}[\rho_p(\cdot; x, t)](\alpha) \, \partial_x \rho_p(\alpha; x, t),$$

where the "diffusion functionals" $\mathfrak{D}_{\mathfrak{o}}[\rho_p(\cdot; x, t)](\alpha)$ are not known even for simple operators. Our main result is the derivation of the exact diffusion functionals for the currents.

### 3.3 Particle-hole excitations and correlation functions

We now specialise the above description to quantum integrable models with fermionic statistics,

$$\mathsf{F}(\epsilon) = -\log(1 + e^{-\epsilon}).$$

This includes for instance the Lieb-Liniger model and the XXZ chain.

One way to compute two-point correlation functions in a generic reference state $|\Omega\rangle$ is by inserting a resolution of the identity between the two operators and summing over all the intermediate states $s$, with momentum $P_s$ and energy $E_s$:

$$\langle\Omega|\mathfrak{o}_i(x,t)\mathfrak{o}_j(0,0)|\Omega\rangle = \sum_s \langle\Omega|\mathfrak{o}_i|s\rangle\langle s|\mathfrak{o}_j|\Omega\rangle e^{ix(P_s-P_\Omega)-it(E_s-E_\Omega)}, \qquad (3.29)$$

where here and below, for any local operator we denote $\mathfrak{o}(x=0, t=0) \equiv \mathfrak{o}$. Let us consider the thermodynamic reference state $|\Omega\rangle = |\rho_\mathrm{p}\rangle$ the quasiparticle state with root density $\rho_\mathrm{p}(\theta)$. The spectral sum is in principle very hard to compute. However, whenever the operators $\mathfrak{o}_j$ are local and conserve the total number of particles, the only non-zero contributions to the sum are the so-called particle-hole excitations. These are given by microscopic changes of a rapidities, namely a set of holes $\theta_\mathrm{h}^j$, $j = 1, \ldots, m$ belonging to the reference state is replaced by a new set of rapidities, the particles $\theta_\mathrm{p}^j$, and *vice versa*. The spectral sum then organises into a sum over numbers $m$ of particle-hole excitations, see Fig. 1, and can be formally written as

$$
\begin{aligned}
\langle\rho_\mathrm{p}|\mathfrak{o}_i(x,t)&\mathfrak{o}_j(0,0)|\rho_\mathrm{p}\rangle^c \\
&= \sum_{m=1}^{\infty} \frac{1}{(m!)^2} \left( \prod_{j=1}^{m} \int_\mathbb{R} \mathrm{d}\theta_\mathrm{p}^j \rho_\mathrm{h}(\theta_\mathrm{p}^j) \fint \mathrm{d}\theta_\mathrm{h}^j \rho_\mathrm{p}(\theta_\mathrm{h}^j) \right) \\
&\quad \times \langle\rho_\mathrm{p}|\mathfrak{o}_i|\{\theta_\mathrm{p}^\bullet,\theta_\mathrm{h}^\bullet\}\rangle\langle\{\theta_\mathrm{p}^\bullet,\theta_\mathrm{h}^\bullet\}|\mathfrak{o}_j|\rho_\mathrm{p}\rangle e^{ixk[\theta_\mathrm{p}^\bullet,\theta_\mathrm{h}^\bullet]-it\varepsilon[\theta_\mathrm{p}^\bullet,\theta_\mathrm{h}^\bullet]},
\end{aligned}
\qquad (3.30)
$$

with an appropriate regularised integral. This is part of the assumptions underlying the validity of the form factor expansion in the thermodynamic limit. The important point about the regularisation is that the form factor representation involves regularised integrations on the real axis only. See a more detailed discussion in the Appendix F.

The integration rapidities are the particle and hole excitations above the reference state, and the measure takes into account the weight of availability of such excitations, proportional, respectively, to the hole and particle densities. By the Bethe ansatz, the total momentum $P_s$ and energy $E_s$ are simply the sums of the individual momenta and energies of the particles and holes, with positive (negative) contributions for particles (holes). Let us denote the momentum and energy of an excitation at rapidity $\theta$ by $k(\theta)$ and $\varepsilon(\theta)$, respectively. Then

$$P_s = k[\theta_\mathrm{p}^\bullet, \theta_\mathrm{h}^\bullet] = \sum_{j=1}^{m} \left( k(\theta_\mathrm{p}^j) - k(\theta_\mathrm{h}^j) \right), \qquad (3.31)$$

$$E_s = \varepsilon[\theta_\mathrm{p}^\bullet, \theta_\mathrm{h}^\bullet] = \sum_{j=1}^{m} \left( \varepsilon(\theta_\mathrm{p}^j) - \varepsilon(\theta_\mathrm{h}^j) \right).$$

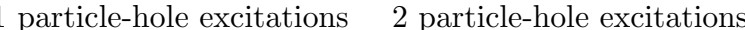

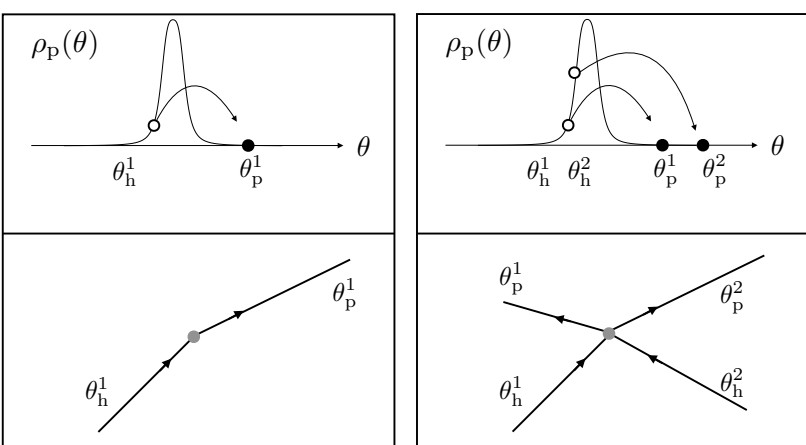

Figure 1: Schematic picture of the particle-hole excitations, entering the expansion for the two-point correlation functions (3.30). *Left*: one particle-hole excitations correspond to processes where one quasiparticle in the local steady state $|\rho_p\rangle$ with rapidity $\theta_h^1$ is changed to a new rapidity $\theta_p^1$ by the action of a local operator $\mathfrak{o}$. The excitation with zero momentum, $\theta_p^1 \to \theta_h^1$, can be regarded as the free propagation of a single quasiparticle with its effective velocity $v^{\text{eff}}(\theta_h^1)$. *Right*: two particle-hole excitations consist into two quasi-particles changing their rapidites to two new different values and it can be regarded as a 2-body scattering process among quasiparticles. Higher particle-hole excitations can be equivalently regarded as higher-body scattering processes.

These functions depend on the reference state via the so-called back-flow function. Namely, given the single-particle energy $E(\theta)$ and the single-particle momentum $p(\theta)$, we have

$$\varepsilon(\theta) = E(\theta) + \int d\alpha\, F(\theta, \alpha) E'(\alpha) n(\alpha)\,, \tag{3.32}$$

$$k(\theta) = p(\theta) + \int d\alpha\, F(\theta, \alpha) p'(\alpha) n(\alpha)\,, \tag{3.33}$$

with the back-flow $F(\theta, \alpha)$ being the amplitude of the global $1/L$ shift of the rapidities close to $\theta$ in the presence of the excitations $\alpha$, [115]. The back-flow is written in terms of the dressed scattering phase shift, more precisely

$$F = \frac{\log S}{2\pi i}(1 - nT)^{-1}\,. \tag{3.34}$$

In particular, one can show that

$$\varepsilon'(\theta) = (E')^{\text{dr}}(\theta)\,, \quad k'(\theta) = (p')^{\text{dr}}(\theta)\,, \tag{3.35}$$

where the dressing operation is defined in (3.9).

The series (3.30) can not be evaluated exactly in general. Some results for its asymptotic were found in [116–122], and the sum was numerically evaluated in the Lieb-Liniger model at finite temperature and for some specific operators in [123–125]. The

most complicated objects are the so-called *thermodynamic form factors* of local operators $\langle\rho_p|o_i|\{\theta_p^\bullet,\theta_h^\bullet\}\rangle$; these are evaluated in the thermodynamic limit, and thus are matrix elements on states with finite densities of excitations. The usual way of evaluating such objects is by evaluating matrix elements in a finite-size system, and taking the thermodynamic limit. Form factors of operators at finite sizes are generically not known in interacting models, except for few cases [126, 127]. Even in the cases where they are known, extracting their thermodynamic limit is a non-trivial task [118, 120, 128] and in field theories, together with the computation of finite temperature correlations, they constitute a long-standing open problem, see for example [119, 129–133]. While in field theory form factors on the vacuum state can be obtained via the so-called form factor bootstrap [134], still today it is not clear how to formulate a bootstrap protocol to obtain the thermodynamic form factors, although some attempts were formulated in [135–137]. Further, in the thermodynamic limit, particle-hole form factors generically contain the so-called kinematic poles [134, 138] on real rapidities, single poles when hole rapidities coincide with particle rapidities. Thus the limit has to be taken properly on the summations over discrete sets of particle and hole rapidities (the form factor expansion itself), not just on form factors. In some cases the regularisation of the integrals can be obtained explicitly by properly taking the thermodynamic limit of finite size regularisations [139, 140], however in general it is not known how to do this. Nevertheless, thanks to Eqs.(3.31), the poles at coinciding particle-hole rapidities can be reabsorbed into contributions to form factors with lower numbers of particle-hole pairs. The resulting integrals are Hadamard regularised, and away from coinciding rapidities, the integrand factorises as a product of functions of the form $\langle\rho_p|o_i|\{\theta_p^\bullet,\theta_h^\bullet\}\rangle\langle\{\theta_p^\bullet,\theta_h^\bullet\}|o_j|\rho_p\rangle$. See the Appendix F.

In interacting models, form factors of generic current or charge density operators on generic reference states are not known. There are however some special limiting cases where their form can be obtained from different methods or guessed from general principles. These are the small-momentum limit of the single-particle-hole form factors and the residue of the kinematic poles.

1. **Single-particle-hole form factors.** Single-particle-hole form factors of local densities and currents do not contain singularities and their small momentum limit is given only in terms of thermodynamic functions [120, 141]

$$\lim_{\theta_p\to\theta_h}\langle\rho_p|q_i|\theta_p,\theta_h\rangle = h_i^{dr}(\theta_h). \tag{3.36}$$

For the density operator in the Lieb-Liniger model this was first derived in [128]. Their general form can be inferred by comparing the expression of the susceptibilities from the thermodynamic Bethe ansatz with that from a form factor expansion. By using the non-linear relations between the Lagrange multiplier $\beta_j$ and the root densities $\rho_p(\theta)$ of a generic GGE, one finds [141]

$$\int dx\,\langle q_i(x,t)q_j(0,0)\rangle^c = -\frac{\partial\langle q_i\rangle}{\partial\beta_j} = -\int d\theta\,\frac{\rho_p(\theta)}{\partial\beta_j}h_i(\theta) \tag{3.37}$$

$$= \int d\theta\,\rho_p(\theta)(1-n(\theta))h_i^{dr}(\theta)h_j^{dr}(\theta). \tag{3.38}$$

The factor $1-n(\theta)$ is in fact, in general statistics, $-d\log n/d\epsilon$ [141], where we recall that $n = dF(\epsilon)/d\epsilon$. As we show in the next subsection, thanks to the continuity equations, leading to (3.40) and (3.40), in the form factor expansion for

susceptibilities of conserved densities, the infinite summation over the number of particle-hole pairs $m$ truncates to $m = 1$. As a consequence, using (3.30) we obtain

$$
\begin{aligned}
\int dx \, \langle q_i(x,t) q_j(0,0) \rangle^c &= \int dx \, \langle \rho_p | q_i(x,t) q_j(0,0) | \rho_p \rangle^c \\
&= (2\pi) \int d\theta_p d\theta_h \rho_p(\theta_h) \rho_h(\theta_p) \delta(k(\theta_p) - k(\theta_h)) \langle \rho_p | q_i | \theta_p, \theta_h \rangle \langle \rho_p | q_j | \theta_p, \theta_h \rangle \\
&= \int d\theta_h \rho_p(\theta_h)(1 - n(\theta_h)) \lim_{\theta_p \to \theta_h} \langle \rho_p | q_i | \theta_p, \theta_h \rangle \langle \rho_p | q_j | \theta_p, \theta \rangle,
\end{aligned}
\tag{3.39}
$$

where we used (3.35), (3.8) and (3.6). This then gives relation (3.36). Note that we used the fact that setting $k(\theta_p) = k(\theta_h)$ is equivalent to setting $\theta_p = \theta_h$ (monotonicity of $k(\theta)$).

2. **Higher particle-hole form factors.** In the limit where particles' rapidities are close to those of holes, there are singularities, the kinematic poles. These are set by integrability. Further, by the continuity equations, form factors of conserved densities and currents are related to each other. These statements are expressed as follows:

(a) *Continuity equations*: there exists a function $f_i(\{\theta_p^\bullet, \theta_h^\bullet\})$ such that

$$
\langle \rho_p | q_i | \{\theta_p^\bullet, \theta_h^\bullet\} \rangle = k[\theta_p^\bullet, \theta_h^\bullet] f_i(\{\theta_p^\bullet, \theta_h^\bullet\}),
\tag{3.40}
$$

$$
\langle \rho_p | j_i | \{\theta_p^\bullet, \theta_h^\bullet\} \rangle = \varepsilon[\theta_p^\bullet, \theta_h^\bullet] f_i(\{\theta_p^\bullet, \theta_h^\bullet\}).
\tag{3.41}
$$

These relations are obtained by using the continuity equations

$$
\partial_t \langle \rho_p | q_i(x,t) | \{\theta_p^\bullet, \theta_h^\bullet\} \rangle = -\partial_x \langle \rho_p | j_i(x,t) | \{\theta_p^\bullet, \theta_h^\bullet\} \rangle,
\tag{3.42}
$$

and the fact that quasiparticle states are eigenstate of the energy and momentum operators

$$
\langle \rho_p | q_i(x,t) | \{\theta_p^\bullet, \theta_h^\bullet\} \rangle = e^{ixk[\theta_p^\bullet, \theta_h^\bullet] - it\varepsilon[\theta_p^\bullet, \theta_h^\bullet]} \langle \rho_p | q_i | \{\theta_p^\bullet, \theta_h^\bullet\} \rangle.
\tag{3.43}
$$

At the level of one particle-hole pair, it is clear, from point (i) above, that

$$
f_i(\theta_p, \theta_h) \sim \frac{h_i^{dr}(\theta_h)}{k'(\theta_h)(\theta_p - \theta_h)}
\tag{3.44}
$$

as $\theta_p \to \theta_h$.

(b) *Kinematic poles (conjecture)*: the above pole structure of $f_i$ generalises to higher particles. The two particle-hole form factor of a local density satisfies, as $\theta_p^2 \to \theta_h^2$,

$$
\langle \rho_p | q_i | \theta_p^1, \theta_h^1, \theta_p^2, \theta_h^2 \rangle = \langle \rho_p | q_i | \theta_p^1, \theta_h^1 \rangle \frac{G(\theta_h^2 | \theta_p^1, \theta_h^1)}{\left(\theta_p^2 - \theta_h^2\right)} + O((\theta_p^2 - \theta_h^2)^0).
\tag{3.45}
$$

Further, the functions $G(\theta_{\rm h}^2|\theta_{\rm p}^1,\theta_{\rm h}^1)$ have an expansion around the line $\theta_{\rm p}^1 \sim \theta_{\rm h}^1$ as [5]

$$G(\theta_{\rm h}^2|\theta_{\rm p}^1,\theta_{\rm h}^1) = (\theta_{\rm p}^1 - \theta_{\rm h}^1)\frac{2\pi T^{\rm dr}(\theta_{\rm h}^1,\theta_{\rm h}^2)}{k'(\theta_{\rm h}^2)} + O((\theta_{\rm p}^1 - \theta_{\rm h}^1)^2), \qquad (3.46)$$

where

$$T^{\rm dr} = (1 - Tn)^{-1}T. \qquad (3.47)$$

Note that thanks to the assumed symmetry of $T(\theta,\alpha)$, this is related to the derivative of the backflow kernel by a transposition,

$$T^{\rm dr} = \left(\frac{{\rm d}F}{{\rm d}\theta}\right)^{\rm T}. \qquad (3.48)$$

We combine this pole structure and expansion statements with the continuity relation (3.40) and assume the *stronger* requirement that the function $f_i$ be a sum over the necessary poles of the form factors, plus a regular part. Equations (3.45) and (3.46) then impose the following:

$$f_i(\theta_{\rm p}^1,\theta_{\rm h}^1,\theta_{\rm p}^2,\theta_{\rm h}^2) = \frac{2\pi T^{\rm dr}(\theta_{\rm h}^2,\theta_{\rm h}^1)h_i^{\rm dr}(\theta_{\rm h}^2)}{k'(\theta_{\rm h}^1)k'(\theta_{\rm h}^2)(\theta_{\rm p}^1 - \theta_{\rm h}^1)} + \frac{2\pi T^{\rm dr}(\theta_{\rm h}^1,\theta_{\rm h}^2)h_i^{\rm dr}(\theta_{\rm h}^1)}{k'(\theta_{\rm h}^2)k'(\theta_{\rm h}^1)(\theta_{\rm p}^2 - \theta_{\rm h}^2)}$$
$$+ \frac{2\pi T^{\rm dr}(\theta_{\rm h}^2,\theta_{\rm h}^1)h_i^{\rm dr}(\theta_{\rm h}^2)}{k'(\theta_{\rm h}^1)k'(\theta_{\rm h}^2)(\theta_{\rm p}^2 - \theta_{\rm h}^1)} + \frac{2\pi T^{\rm dr}(\theta_{\rm h}^1,\theta_{\rm h}^2)h_i^{\rm dr}(\theta_{\rm h}^1)}{k'(\theta_{\rm h}^2)k'(\theta_{\rm h}^1)(\theta_{\rm p}^1 - \theta_{\rm h}^2)}$$
$$+ \text{regular}, \qquad (3.49)$$

where the regular terms have a well defined multi-variable Taylor expansion in $\theta_{\rm p,h}^{1,2}$ around any real values of these variables. One can easily check that this function, multiplied by $k[\theta_{\rm p}^\bullet,\theta_{\rm h}^\bullet]$ or $\varepsilon[\theta_{\rm p}^\bullet,\theta_{\rm h}^\bullet]$ as in equation (3.40) and (3.41), gives the correct residues (3.45) with the expansion (3.46). Equation (3.49) is expected to hold in general, without the assumption of symmetry of $T$ and with any parametrisation sign.

In the following we shall show that

i. Single particle-hole form factors in the small momentum limit completely determine the Euler-scale hydrodynamic theory.

ii. Two particle-hole form factors in the small momentum limit completely determine the diffusive corrections to the Euler-scale hydrodynamic theory.

Point (i) is shown in the next subsection, while we shall address point (ii) in the next section.

---

[5]This relation was found first for the thermodynamic form factor of the density operator in the Lieb-Liniger gas [120]. We here make the conjecture that this form is universal for any local operator. Notice that the same conjecture is also put forward in [137].

### 3.4 The Euler scale: single particle-hole excitations

We first recall the derivation presented in point (1) of the previous subsection for the static covariance matrix (or susceptibility matrix) $C_{ij} = \int dx \, \langle q_i(x,t) q_j(0,0) \rangle^c$. In the derivation (3.39), only the one particle-hole contribution was taken. Now consider the full form factor expansion (3.30) for $C_{ij}$. At any particle-hole number $m > 1$, note that the integration over $x$ produces $\delta(k[\theta_p^\bullet, \theta_h^\bullet])$. Note also the factor $k[\theta_p^\bullet, \theta_h^\bullet]$ in (3.40), which vanishes on the hyper-plane $k[\theta_p^\bullet, \theta_h^\bullet] = 0$. Since, at $m > 1$, the singularities of $f_i(\{\theta_p^\bullet, \theta_h^\bullet\})$ lie on a subset of measure zero of this hyper-plane, by the principal-value prescription they do not contribute. As a consequence, and the restriction on this hyper-plane imposed by $\delta(k[\theta_p^\bullet, \theta_h^\bullet])$ vanishes. This is not true for one particle-hole pair, as in this case the singularity lies exactly on this hyper-plane, (3.44), on which the resulting form factor has a finite limit.

The same argument can be used to evaluate the $B$-matrix [72]

$$B_{ij} = -\frac{\partial \langle j_i \rangle}{\partial \beta_j} = \int dx \, \langle j_i(x,t) q_j(0,0) \rangle^c \tag{3.50}$$

in order to obtain

$$B_{ij} = \int d\theta \, \rho_p(\theta)(1-n(\theta))v^{\text{eff}}(\theta)h_i^{\text{dr}}(\theta)h_j^{\text{dr}}(\theta) \tag{3.51}$$

in agreement with the expression found in [114, 141]. As recalled in section 2, the Euler-scale hydrodynamic theory is completely determined by the expression of the expectation values of the currents on a generic GGE state as in eq. (3.27):

$$\bar{j}_i(x,t) \overset{\text{Euler}}{=} \mathcal{F}_i(\bar{q}.(x,t)). \tag{3.52}$$

In particular, it is fully encoded within the flux Jacobian $A_i^{\;j} = \partial \mathcal{F}_i / \partial \bar{q}_j$ (2.20) given, in matrix notation, by [72, 141] as $A = BC^{-1}$. Therefore, the one particle-hole form factors determine $A_i^{\;j}$, hence the Euler hydrodynamics.

The Drude weights can also be obtained directly by using similar arguments as above, evaluating the expression (2.15) by a form factor expansion. Here, the space integral does not make higher particle-hole form factor contributions vanish, as the current form factors are proportional to $\varepsilon[\theta_p^\bullet, \theta_h^\bullet]$. However, the time integral in (2.15) provides the necessary delta function $\delta(\epsilon[\theta_p^\bullet, \theta_h^\bullet])$. At one particle-hole level, the space integral makes the term time-independent, on which the time average in (2.15) acts trivially. The result is then

$$D_{ij} = 2\pi \int d\theta_h \frac{\rho_p(\theta_h)\rho_h(\theta_h)}{k'(\theta_h)} \lim_{\theta_p \to \theta_h} \langle \rho_p | j_i | \theta_p, \theta_h \rangle \langle \theta_p, \theta_h | j_j | \rho_p \rangle \tag{3.53}$$

$$= \int d\theta \, \rho_p(\theta)(1-n(\theta))(v^{\text{eff}}(\theta))^2 h_i^{\text{dr}}(\theta)h_j^{\text{dr}}(\theta) \tag{3.54}$$

using (3.41), (3.44), (3.20) and (3.35). This is again in agreement with the results found in [114, 141].

That is, we have shown that the single particle-hole contribution completely determines the Euler-scale hydrodynamic coefficients, as under spatial or temporal integrations all the higher particle-hole contributions vanish.

## 4 Diffusion matrix

In this section we derive an expression for the diffusion matrix by using the definition (2.16) of the Onsager matrix as time integrated current-current correlator, and its relation (2.35) with the diffusion matrix. This relation is valid at least under the $\mathcal{PT}$-symmetry assumption, which we expect to hold in many integrable models of interest (including the classical hard rod gas, the quantum Lieb-Liniger model, and the quantum Heisenberg chain, isotropic and anisotropic) and for the choices of conserved density and current operators with form factors as described in subsection 3.3. We compute the integrated correlation function by representing it as a sum over particle-hole excitations as in equation (3.30) and show that only the terms with two particle-hole excitations contribute to the full Onsager matrix. We then generalise the result, by analogy, to arbitrary integrable models (with arbitrary number of quasiparticle types, and in an arbitrary spectral space parametrisation), including classical models.

### 4.1 Exact diffusion matrix from two-particle-hole excitations

In this subsection, we assume that there is a single quasiparticle type, but we keep the differential scattering kernel generic (that is, not necessarily symmetric). With a single quasiparticle type, it is somewhat superfluous to consider nontrivial parametrisation parities, however for the sake of generalisation to many quasiparticle types, see Section 4.2, it is convenient to keep the parity arbitrary, with

$$\sigma = \text{sgn}(k'(\theta)). \tag{4.1}$$

We focus now on the $\mathfrak{D}C$ operator, namely on the integrated $\Gamma_{ij}(x,t)$ (see (3.1)) as per (2.35) and (2.16). Expanding in particle-hole contributions as in (3.30) we have, in a self-explanatory notation,

$$\Gamma_{ij}(x,t) = [jj]^{1\text{ph}}_{ij}(x,t) + [jj]^{2\text{ph}}_{ij}(x,t) + \dots. \tag{4.2}$$

As shown in the previous section, the single-particle-hole contribution gives, under space integral and then time average, the Drude weight coefficients. In fact, recall that the one-particle-hole contribution is, under space integral, time-independent, as the factor $\delta(k(\theta_\text{p}) - k(\theta_\text{h}))$ imposes $\theta_\text{p} = \theta_\text{h}$. Therefore

$$\int_{-t}^{t} \text{d}s \int \text{d}x \, [jj]^{1\text{ph}}_{ij}(x,s) = 2t D_{ij}, \tag{4.3}$$

so that only higher-particle-hole form factors contribute to the Onsager matrix.

We first consider the two-particle-hole contribution

$$\lim_{t\to\infty} \int_{-t}^{t} \text{d}s \int \text{d}x \, [jj]^{2\text{ph}}_{ij}(x,t) =$$
$$\lim_{t\to\infty} \frac{(2\pi)^2}{2!^2} \int \text{d}\theta_\text{h}^1 \text{d}\theta_\text{h}^2 \rho_\text{p}(\theta_\text{h}^1)\rho_\text{p}(\theta_\text{h}^2) \fint \text{d}\theta_\text{p}^1 \text{d}\theta_\text{p}^2 \rho_\text{h}(\theta_\text{p}^1)\rho_\text{h}(\theta_\text{p}^2)$$
$$\times \delta(k[\theta_\text{p}^\bullet, \theta_\text{h}^\bullet])\delta_t(\varepsilon[\theta_\text{p}^\bullet, \theta_\text{h}^\bullet])\langle \rho_\text{p}|j_i|\theta_\text{p}^1, \theta_\text{h}^1, \theta_\text{p}^2, \theta_\text{h}^2\rangle\langle\theta_\text{p}^1, \theta_\text{h}^1, \theta_\text{p}^2, \theta_\text{h}^2|j_j|\rho_\text{p}\rangle, \tag{4.4}$$

where $\delta_t(\varepsilon) = (2\pi)^{-1} \int_{-t}^{t} \mathrm{d}s\, e^{\mathrm{i}s\varepsilon}$, with $\lim_{t\to\infty} \delta_t(\varepsilon) = \delta(\varepsilon)$. We are careful in first executing integrals against the delta function implementing the condition $k[\theta_\mathrm{p}^\bullet, \theta_\mathrm{h}^\bullet] = 0$, before taking the limit towards the delta function implementing $\varepsilon[\theta_\mathrm{p}^\bullet, \theta_\mathrm{h}^\bullet] = 0$, as the space integral is performed before the time integral. We note that in the opposite order, the result would vanish because of (3.41). The conditions coming from both delta functions read

$$k(\theta_\mathrm{p}^1) + k(\theta_\mathrm{p}^2) = k(\theta_\mathrm{h}^1) + k(\theta_\mathrm{h}^2), \quad \varepsilon(\theta_\mathrm{p}^1) + \varepsilon(\theta_\mathrm{p}^2) = \varepsilon(\theta_\mathrm{h}^1) + \varepsilon(\theta_\mathrm{h}^2). \tag{4.5}$$

Interpreting incoming hole excitations as outgoing particle excitations, see Fig. 1, this has the form of a two-particle scattering process where both momentum and energy are conserved. In two dimensions, such scattering processes lead to $k(\theta_\mathrm{p}^1) = k(\theta_\mathrm{h}^1)$, $k(\theta_\mathrm{p}^2) = k(\theta_\mathrm{h}^2)$ or $k(\theta_\mathrm{p}^1) = k(\theta_\mathrm{h}^2)$, $k(\theta_\mathrm{p}^2) = k(\theta_\mathrm{h}^1)$, namely

$$\{\theta_\mathrm{p}^\bullet\} = \{\theta_\mathrm{h}^\bullet\}. \tag{4.6}$$

Since the product of form factors has poles at $\theta_\mathrm{p}^i = \theta_\mathrm{h}^j$ with $i, j \in \{1, 2\}$, the kinematic conditions (4.6) evaluate the form factors exactly at their poles, which appear as double poles in (4.4). However, the equality (4.6) is only *approached* as the limit $t \to \infty$ in (4.4) is taken. Indeed, first the measure of the multiple integral concentrates on the hypersurface $k[\theta_\mathrm{p}^\bullet, \theta_\mathrm{h}^\bullet] = 0$ in rapidity space, and the poles at (4.6) lie on a submanifold which has measure zero on this hypersurface; they are avoided by the principal-value prescription. Then, the limit $t \to \infty$ is taken, whereby $\delta_t(\varepsilon[\theta_\mathrm{p}^\bullet, \theta_\mathrm{h}^\bullet]) \to \delta(\varepsilon[\theta_\mathrm{p}^\bullet, \theta_\mathrm{h}^\bullet])$. This effectively reduces the hypersurface where the integration measure concentrates to a vanishing neighbourhood of (4.6). In this neighbourhood, the double-pole divergence coming form the form factors is simplified by the factor $(\varepsilon[\theta_\mathrm{p}^\bullet, \theta_\mathrm{h}^\bullet])^2$ from (3.41), and thus the result of the limit process is finite and non-zero.

At higher particle-hole numbers, an expression similar to (4.4) occurs, but the kinematic conditions $k[\theta_\mathrm{p}^\bullet, \theta_\mathrm{h}^\bullet] = 0$ and $\varepsilon[\theta_\mathrm{p}^\bullet, \theta_\mathrm{h}^\bullet] = 0$ have a continuum of solutions that do not impose any coincidence of momenta: the poles of the form factors lie on a submanifold of measure zero on the resulting integration hypersurface (and are avoided by the principal value prescription). On the other hand the factor $(\varepsilon[\theta_\mathrm{p}^\bullet, \theta_\mathrm{h}^\bullet])^2$ from (3.41) is zero everywhere on this hypersurface, and thus the contribution vanishes.

Hence, the $\mathcal{D}C$ operator is fully given by the two-particle-hole contribution,

$$(\mathcal{D}C)_{ij} = \int \mathrm{d}t \int \mathrm{d}x\, [\mathrm{jj}]_{ij}^{2\mathrm{ph}}(x, t) \tag{4.7}$$

and

$$\int \mathrm{d}t \int \mathrm{d}x\, [\mathrm{jj}]_{ij}^{n\mathrm{ph}}(x, t) = 0 \qquad n > 2. \tag{4.8}$$

The explicit calculation for the coefficients $(\mathcal{D}C)_{ij}$ is as follows. The two kinematic conditions $\delta(k[\theta_\mathrm{p}^\bullet, \theta_\mathrm{h}^\bullet])\delta(\varepsilon[\theta_\mathrm{p}^\bullet, \theta_\mathrm{h}^\bullet])$ only have solutions of the type (4.6), therefore we can evaluate the integrand with the form factors replaced by their leading behaviour near the poles. There are two separate contributions: $\theta_\mathrm{p}^i \sim \theta_\mathrm{h}^j$ with $i = j$ and with $i \neq j$. Both contributions give the same final result so we can simply consider the first case and multiply the final result by 2. We denote the new set of variables

$$\Delta\theta_1 = \theta_\mathrm{p}^1 - \theta_\mathrm{h}^1 \tag{4.9}$$

$$\Delta\theta_2 = \theta_\mathrm{p}^2 - \theta_\mathrm{h}^2 \tag{4.10}$$

and rename $\theta_{\mathrm{h}}^{1,2} \equiv \theta_{1,2}$, such that inside the integral we can expand the energy and the momentum in powers of $\Delta\theta_1$ and $\Delta\theta_2$

$$\varepsilon[\theta_{\mathrm{p}}^{\bullet}, \theta_{\mathrm{h}}^{\bullet}] = v^{\mathrm{eff}}(\theta_1)k'(\theta_1)\Delta\theta_1 + v^{\mathrm{eff}}(\theta_2)k'(\theta_2)\Delta\theta_2 + O((\Delta\theta_1)^2,(\Delta\theta_2)^2), \quad (4.11)$$

$$k[\theta_{\mathrm{p}}^{\bullet}, \theta_{\mathrm{h}}^{\bullet}] = k'(\theta_1)\Delta\theta_1 + k'(\theta_2)\Delta\theta_2 + O((\Delta\theta_1)^2,(\Delta\theta_2)^2), \quad (4.12)$$

where we used (3.35) and (3.20). We first integrate over $\Delta\theta_2$, which is fixed by the total momentum conservation to leading order as

$$\Delta\theta_2 = -\Delta\theta_1 \frac{k'(\theta_1)}{k'(\theta_2)}. \quad (4.13)$$

An extra factor

$$\frac{1}{|k'(\theta_2)|}$$

appears in the integrand after integrating against the delta function $\delta(k[\theta_{\mathrm{p}}^{\bullet}, \theta_{\mathrm{h}}^{\bullet}])$. The energy is now given, to leading order, by

$$\varepsilon[\theta_{\mathrm{p}}^{\bullet}, \theta_{\mathrm{h}}^{\bullet}] = (v^{\mathrm{eff}}(\theta_1) - v^{\mathrm{eff}}(\theta_2))\Delta\theta_1 k'(\theta_1). \quad (4.14)$$

Its square $\varepsilon[\theta_{\mathrm{p}}^{\bullet}, \theta_{\mathrm{h}}^{\bullet}]^2$, which appears in the product of form factors in (4.4) as per (3.41), is therefore proportional to $(\Delta\theta_1)^2$. The product of form factors in (4.4) also contains the product $f_i(\theta_{\mathrm{p}}^1, \theta_{\mathrm{h}}^1, \theta_{\mathrm{p}}^2, \theta_{\mathrm{h}}^2)f_j(\theta_{\mathrm{p}}^1, \theta_{\mathrm{h}}^1, \theta_{\mathrm{p}}^2, \theta_{\mathrm{h}}^2)$. The non-vanishing contributions come from the first two terms in (3.49), which are thus four terms in the product. Using (4.13), each of these is proportional to $(\Delta\theta_1)^{-2}$, which is therefore cancelled by the $(\Delta\theta_1)^2$ in $\varepsilon[\theta_{\mathrm{p}}^{\bullet}, \theta_{\mathrm{h}}^{\bullet}]^2$. The integration over $\Delta\theta_1$ against the delta function $\delta(\varepsilon[\theta_{\mathrm{p}}^{\bullet}, \theta_{\mathrm{h}}^{\bullet}])$ can now be taken, and it provides an additional factor

$$\frac{1}{|v^{\mathrm{eff}}(\theta_1) - v^{\mathrm{eff}}(\theta_2)|\,|k'(\theta_1)|}.$$

Putting all factors together, in particular using (3.49) and $2\pi\sigma_{1,2}\rho_{\mathrm{s}}(\theta_{1,2}) = k'(\theta_{1,2})$, we find the $\mathfrak{D}C$ matrix

$$(\mathfrak{D}C)_{ij} = \int \frac{\mathrm{d}\theta_1 \mathrm{d}\theta_2}{2}\rho_{\mathrm{p}}(\theta_1)(1 - n(\theta_1))\rho_{\mathrm{p}}(\theta_2)(1 - n(\theta_2))|v^{\mathrm{eff}}(\theta_1) - v^{\mathrm{eff}}(\theta_2)|$$

$$\times \left(\frac{T^{\mathrm{dr}}(\theta_2,\theta_1)h_i^{\mathrm{dr}}(\theta_2)}{\sigma_2\rho_{\mathrm{s}}(\theta_2)} - \frac{T^{\mathrm{dr}}(\theta_1,\theta_2)h_i^{\mathrm{dr}}(\theta_1)}{\sigma_1\rho_{\mathrm{s}}(\theta_1)}\right)\left(\frac{T^{\mathrm{dr}}(\theta_2,\theta_1)h_j^{\mathrm{dr}}(\theta_2)}{\sigma_2\rho_{\mathrm{s}}(\theta_2)} - \frac{T^{\mathrm{dr}}(\theta_1,\theta_2)h_j^{\mathrm{dr}}(\theta_1)}{\sigma_1\rho_{\mathrm{s}}(\theta_1)}\right), \quad (4.15)$$

which leads to the diffusion kernel reported in section 4.3. Naturally, as there is a single particle type, the signs $\sigma$ in the above expression cancel out. We recall the definition of the dressed scattering kernel,

$$T^{\mathrm{dr}} = (1 - Tn\sigma)^{-1}T. \quad (4.16)$$

Notice that in (4.15), we used the fact that after integration over $\Delta\theta_2$ the integrand becomes regular, making the result an ordinary integral, such that no regularisation scheme is needed.

## 4.2 Generalisation to arbitrary integrable models

Result (4.15) has been derived using the particle-hole form factor expansion, and thus it holds in the context of integrable models with fermionic excitations. It has also been derived explicitly with the understanding that there is a single quasiparticle type (and with a choice of positive parity). It is simple to propose its extension to integrable models with arbitrary number of quasiparticle types and arbitrary quasiparticle statistics, thus including classical integrable models. We provide arguments for the validity of this in quantum models with many string lengths in Appendix G, and check the proposal against the known classical hard rod result in Appendix H.

Recall from Remark 1 of subsection 3.1 that it is a simple matter in TBA and Euler-scale GHD to take into account many quasiparticle types: one then simply replaces each rapidity integral by the combination of a rapidity integral and a sum over quasiparticle types,

$$\int d\theta \mapsto \sum_a \int d\theta. \tag{4.17}$$

Further, the natural generalisation to models with *different statistics* is obtained by analogy with the formulae for correlation functions and Drude weights, making the replacement (see [101] for further remarks)

$$1 - n(\theta) \mapsto f(\theta) = -\frac{d^2 F(\epsilon)/d\epsilon^2}{dF(\epsilon)/d\epsilon}\bigg|_{\epsilon = \epsilon(\theta)}. \tag{4.18}$$

Therefore, the general formula is expected to be

$$
\begin{aligned}
(\mathfrak{D}C)_{ij} = \sum_{a,b} \int \frac{d\theta_1 d\theta_2}{2} & \rho_{\mathrm{p};a}(\theta_1) f_a(\theta_1) \rho_{\mathrm{p};b}(\theta_2) f_b(\theta_2) |v_a^{\mathrm{eff}}(\theta_1) - v_b^{\mathrm{eff}}(\theta_2)| \\
& \times \left( \frac{T_{b,a}^{\mathrm{dr}}(\theta_2, \theta_1) h_{i;b}^{\mathrm{dr}}(\theta_2)}{\sigma_b \rho_{\mathrm{s};b}(\theta_2)} - \frac{T_{a,b}^{\mathrm{dr}}(\theta_1, \theta_2) h_{i;a}^{\mathrm{dr}}(\theta_1)}{\sigma_a \rho_{\mathrm{s};a}(\theta_1)} \right) \\
& \times \left( \frac{T_{b,a}^{\mathrm{dr}}(\theta_2, \theta_1) h_{j;b}^{\mathrm{dr}}(\theta_2)}{\sigma_b \rho_{\mathrm{s};b}(\theta_2)} - \frac{T_{a,b}^{\mathrm{dr}}(\theta_1, \theta_2) h_{j;a}^{\mathrm{dr}}(\theta_1)}{\sigma_a \rho_{\mathrm{s};a}(\theta_1)} \right).
\end{aligned}
\tag{4.19}
$$

In this expression, recall that $\sigma_a$ is the sign of the parametrisation for quasiparticle type $a$,

$$\sigma_a = \mathrm{sgn}(k_a'(\theta)), \tag{4.20}$$

which enters, for instance, into the relation between $k_a'(\theta)$ and $\rho_{\mathrm{s};a}(\theta)$:

$$k_a'(\theta) = 2\pi \sigma_a \rho_{\mathrm{s};a}(\theta). \tag{4.21}$$

We recall that it is always possible to choose a parametrisation where $\sigma_a = 1$ for all $a$. It is a simple matter to see that formula (4.19) is reparametrisation invariant.

In many models, it is possible to choose a parametrisation such that $T$ is symmetric,

in which case $T^{\mathrm{dr}}$ also is. In this case, the formula simplifies to

$$(\mathfrak{D}C)_{ij} = \sum_{a,b} \int \frac{\mathrm{d}\theta_1 \mathrm{d}\theta_2}{2} \rho_{\mathrm{p};a}(\theta_1) f_a(\theta_1) \rho_{\mathrm{p};b}(\theta_2) f_b(\theta_2) |v_a^{\mathrm{eff}}(\theta_1) - v_b^{\mathrm{eff}}(\theta_2)|$$

$$\times \left[ T_{a,b}^{\mathrm{dr}}(\theta_1, \theta_2) \right]^2 \left( \frac{h_{i;b}^{\mathrm{dr}}(\theta_2)}{\sigma_b \rho_{\mathrm{s};b}(\theta_2)} - \frac{h_{i;a}^{\mathrm{dr}}(\theta_1)}{\sigma_a \rho_{\mathrm{s};a}(\theta_1)} \right) \left( \frac{h_{j;b}^{\mathrm{dr}}(\theta_2)}{\sigma_b \rho_{\mathrm{s};b}(\theta_2)} - \frac{h_{j;a}^{\mathrm{dr}}(\theta_1)}{\sigma_a \rho_{\mathrm{s};a}(\theta_1)} \right) \quad (4.22)$$

$$(T \text{ symmetric}).$$

As mentioned, this is valid in the XXZ chain, in the Lieb-Liniger model, in the hard rod gas, and in many other field theories, and nontrivial parities occur in the gapless regime of the XXZ chian at roots of unity [112] and in fermionic models like the Fermi-Hubbard chain [113].

## 4.3 Diffusion kernel and Markov property

From the above result, it is possible to extract the integral kernel $D_{a,b}(\theta, \alpha)$ for the diffusion matrix $\mathfrak{D}_i^{\ j}$. Using the dual integral-kernel form,

$$(\mathfrak{D}C)_{ij} = (h_i, \mathfrak{D}Ch_j) = \sum_{a,b} \int \mathrm{d}\theta \mathrm{d}\alpha \, h_{i;a}(\theta)(\mathfrak{D}C)_{a,b}(\theta, \alpha) h_{j;b}(\alpha), \quad (4.23)$$

and writing as usual $n$, $f$, $\rho_{\mathrm{s}}$, $\sigma$ and $1$ as diagonal integral kernels, we define the integral kernel $\widetilde{D}$ via

$$\mathfrak{D}C = (1 - \sigma n T)^{-1} \rho_{\mathrm{s}} \sigma \widetilde{\mathfrak{D}} \sigma n f (1 - T^{\mathrm{T}} n \sigma)^{-1}. \quad (4.24)$$

Since from (3.37) we have

$$C = (1 - \sigma n T)^{-1} \rho_{\mathrm{p}} f (1 - T^{\mathrm{T}} n \sigma)^{-1}, \quad (4.25)$$

we find

$$\mathfrak{D} = (1 - \sigma n T)^{-1} \rho_{\mathrm{s}} \sigma \widetilde{\mathfrak{D}} \sigma \rho_{\mathrm{s}}^{-1} (1 - \sigma n T). \quad (4.26)$$

Using (4.24) with (4.19), we can extract the diffusion kernel, which can be written in the form:

$$\frac{1}{2} \widetilde{\mathfrak{D}}_{a,b}(\theta, \alpha) = \delta_{a,b} \delta(\theta - \alpha) \widetilde{w}_a(\theta) - \widetilde{W}_{a,b}(\theta, \alpha). \quad (4.27)$$

The off-diagonal elements of this kernel can be interpreted as the effects of interparticle scatterings with different velocities,

$$\widetilde{W}_{a,b}(\theta, \alpha) = \frac{1}{2} \rho_{\mathrm{p};a}(\theta) f_a(\theta) \frac{T_{a,b}^{\mathrm{dr}}(\theta, \alpha) T_{b,a}^{\mathrm{dr}}(\alpha, \theta)}{\rho_{\mathrm{s};a}(\theta)^2} |v_a^{\mathrm{eff}}(\theta) - v_b^{\mathrm{eff}}(\alpha)|, \quad (4.28)$$

while the diagonal part, $\widetilde{w}(\theta)$ can be seen as the effective variance for the quasiparticle fluctuations with rapidity $\theta$ inside the local stationary state, caused by the random scattering processes (see also [66])

$$\widetilde{w}_a(\theta) = \frac{1}{2} \sum_b \int \mathrm{d}\alpha \, \rho_{\mathrm{p};b}(\alpha)(1 - n_b(\alpha)) \left( \frac{T_{a,b}^{\mathrm{dr}}(\theta, \alpha)}{\rho_{\mathrm{s};a}(\theta)} \right)^2 |v_a^{\mathrm{eff}}(\theta) - v_b^{\mathrm{eff}}(\alpha)|, \quad (4.29)$$

where the ratio $T^{\mathrm{dr}}_{a,b}(\theta,\alpha)/\rho_{\mathrm{s};a}(\theta)$ can be thought as the average displacement of the quasiparticle of type $a$ due to multiple scattering processes with all the other quasiparticles present in the local stationary state. Clearly, under reparametrisation, the kernels $\widetilde{\mathfrak{D}}(\theta,\alpha)$ and $\widetilde{W}(\theta,\alpha)$ are both scalars in $\theta$ and vector fields in $\alpha$, the function $\widetilde{w}(\theta)$ is a scalar, and the kernel $\mathfrak{D}(\theta,\alpha)$ is a vector field in $\theta$ and a scalar in $\alpha$.

If $T$ is symmetric, then we can write

$$\widetilde{w}_a(\theta) = \rho_{\mathrm{s};a}(\theta)^{-2} \sum_b \int \mathrm{d}\alpha \, \widetilde{W}_{b,a}(\alpha,\theta)\rho_{\mathrm{s};b}(\alpha)^2. \tag{4.30}$$

In this case, the operator $\mathfrak{D}$ is a Markov operator with respect to the measure $\mathrm{d}p_a(\theta)$, which can be interpreted, much like in the hard rod case [64, 72, 80, 142], as describing the random exchange of velocities among the quasiparticles under collisions. Indeed, in this case

$$\sum_a \int \mathrm{d}\theta \, p'_a(\theta)\mathfrak{D}_{a,b}(\theta,\alpha) = 0, \tag{4.31}$$

which can be seen either by combining (4.30) with (4.27), or directly by choosing $h_{i;a}(\theta) = p'_a(\theta)$ in (4.22): with the latter, we immediately find (for any parity) that $(\mathfrak{D}C)_{ij} = 0$ using $(p')^{\mathrm{dr}}_a(\theta) = k'_a(\theta) = 2\pi\sigma_a\rho_{\mathrm{s};a}(\theta)$, and since this holds for all $h_{j;b}(\theta)$, it implies (4.31). The Markov property (4.31) implies the lack of diffusion for the conserved quantity corresponding to this choice of $h_a(\theta)$.

As mentioned in section 2.5, this sum rule – the lack of diffusion – follows from Galilean or relativistic invariance as then the current of mass (energy) is the momentum density, and agrees with the general argument, explained in Remark 4 of section 3.1, according to which for any model where there is a parametrisation making $T$ symmetric, the diffusion associated to the charge corresponding to $p'(\theta)$ must be zero. In particular, it is the case in the XXZ spin chain, where the energy current is a conserved density, implying the absence of energy diffusion in the chain.

# 5 GHD Navier-Stokes equation, entropy increase, linear regime

In this section, we derive consequences of the form of the diffusion kernel found in the previous section. We write the Navier-Stokes equation corresponding to it in various forms, prove that this leads to entropy production, analyse its linear regime, and apply these to the Lieb-Liniger model as an explicit example. For lightness of notation, we restrict ourselves to the case of a single quasiparticle type and a symmetric differential scattering phase (3.5), but all results can be generalise by following the principles discussed in Remarks 1 and 2 of subsection 3.1.

## 5.1 Navier-Stokes GHD equation for the quasiparticle densities and occupation numbers

The expression (3.28) for the currents in terms of the root densities, allows to write directly the Navier-Stokes equation for the density of quasi-particles

$$\partial_t \rho_{\mathrm{p}} + \partial_x(v^{\mathrm{eff}}\rho_{\mathrm{p}}) = \partial_x \mathcal{N}, \qquad \mathcal{N} = \frac{1}{2}\mathfrak{D}\partial_x\rho_{\mathrm{p}}, \tag{5.1}$$

where as usual we see $\partial_x \rho_{\rm p}$ as a vector in the space of spectral functions, and the diffusion operator $\mathfrak{D}$ of eq. (4.26) as an integral operator acting on this space. We shall write this equation for the occupation numbers $n(\theta) = \rho_{\rm p}(\theta)/\rho_{\rm s}(\theta)$. We will show that this can be expressed in terms of the diffusion kernel introduced in eq. (4.27), as

$$\partial_t n + v^{\rm eff}\partial_x n = \frac{1}{\rho_{\rm s}}(1-nT)\partial_x \mathcal{N} = \frac{1}{2\rho_{\rm s}}(1-nT)\partial_x\left((1-nT)^{-1}\rho_{\rm s}\widetilde{\mathfrak{D}}\partial_x n\right). \qquad (5.2)$$

Since the full solution to this equation is technically very involved it is useful to consider the so-called linear regime, where we can neglect second and higher powers of $(\partial_x n)$ (for instance, if $n(\theta; x, t)$, in addition to being smooth, stays close to some equilibrium value $n_{\rm eq}(\theta)$). In this limit equation (5.2) reduces to the equation

$$\partial_t n + v^{\rm eff}\partial_x n = \frac{1}{2}\widetilde{\mathfrak{D}}\partial_x^2 n + O((\partial_x n)^2) \qquad (5.3)$$

that simply account for diffusion spreading on top of the ballistic propagation of the quasiparticles with their effective velocities.

In order to derive equation (5.2), first, we show that

$$(1-nT)\mathcal{N} = \frac{1}{2}\rho_{\rm s}\widetilde{\mathfrak{D}}\partial_x n. \qquad (5.4)$$

Using the Bethe equations $T\rho_{\rm p} = \rho_{\rm s} - p'/2\pi$ we find

$$(1-nT)\partial_x\rho_{\rm p} = \partial_x\rho_{\rm p} - n\partial_x(T\rho_{\rm p}) = \rho_{\rm p}\partial_x \log n, \qquad (5.5)$$

and combined with

$$(1-nT)\mathcal{N} = \frac{1}{2}\rho_{\rm s}\widetilde{\mathfrak{D}}\rho_{\rm s}^{-1}(1-nT)\partial_x\rho_{\rm p} \qquad (5.6)$$

this shows (5.4). We are then in position to arrive at the following equation for the distribution $n$:

$$\partial_t n + v^{\rm eff}\partial_x n = \frac{1}{\rho_{\rm s}}(1-nT)\partial_x \mathcal{N}. \qquad (5.7)$$

To prove this relation we recall that, for any parameter $\alpha$ on which $n$ depends, we have the following identity

$$\partial_\alpha\left(n(1-Tn)^{-1}\right) = (1-nT)^{-1}\partial_\alpha n(1-Tn)^{-1}. \qquad (5.8)$$

Therefore we find

$$2\pi\partial_t\rho_{\rm p} = \partial_t\left(n(1-Tn)^{-1}p'\right) = (1-nT)^{-1}\partial_t n(1-Tn)^{-1}p', \qquad (5.9)$$

and similarly for $2\pi\partial_x\left(v^{\rm eff}\rho_{\rm p}\right) = \partial_x\left(n(1-Tn)^{-1}E'\right)$, so we have from (5.1)

$$(1-nT)^{-1}\partial_t n(p')^{\rm dr} + (1-nT)^{-1}\partial_x n(E')^{\rm dr} = 2\pi\partial_x\mathcal{N}, \qquad (5.10)$$

and therefore

$$\partial_t n + v^{\rm eff}\partial_x n = \frac{2\pi}{(p')^{\rm dr}}(1-nT)\partial_x\mathcal{N}, \qquad (5.11)$$

which shows (5.7) using $(p')^{\rm dr} = 2\pi\rho_{\rm s}$.

## 5.2 Entropy increase due to diffusion

It is a well established notion in classical hydrodynamic theory that the diffusion terms in the Navier-Stokes equation breaks time-reversal symmetry. This is most clearly shown by constructing a function of the hydrodynamic state, expressed as an integral over local states, which is strictly increasing with time. This function then has the usual interpretation as the total entropy of all fluid cells, expressed as an integral over an entropy density. The fact that the total entropy of fluid cells increase with time is of course not in contradiction with the underline unitary (deterministic) evolution of the system. Indeed, the total entropy of the system, say the von Neumann entropy of the whole system's density matrix, is invariant under time evolution; however by separation of scales, it should be, intuitively, composed of the total entropy of local fluid cells plus the entropy stored in large-scale structures. Because of diffusion, entropy stored in large-scale structures passes to microscopic scales (large-scale structures are smoothed out, reducing the large-scale configuration space). Thus, each fluid cell sees its entropy increase. See also the related recent discussions on entropy and related concepts in isolated systems [143–145].

In the specific case of GHD, each fluid cell is specified by the densities of quasiparticles $\rho_p(\theta; x, t)$. Following (3.2), the entropy of a fluid cell at position $x, t$ is given by the formula

$$s(x, t) = \int d\theta\, \rho_s(\theta; x, t) g(n(\theta; x, t)). \tag{5.12}$$

The function $g(n)$ depends on the statistics of the quasiparticles, and is given by (3.10). The solution of this equation is specified up to terms of zeroth and first powers of $n$, which do not affect total entropy variations. One can show that in the Fermionic case the Yang-Yang [115] formula is recovered,

$$s = -\int d\theta\, \rho_s (n \log n + (1-n) \log(1-n)), \tag{5.13}$$

in the bosonic case that of [146] is obtained

$$s = -\int d\theta\, \rho_s (n \log n - (1+n) \log(1+n)), \tag{5.14}$$

in the case of classical particles one finds the usual classical entropy,

$$s = -\int d\theta\, \rho_s n \log n, \tag{5.15}$$

and the correct classical field entropy is found in the case of radiative modes,

$$s = \int d\theta\, \rho_s \log n. \tag{5.16}$$

If the fluid cell's state is described by a GGE, then expression (5.12) is exactly the density per unit length of the GGE's von Neumann entropy [147, 148]. As discussed in subsection 3.2, at the Euler scale, local states are described by GGEs, and thus in this case $s(x, t)$ is the von Neumann entropy, per unit length, of the reduced density matrix (or marginal distribution) of the fluid cell at $x, t$. However, beyond the Euler scale, the GGE description of local states is not valid any more, as the GGE form of averages of

local observables if generically modified by derivative (diffusive) terms. In this case it is not clear if $s(x, t)$ is equal to the von Neumann entropy of the reduced density matrix of fluid cells. Yet, we now show that the total entropy $S(t) = \int \mathrm{d}x\, s(x, t)$ associated to this entropy density is strictly increasing with time under the Navier-Stokes equation (5.1). In this sense, $s(x, t)$ can be interpreted as an entropy density of the local fluid cell.

At Euler scale the local entropy $s(x, t)$ satisfies a continuity equation [149, 150], namely

$$\partial_t s + \partial_x j_s^{(1)} = 0, \tag{5.17}$$

with the Euler-scale entropy current given by

$$j_s^{(1)}(x, t) = \int \mathrm{d}\theta\, v^{\mathrm{eff}} \rho_s g(n). \tag{5.18}$$

This means that, in systems of finite extent (where the entropy density vanishes beyond this extent), the total entropy $S(t) = \int \mathrm{d}x\, s(x, t)$ is preserved, $\mathrm{d}S(t)/\mathrm{d}t = 0$. We now introduce the Navier-Stokes terms, and we show that the total entropy $S(t)$ *increases with time*.

For simplicity we introduce $G(n) = g(n)/n$ and write

$$s = \int \mathrm{d}\theta\, \rho_{\mathrm{p}} G(n). \tag{5.19}$$

Extracting the terms involving $\mathcal{N}$ in equations (5.2) and (5.1), and using (5.4), we find

$$
\begin{aligned}
&\partial_t s + \partial_x j_s^{(1)} \\
&= \int \mathrm{d}\theta \left[ G(n)\partial_x \mathcal{N} + \rho_{\mathrm{p}} G'(n)\left( \frac{1}{\rho_{\mathrm{s}}}(1 - nT)\partial_x \mathcal{N} \right) \right] \\
&= -\partial_x j_s^{(2)} - \int \mathrm{d}\theta\, (2G'(n) + nG''(n))\partial_x n(1 - nT)\mathcal{N} \\
&= -\partial_x j_s^{(2)} - \frac{1}{2} \int \mathrm{d}\theta\, (2G'(n) + nG''(n))\partial_x n\, (\rho_{\mathrm{s}}\widetilde{\mathfrak{D}}\rho_{\mathrm{s}}^{-1})\rho_{\mathrm{s}}\, \partial_x n,
\end{aligned}
\tag{5.20}
$$

where the correction to the entropy current given by diffusive terms is

$$j_s^{(2)} = - \int \mathrm{d}\theta \left[ \mathcal{N}G(n) + nG'(n)(1 - nT)\mathcal{N} \right]. \tag{5.21}$$

Now we note that $2G'(n) + nG''(n) = g''(n) = -1/(nf)$ (the last equation obtained using (3.11) and (4.18)), and we find

$$\partial_t s + \partial_x (j_s^{(1)} + j_s^{(2)}) = \frac{1}{2} \int \mathrm{d}\theta \int \mathrm{d}\alpha\, \frac{\partial_x n(\theta)}{n(\theta)f(\theta)} \rho_{\mathrm{s}}(\theta)\widetilde{\mathfrak{D}}(\theta, \alpha)\partial_x n(\alpha). \tag{5.22}$$

Therefore, using (4.24), the equation for the hydrodynamic evolution of the entropy density can be more compactly written as

$$\partial_t s + \partial_x \left( j_s^{(1)} + j_s^{(2)} \right) = \frac{1}{2}(\Sigma, \mathfrak{D}C\Sigma), \tag{5.23}$$

with the spectral functio $\Sigma$ defined as $(1 - Tn)^{-1}\Sigma = \partial_x n/(fn)$. As a consequence, the total entropy $S(t)$ is only conserved up to terms of order $(\partial_x n(x, t))^2$, and

$$\frac{\mathrm{d}S(t)}{\mathrm{d}t} = \frac{1}{2} \int \mathrm{d}x\, (\Sigma, \mathfrak{D}C\Sigma)(x, t) \geq 0, \tag{5.24}$$

where the inequality follows from the positivity of the operator $\mathfrak{D}C$, which is clear from the expression (4.19). It is a simple matter to see that, generically, the inequality is actually strict. Further, we note that the entropy current is modified by terms of order $\partial_x n(x, t)$ (an effect also known in the hard rod gas [64, 72]), and given by

$$
\begin{aligned}
j_s^{(1)} + j_s^{(2)} = \int \mathrm{d}\theta \Big( &\rho_s(\theta) g(n(\theta)) v^{\mathrm{eff}}(\theta) \\
&- \mathcal{N}(\theta) G(n(\theta)) - \int \mathrm{d}\alpha\, n(\theta) G'(n(\theta))(\delta(\theta - \alpha) - n(\theta) T(\theta, \alpha)) \mathcal{N}(\alpha) \Big).
\end{aligned}
\tag{5.25}
$$

### 5.3 Solution of the partitioning protocol in the linear regime

One of the main application of the hydrodynamic equation (5.2) is the dynamics given by a bi-partite initial state, namely when the initial state is chosen to be the tensor product of two macroscopically different state. Such a non-equilibrium dynamics has been studied extensively in the past years. The solution at the Euler scale in integrable models is a continuum of contact singularities, one for each value of $\theta$ [49, 151]. Such singularities are a feature of the Euler scale, and are smoothed out at shorter space-time scales by diffusive spreading effects, which, upon integration over $\theta$, give rise to $1/\sqrt{t}$ corrections to local observables.

We here consider the equation (5.3) for the occupation numbers in the linear regime

$$
\partial_t n + v^{\mathrm{eff}} \partial_x n = \frac{1}{2} \widetilde{\mathfrak{D}} \partial_x^2 n + O((\partial_x n)^2).
\tag{5.26}
$$

We aim to solve this equation with step initial conditions given by

$$
n(\theta; x, 0) = n_L(\theta) \Theta(-x) + n_R(\theta) \Theta(x),
\tag{5.27}
$$

with $n_L \sim n_R$. In the following we shall show that the solution of this equation up to corrections of order $t^{-1}$ at large times, is given by

$$
\begin{aligned}
n(\theta; x, t) = n_L(\theta) + (n_R(\theta) - n_L(\theta)) \frac{1 - \mathrm{erf}\left(\sqrt{\frac{1}{4t\widetilde{w}(\theta)}}(t v^{\mathrm{eff}}(\theta) - x)\right)}{2} \\
- \int \mathrm{d}\theta' \Delta n(\theta, \theta')(n_R(\theta') - n_L(\theta')) + O(t^{-1}),
\end{aligned}
\tag{5.28}
$$

with $\mathrm{erf}(u)$ the error function and

$$
\Delta n(\theta, \theta') = \left( \frac{e^{-\frac{(x - t v^{\mathrm{eff}}(\theta))^2}{4\widetilde{w}(\theta)t}}}{\sqrt{4\pi \widetilde{w}(\theta)t}} - \frac{e^{-\frac{(x - t v^{\mathrm{eff}}(\theta'))^2}{4\widetilde{w}(\theta')t}}}{\sqrt{4\pi \widetilde{w}(\theta')t}} \right) \frac{\widetilde{W}(\theta, \theta')}{v^{\mathrm{eff}}(\theta') - v^{\mathrm{eff}}(\theta)}.
\tag{5.29}
$$

It is interesting to compare this solution with the one obtained in the standard ballistic GHD case (obtained in the limit of zero diffusion $\widetilde{w} \to 0$ and $\widetilde{W} \to 0$), which reads

$$
n(\theta; x, t) = n_L(\theta) + (n_R(\theta) - n_L(\theta)) \Theta\left(t v^{\mathrm{eff}}(\theta) - x\right),
\tag{5.30}
$$

with the Heaviside theta function $\Theta(x)$. At each $x, t$ there is a contact singularity at $\theta^*$, fixed by the condition $v^{\mathrm{eff}}(\theta^*) = x/t$. This is instead smoothed out by the diffusive terms

in (5.28). This on the other hand is not the only effect of diffusion: the interparticle scatterings also lead to non-local changes in rapidities between $n(\theta)$ and $n(\theta')$. This leads to rearrangements of rapidities among particles due to the off-diagonal elements $\widetilde{W}(\theta, \theta')$ of the diffusion kernel, that give the second term in (5.28).

In order to arrive to (5.28) we use that we can solve the equation by Fourier transform,

$$n(\theta; x, t) = n_L(\theta) + \int \frac{dk}{2\pi} e^{ikx} \left( \exp\left[ -(ikv^{\text{eff}} + \frac{1}{2}k^2\widetilde{\mathfrak{D}})t \right] n_k \right)(\theta). \tag{5.31}$$

By neglecting the diffusion terms, there would be only a diagonal matrix $-ikv^{\text{eff}}$ in the exponential. The set of these matrices for all $k$ all commute with each other, so one could diagonalise them simultaneously. A possible basis of eigenvectors is given by the vectors $u(\theta'; \theta) = \delta(\theta - \theta')$ with eigenvalues $kv^{\text{eff}}(\theta')$ and one decomposes as $n_k(\theta) = \int d\theta' C_k(\theta')u(\theta'; \theta)$ thus getting $\int \frac{dk}{2\pi} d\theta' e^{ikx - ikv^{\text{eff}}(\theta')t} C_k(\theta')$. The initial condition then fixes the final solution as

$$n_k(\theta) = \frac{n_R(\theta) - n_L(\theta)}{ik}. \tag{5.32}$$

Including the diffusion kernels, one is left, in the exponential, with the matrices

$$iv^{\text{eff}}(\theta)\delta(\theta - \alpha) + k\widetilde{\mathfrak{D}}(\theta, \alpha)/2 \tag{5.33}$$

for all $k$'s. These matrices do not commute with each other for different $k$'s, therefore we cannot find a set of eigenvectors that simultaneously diagonalise all of them. Instead, we find, for each $k$, a set of $k$−dependent eigenvectors by first order perturbation, keeping in mind that the propagator (5.33) is valid up to terms of order $k^3$, namely up to terms of order $t^{-1}$ in the solution of equation (5.3). We then proceed to express each $n_k(\theta)$ as a linear combination on all the $k$-set: we search for vectors satisfying

$$\begin{aligned} &\left( (iv^{\text{eff}} + k\widetilde{w} - k\widetilde{W})(u(\theta') + k\Delta u(\theta')) \right)(\theta) \\ &= (\lambda(\theta') + k\Delta\lambda(\theta'))(u(\theta'; \theta) + k\Delta u(\theta'; \theta)) + O(k^2), \end{aligned} \tag{5.34}$$

where $\lambda = iv^{\text{eff}}(\theta') + k\widetilde{w}(\theta')$ and $u(\theta'; \theta) = \delta(\theta' - \theta)$. We find

$$\begin{aligned} &-k\widetilde{W}(\theta, \theta') + k(iv^{\text{eff}}(\theta) + k\widetilde{w}(\theta))\Delta u(\theta'; \theta) \\ &= k\Delta\lambda(\theta')\delta(\theta' - \theta) + k(iv^{\text{eff}}(\theta') + k\widetilde{w}(\theta'))\Delta u(\theta'; \theta). \end{aligned} \tag{5.35}$$

This is solved as

$$\Delta u(\theta'; \theta) = \frac{-\widetilde{W}(\theta, \theta') - \Delta\lambda(\theta')\delta(\theta - \theta')}{i(v^{\text{eff}}(\theta') - v^{\text{eff}}(\theta)) + k(\widetilde{w}(\theta') - \widetilde{w}(\theta))}. \tag{5.36}$$

The condition of well-definiteness of the vector fixes the eigenvalue,

$$\Delta\lambda = -\widetilde{W}(\theta, \theta)d\theta, \tag{5.37}$$

which guarantees that the numerator is zero at the pole $\theta = \theta'$ of the denominator. This means that the shift of eigenvalue is infinitesimal, and that the resulting shift of vector $\Delta u(\theta'; \theta)$ is to be interpreted, under integration over $\theta'$, as a principal value. But since

$\widetilde{W}(\theta,\theta) = 0$ there is no need for principal value integration, and $\Delta\lambda = 0$. Therefore we decompose as

$$n_k(\theta) = C_k(\theta) - \int \mathrm{d}\theta' \, C_k(\theta') \frac{k\widetilde{W}(\theta,\theta')}{\mathrm{i}(v^{\mathrm{eff}}(\theta') - v^{\mathrm{eff}}(\theta)) + k(\widetilde{w}(\theta') - \widetilde{w}(\theta))} + O(k^2). \quad (5.38)$$

The initial condition now fixes the coefficients as

$$C_k(\theta) = \frac{n_R(\theta) - n_L(\theta)}{\mathrm{i}k + 0^+} - \mathrm{i} \int \mathrm{d}\theta' \frac{(n_R(\theta') - n_L(\theta'))\widetilde{W}(\theta,\theta')}{\mathrm{i}(v^{\mathrm{eff}}(\theta') - v^{\mathrm{eff}}(\theta)) + k(\widetilde{w}(\theta') - \widetilde{w}(\theta))} + O(k). \quad (5.39)$$

The full solution is then given by

$$n(\theta; x, t) = n_L(\theta) + \int \frac{\mathrm{d}k}{2\pi} \mathrm{d}\theta' \, e^{\mathrm{i}kx - \mathrm{i}k t v^{\mathrm{eff}}(\theta') - k^2 t \widetilde{w}(\theta')} C(\theta, \theta'; k),$$

with

$$C(\theta, \theta'; k) = C_k(\theta') \left[ \delta(\theta - \theta') - \frac{k\widetilde{W}(\theta,\theta')}{\mathrm{i}(v^{\mathrm{eff}}(\theta') - v^{\mathrm{eff}}(\theta)) + k(\widetilde{w}(\theta') - \widetilde{w}(\theta))} \right]. \quad (5.40)$$

By substituting the functions $C_k$ and keeping only terms up to the order calculated (because of the initial condition giving a $1/k$, this means neglecting $O(k^2)$ terms), one finds

$$n(\theta; x, t) = n_L(\theta) + \int \frac{\mathrm{d}k}{2\pi} e^{\mathrm{i}kx} \left[ e^{-\mathrm{i}k t v^{\mathrm{eff}}(\theta) - k^2 t(\theta)\widetilde{w}(\theta)} \frac{n_R(\theta) - n_L(\theta)}{\mathrm{i}k + 0^+} \right.$$
$$\left. - \int \mathrm{d}\theta' \frac{\left( e^{-\mathrm{i}k t v^{\mathrm{eff}}(\theta) - k^2 t \widetilde{w}(\theta)} - e^{-\mathrm{i}k t v^{\mathrm{eff}}(\theta') - k^2 t \widetilde{w}(\theta')} \right)(n_R(\theta') - n_L(\theta'))\widetilde{W}(\theta,\theta')}{v^{\mathrm{eff}}(\theta') - v^{\mathrm{eff}}(\theta)} \right].$$
$$(5.41)$$

The integration over $k$ can now be trivially performed in terms of the error function $\mathrm{erf}(u)$, giving

$$\int \frac{\mathrm{d}k}{2\pi} e^{\mathrm{i}kx - \mathrm{i}k t v^{\mathrm{eff}}(\theta) - k^2 t \widetilde{w}(\theta)} \frac{1}{\mathrm{i}k + 0^+} = \frac{1 - \mathrm{erf}\left( \sqrt{\frac{1}{4t\widetilde{w}(\theta)}}(t v^{\mathrm{eff}}(\theta) - x) \right)}{2}, \quad (5.42)$$

which finally leads to equation (5.28).

## 5.4 Diffusion in the Lieb-Liniger Bose gas

The Lieb-Liniger Bose gas is a model that describes bosons on a line interacting point-wise (here with mass $m = 1/2$) [152]

$$H = \sum_{j=1}^{N} \partial_{x_j}^2 + 2c \sum_{i<j=1}^{N} \delta(x_i - x_j), \quad (5.43)$$

on a line of length $L$ and where the coupling $c$ can be taken both positive and negative (attractive Bose gas). Below we consider the repulsive regime $c > 0$, where the TBA formulation is simpler due to the absence of bound states. The model is Galilean invariant and as a consequence the momentum is exactly given by the rapidity $p(\theta) = \theta$ with

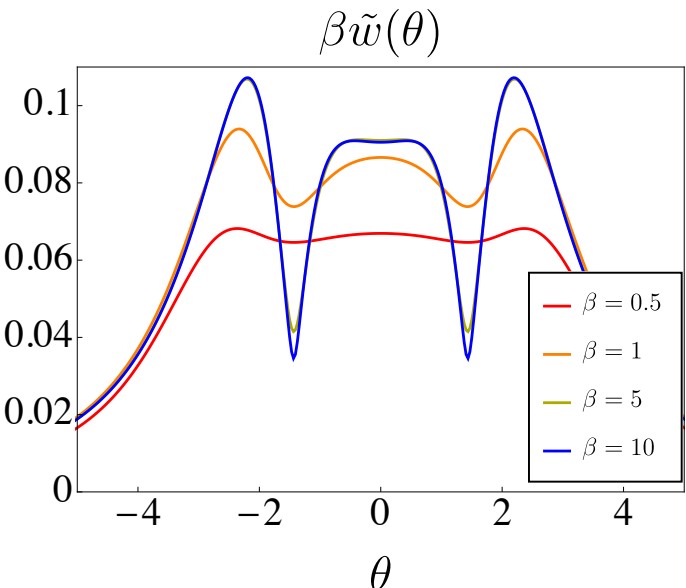

Figure 2: Plot of the quasiparticle variance (5.47) multiplied by the inverse temperature $\beta = 1/T$ for a thermal homogeneous Lieb-Liniger gas with coupling $c = 1$ and density $n_0 = 1$. At small temperature it converges to an asymptotic function which becomes equal to zero at the two Fermi points $\theta = \pm\theta_F$ in the limit $\beta \to \infty$.

$\theta \in (-\infty, +\infty)$ (with this choice of parametrisation, the rapidity is related to the velocity as $v(\theta) = 2\theta$). Transport of charges is always ballistic, with its Drude weights written in [141]. However there are diffusive corrections on top of it, except for the local density $n_0(x, t) = \int_{-\infty}^{+\infty} d\theta \rho_p(\theta; x, t)$ whose diffusion matrix element is zero, due to (4.31). It is interesting to compute the diffusion operator in the strong coupling limit $c \to \infty$. In this limit the scattering kernel becomes a constant function

$$T(\theta, \alpha) = \frac{1}{\pi} \frac{c}{(\theta - \alpha)^2 + c^2} = \frac{1}{\pi c} + O(c^{-2}), \tag{5.44}$$

and moreover some different analytical results can be obtained [153, 154] for the ground state proprieties of the gas. The diffusion kernel in this limits then becomes

$$\mathfrak{D}(\theta, \alpha) = \left(\frac{2}{c}\right)^2 \left(1 + \frac{2n_0}{c}\right)^{-2} \left[\left(\int d\kappa \rho_p(\kappa)(1 - n(\kappa))|v^{\text{eff}}(\theta) - v^{\text{eff}}(\kappa)|\right) \delta(\theta - \alpha)\right.$$
$$\left. - \rho_p(\theta)(1 - n(\theta))|v^{\text{eff}}(\theta) - v^{\text{eff}}(\alpha)|\right], \tag{5.45}$$

with the effective velocity given in this limit by

$$v^{\text{eff}}(\theta) = v(\theta) + \frac{2}{c} \int \frac{d\alpha}{2\pi} v(\alpha) n(\alpha) + O(c^{-2}). \tag{5.46}$$

It is interesting to notice that diffusion only take place at order $c^{-2}$, as at order $c^{-1}$ the quasiparticles are simply non-interacting dressed free fermionic particles [155]. Notice

that the diffusion operator (5.45) becomes almost equivalent in this limit to the one of a hard rod gas (H.7) up to the statistical factor $f = 1 - n$ in the case of the Bose gas. The dressed velocity (5.46) corresponds exactly to the one of a hard-rod gas with rod length $a = -2/c$. Therefore, as was noticed before [156, 157], the Lieb-Liniger gas at large coupling positive $c$ is completely analogous to a hard-rod gas of particles at the Euler scales (albeit with negative rod lengths), but due to the explicit dependence of diffusion from the statistics of the quasiparticles, the equivalence does not hold at diffusive scales.

The quasiparticle variance (4.29)

$$\widetilde{w}(\theta) = \frac{1}{2} \int d\alpha \, \rho_p(\alpha)(1 - n(\alpha)) \left( \frac{T^{dr}(\theta, \alpha)}{\rho_s(\theta)} \right)^2 |v^{eff}(\theta) - v^{eff}(\alpha)| \tag{5.47}$$

can be shown to be proportional the amplitude of the space fluctuations of the quasiparticles around their average position, see [66]. In Fig. 2 we plot this quantity for a thermal Lieb-Liniger gas at $c = 1$, particle density $n_0 = 1$ and for different temperatures.

## 6  Gapped XXZ chain and spin diffusion at half-filling

In this section we focus on the XXZ spin-1/2 chain, defined by the Hamiltonian

$$H = \sum_{j=1}^{L} \left( \frac{1}{2}(S_j^+ S_{j+1}^- + S_j^- S_{j+1}^+) + \Delta S_j^z S_{j+1}^z \right), \tag{6.1}$$

with the spin 1/2 operators $S_j^{\pm,z} \equiv S^{\pm,z}(j)$ at site $j$. We describe the physics of the spin transport of the model at equilibrium, for example the dynamics observed when two semi-infinite chains at thermal equilibrium with slightly different magnetizations are joined together [91, 98, 142, 158, 159] or when the initial state is an equilibrium state with a small local perturbation in the magnetization density, as done for example with tDMRG numerical simulations in [91]. The observed spin dynamics can be very different depending on the parameters of the system, namely the value of the anisoptropy $\Delta$ and the filling, i.e. the total magnetization of the state

$$S_0^z = \lim_{L \to \infty} L^{-1} \sum_{j=1}^{L} \langle S^z(j) \rangle. \tag{6.2}$$

It is useful therefore to make here a short review of the different regimes for spin transport in the XXZ chain *at thermal equilibrium,*

1. $|\Delta| < 1$, $S_0^z \in (-1/2, 1/2)$ and $|\Delta| \geq 1$, $S_0^z \neq 0$: spin transport is ballistic with diffusive corrections. The presence of a ballistic spin current at $S_0^z = 0$ is due to the presence of extra conserved quantities in the regime $|\Delta| < 1$ which are odd under spin flip on the whole chain [60, 97, 160, 161]. At finite $S_0^z$ all the other spin even (and parity odd) conserved quantities contribute to the spin current [93, 162], since the state is not spin-flip invariant. The Drude weights for the ballistic current were obtained in [39, 94, 158, 163].

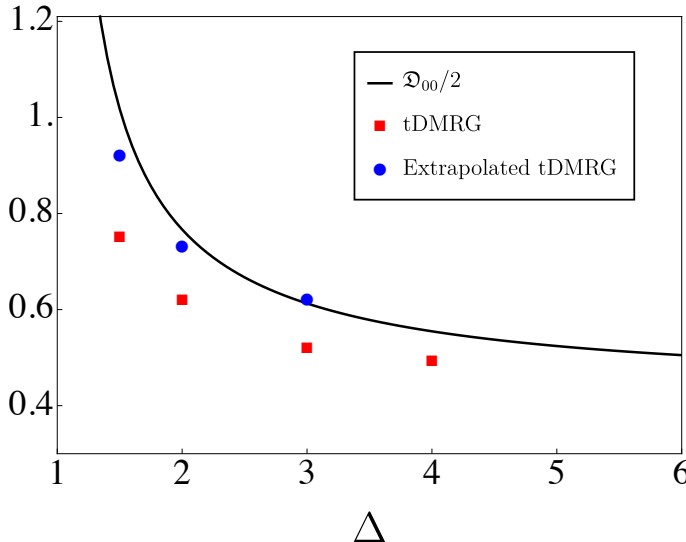

Figure 3: Plot of the spin diffusion constant $\mathfrak{D}_{00} = (\mathfrak{D})_{S^z, S^z}$ for a gapped XXZ chain at infinite temperature $\beta = 0$ and half-filling as given by (6.6) (black line) as function of $\Delta$. Red squared are data obtained by integrating the time-dependent current-current correlator obtained from tDMRG simulations with a finite maximal time from [35, 91, 92] and blue dots are the result of their extrapolation at large times, see Appendix I. The large $\Delta$ limit is given by $\frac{1}{2} \lim_{\Delta \to \infty} \mathfrak{D}_{00} \simeq 0.424$ and close to $\Delta = 1$ it diverges as $\sim (\Delta - 1)^{-1/2}$, signalling super-diffusive transport.

2. $|\Delta| > 1$, $S_0^z = 0$: spin transport is purely diffusive since there are no spin-flip odd conserved charges besides the total magnetization $S_0^z$ and therefore spin Drude weight is zero, see for example [99]. Numerical and some analytical analysis were provided in [35,98,164–167] and in the low temperature regime some results were found within the low-energy field theory description [168–170]. However up to now a proof of diffusive dynamics at any finite temperatures was missing and no closed formula for the diffusion constant was known.

3. $\Delta = \pm 1$, $S_0^z = 0$: spin transport at finite temperature is super-diffusive, namely with an infinite diffusion constant and with zero Drude weight [6]. This is suggested by numerical simulations [98, 172], which show a dynamical exponent $z = 3/2$ instead of the usual $z = 2$ characterizing diffusive transport, and by some analytic results [99] which prove the divergence of the diffusion constant. However an understanding of this phenomena remains elusive for now.

In reference [81] the first case was studied, namely the dynamics at $|\Delta| < 1$ and half-filling $S_0^z = 0$. There we showed that by choosing a $\Delta$ close to $1^-$, the ballistic dynamics

---

[6]While this is true at thermal equilibrium [99], non-equilibrium states can display normal diffusion, see for example [171].

slows down and the viscous terms become important, so that the full Navier-Stokes corrections are necessary to describe the exact dynamics at intermediate times. We here instead focus on the gapped regime $|\Delta| > 1$ and we show that transport is indeed purely diffusive on equilibrium states at half-filling. We define the hydrodynamic local magnetization density as a function of a smooth space variable $x \in \mathbb{R}$, such that

$$\langle S^z(j,t) \rangle \Big|_{j=x} \equiv s_0^z(x,t) \tag{6.3}$$

for any site $j$ in the chain; this is valid when variations of $s_0^z(x,t)$ occur on large distances. The same can be done for all the local conserved densities $\mathfrak{q}(x,t)$. As showed in the following sections, the hydrodynamic equation for the local magnetization reads as

$$\partial_t s_0^z(x,t) + \partial_x \left( v_\infty^{\text{eff}}(x,t) s_0^z(x,t) \right) = \frac{1}{2} \partial_x \mathfrak{D}_{00}(x,t) \partial_x s_0^z(x,t)$$
$$+ \frac{1}{2} \partial_x \sum_{j=1}^\infty \mathfrak{D}_{0j}(x,t) \partial_x \langle \mathfrak{q}_j(x,t) \rangle, \tag{6.4}$$

with the velocity $v_\infty^{\text{eff}}$ defined after equation (6.21) as the velocity of the largest quasiparticle inside the reference state. In the limit of linear perturbation on top of a *half-filled equilibrium state* (e.g. a thermal state), both the velocity $v_\infty^{\text{eff}}$ and the diffusion coefficients $\mathfrak{D}_{0j}$ with $j > 0$, evaluated on the reference state, vanish (in accordance with the absence of a finite spin Drude weight at half-filling [99]). Therefore we obtain a standard diffusion equation for the local magnetization

$$\partial_t s_0^z(x,t) = \frac{\mathfrak{D}_{00}}{2} \partial_x^2 s_0^z(x,t), \tag{6.5}$$

where the diffusion constant $\mathfrak{D}_{00}$ is a functional of the thermodynamic properties of the reference state (e.g. its temperature). In the next sections we shall show, as consequence of result (4.22), that the spin diffusion constant is exactly given as

$$\frac{\mathfrak{D}_{00}}{2} = \widetilde{w}_\infty \equiv \frac{1}{2} \sum_{b=1}^\infty \int_{-\pi/2}^{\pi/2} d\alpha \, \rho_{p,b}(\alpha)(1 - n_b(\alpha)) |v_b^{\text{eff}}(\alpha)| \, (\mathcal{W}_b)^2, \tag{6.6}$$

with $\widetilde{w}_\infty$ the variance (6.33) for the large-scale fluctuations of the bound state with infinite size $a = \infty$. The function

$$\mathcal{W}_b = \lim_{a \to \infty} \left( T_{a,b}^{\text{dr}}(\theta, \alpha) / \rho_{s,a}(\theta) \right), \tag{6.7}$$

which can be seen as the effective shift of the trajectory of the largest quasipartice with $a = \infty$ when it scatters with the quasiparticle $b$, it is a constant function in $\theta$ and $\alpha$. Formula (6.6) is a direct consequence of our result (4.15) and therefore it provides the *exact spin diffusion constant* of the model, provided the validity of our conjecture on the poles of the form factors, see sec. 3.3. It can be numerically evaluated, see Figure 3. In the infinite temperature limit $T \to \infty$ it agrees with another recent result [173] found at the same time as this paper appeared. Notice that for reference states with vanishing particle or hole density $\rho_{p,a}(\theta)(1 - n_a(\theta)) = 0$ the diffusion constant is zero. This is the case for example for the fully polarized domain-wall state or other reference states that display sub-diffusive corrections to ballistic spin transport, see [174].

In the following sections we shall review how to describe the thermodynamic limit of the a gapped XXZ chain and arrive to the final result (6.6).

## 6.1 A reminder of the the thermodynamic limit of the XXZ chain

The gapped regime $|\Delta| \geq 1$ is characterized by the presence of an infinite number of bound states or strings and by a compact support for the rapidites in the model $\theta \in [-\pi/2, \pi/2]$. Due to the infinite number of strings, the dressing integral equations $h_a^{\mathrm{dr}} = [(1 - Tn)^{-1}]_{ab} h_b$ can be recast into a factorized form, which is given by

$$h_{i;a}^{\mathrm{dr}} = d_{i;a} + s * (h_{i;a-1}^{\mathrm{dr}}(1 - n_{a-1}) + h_{i;a+1}^{\mathrm{dr}}(1 - n_{a+1})), \tag{6.8}$$

with $s(\theta) = (2\cosh(\eta\theta))^{-1}$ and with $*$ denoting convolution operation $f * g = \int_{-\pi/2}^{\pi/2} f(u - v) g(v) \mathrm{d}v$. The driving function $d_{i;a}(\theta)$ depends on the type of conserved quantity considered. Energy, momentum and all the other ultra-local (in contrast with quasi-local) conserved quantities have drivings terms of the type

$$d_{i;a} = \delta_{a,1}(1 + T_{11})^{-1} h_i, \tag{6.9}$$

with $h_i$ the single-particle eigenvalue. Quasi-local conserved charges [95, 175, 176] represent higher strings and their driving terms couple to larger strings, as $d_{i;a} \sim \delta_{a,s}$ with $s > 1$. The only conserved quantity which is not included in this set of families is the spin magnetization $\mathfrak{q}_0 = L^{-1} \sum_j S_j^z$, which is orthogonal to all the other charges, being the only spin-flip non invariant conserved operator. Its absolute value enters the dressing (6.8) via the value of the asymptotic of the dressed functions (and, equivalently, of the root densities), namely

$$\lim_{a \to \infty} a^{-1} \log h_{i;a}^{\mathrm{dr}} = -\mu, \tag{6.10}$$

where $\mu$ is the chemical potential associated to the magnetization charge $\mathfrak{q}_0$.

Let us now consider a thermal state at finite temperature $\beta$ and chemical potential $\mu$ and analyse the infinite temperature limit. In the limit $\beta \to 0$ the occupation numbers can be expressed analytically and they are constants in $\theta$ [99, 112]

$$n_a(\theta) = \frac{\sinh \mu}{\sinh(\mu(a+1))^2}. \tag{6.11}$$

In the limit $\mu \to 0$ also the root densities and the derivative of the energies take a simple form

$$n_a(\theta) = \frac{1}{(a+1)^2}, \tag{6.12}$$

$$\rho_{\mathrm{p},a}(\theta) = \frac{1}{2}(a+1)^{-1} \left( \frac{K_a(\theta)}{a} - \frac{K_{a+1}(\theta)}{a+2} \right), \tag{6.13}$$

$$\varepsilon_a'(\theta) = -\pi \frac{s+1}{2} \left( \frac{K_a'(\theta)}{a} - \frac{K_{a+2}'(\theta)}{a+2} \right), \tag{6.14}$$

with the functions $K_a(\theta)$ defined with the notation $\Delta = \cosh \eta$ as

$$K_a(\theta) = \frac{\sinh(\eta a)}{\pi(\cosh(\eta a) - \cos(2\theta))}. \tag{6.15}$$

From these expressions and from the ones at finite chemical potential one can notice that all thermodynamic functions converge to constant function in $\theta$ at large string number $a$,

with deviation of order $e^{-a\eta}$. Such constants also decay at large $a$ for any thermal state, not necessarily with infinite temperature, with the following different behaviours at large $a$

$$\rho_{\mathrm{p},a}(\theta) \sim \begin{cases} e^{-\mu a} & \text{if } \mu > 0 \\ 1/a^3 & \text{if } \mu = 0 \end{cases} \quad , \quad \rho_{\mathrm{s},a}(\theta) \sim \begin{cases} \text{const} & \text{if } \mu > 0 \\ 1/a & \text{if } \mu = 0 \end{cases}, \tag{6.16}$$

with a constant in front which is $\beta$−dependent. Moreover at thermal equilibrium the velocities decay exponentially in $a$ as [59, 99]

$$v_a^{\mathrm{eff}} \sim e^{-a\eta} \tag{6.17}$$

for any $\mu$. The behavior of the dressed magnetization $h_{0;a}^{\mathrm{dr}}(\theta)$ at finite temperature in this limit is very non-trivial. We have

$$h_{0;a}^{\mathrm{dr}}(\theta) = \begin{cases} \mu/3(a + \kappa(\beta))^2 + O(\mu^2) & \text{if } a \ll 1/\mu \\ a & \text{if } a \gg 1/\mu \end{cases}, \tag{6.18}$$

with $\kappa(0) = 1$ at infinite temperature, where the expression indeed reads [99]

$$\lim_{\beta \to 0} h_{0;a}^{\mathrm{dr}}(\theta) = \frac{1}{2} \frac{\sinh(\mu(a+1))}{\sinh \mu} \left( \frac{a}{\sinh(\mu a)} - \frac{(a+2)}{\sinh(\mu(a+2))} \right), \tag{6.19}$$

and it is easy from this expression to infer the behaviour (6.18).

## 6.2 Spin dynamics in the gapped XXZ chain

Given a reference equilibrium state $|\rho_{\mathrm{p},a}\rangle$, the eigenvalue of the local magnetization $s_0^z$, namely $h_{0;a}(\theta) = a$, is such that only the asymptotic density of holes determine the absolute value of the local magnetization

$$|s_0^z| = \frac{1}{2} - \sum_{a \geq 1} \int_{-\pi/2}^{\pi/2} \mathrm{d}\theta \rho_{\mathrm{p},a}(\theta) a = \lim_{a \to \infty} \rho_{\mathrm{h},a} \equiv \rho_{\mathrm{h},\infty}, \tag{6.20}$$

where $\rho_{\mathrm{h},\infty}$ is indeed a constant in $\theta$ for all stationary states. This relation is due to the recursive structure of the dressing $\rho_{\mathrm{p}} = n(1 - Tn)^{-1}p'$. It is easy to show that such a structure leads to a telescopic sum in (6.20). The same relation is valid for the absolute value of the expectation value of the spin current [59]

$$|\bar{\jmath}_0^z| = \sum_{a \geq 1} \int_{-\pi/2}^{\pi/2} \mathrm{d}\theta \, \rho_{\mathrm{p},a}(\theta) v_a^{\mathrm{eff}}(\theta) a = \rho_{\mathrm{h},\infty} v_\infty^{\mathrm{eff}} \equiv v_\infty^{\mathrm{eff}} |s_0^z|, \tag{6.21}$$

with $v_\infty^{\mathrm{eff}} = \lim_{a \to \infty} v_a^{\mathrm{eff}}(\theta)$ is the local velocity of the largest bound state, namely the one with infinite size $a$. The sign of the magnetization $\mathfrak{f}$, see [59], and of the spin current is given by the choice of the reference state used to construct the thermodynamic states and it is an information not contained in the set $\rho_{\mathrm{p},a}$. If the reference state is the spin up ferromagnetic state $|\uparrow \ldots \uparrow\rangle$ then $\mathfrak{f} = 1$, otherwise one can also choose the opposite state $|\downarrow \ldots \downarrow\rangle$ and have $\mathfrak{f} = -1$. Namely, given a choice of the reference state, any state $|\rho_{\mathrm{p}}\rangle$ can have either magnetization $s_0^z \in [0, 1/2]$ or $s_0^z \in [-1/2, 0]$. Therefore a generic

stationary state in the gapped regime $\Delta \geq 1$ is specified by the root densities and the magnetization sign $\mathfrak{f}$, namely by the set

$$|\rho_{\mathrm{p},a}, \mathfrak{f}\rangle, \tag{6.22}$$

for any normalizable densities $\sum_{a \geq 1} a \int_{-\pi/2}^{\pi/2} \rho_{\mathrm{p},a}(\theta) \mathrm{d}\theta \leq 1/2$ and $\mathfrak{f} = \pm 1$. This is now a complete set of states in the thermodynamic limit.

In [59] it was found that given an initial profile of magnetization that crosses zero in some points, its sign $\mathfrak{f}(x)$ evolves in time by a local continuity equation

$$\partial_t \mathfrak{f}(x,t) + v_\infty^{\mathrm{eff}}(x,t)\partial_x \mathfrak{f}(x,t) = 0, \tag{6.23}$$

where the equation is to be intended as an equation for the positions $\{x_0^j(t)\}$ of the zeros of the magnetization, since $\partial_x \mathfrak{f}(x,t) = \sum_j \delta(x - x_0^j(t))$. This simply means that the zero of the magnetization are transported by the flow with $\dot{x}_0^j(t) = v_\infty^{\mathrm{eff}}(x_0^j(t),t)$. Using $|s_0^z|\mathfrak{f} = s_0^z$, we arrive to equation (6.4) with the values of the local charges (even under spin-flip) given as usual by the root densities

$$\langle \mathfrak{q}_i(x,t)\rangle = \sum_a \int_{-\pi/2}^{\pi/2} \mathrm{d}\theta \, \rho_{\mathrm{p},a}(\theta; x, t) h_{i;a}(\theta) \qquad \forall i > 0. \tag{6.24}$$

### 6.3 Spin diffusion constant at half-filling

We consider the spin dynamics in the linear regime at half-filling, namely when the total magnetization is zero $S_0^z = 0$. This limit is particularly important since the spin ballistic current vanishes and the local magnetization evolves according to equation (6.5), which is a purely diffusive dynamics. On the other hand this limit presents numerous technical difficulties that we will here address and show how to compute the diffusion constant in equation (6.6).

Due to the spin-flip symmetry of the reference state at half-filling, we find

$$(\mathfrak{D}C)_{0i} = \delta_{i0}(\mathfrak{D}C)_{00}, \tag{6.25}$$

with $\delta_{ij}$ the Kronecker delta. This is due to the fact that all conserved charges with $i > 0$ are invariant under a global spin flip transformation. Moreover for the same reason we also find

$$C_{0i} = \delta_{i0} C_{00}. \tag{6.26}$$

We then conclude that the diffusion matrix evaluated on a half-filling state is diagonal on the line corresponding to the magnetization charge and it can be then expressed from one single element of the $(\mathfrak{D}C)$ matrix as

$$\lim_{\mu \to 0} \mathfrak{D}_{00} = \frac{\lim_{\mu \to 0}(\mathfrak{D}C)_{00}}{\lim_{\mu \to 0} C_{00}}. \tag{6.27}$$

The spin susceptibility of a stationary state, defined as [114, 141, 177]

$$C_{00} = \lim_{L \to \infty} L^{-1} \sum_j \langle S^z(j,t) S^z(0,0)\rangle^c \tag{6.28}$$

is given by equation (3.37) trivially generalised to the presence of different strings

$$C_{00} = \sum_{a \geq 1} \int_{-\pi/2}^{\pi/2} d\theta \, \rho_{p,a}(\theta)(1 - n_a(\theta))(h_{0;a}^{dr}(\theta))^2. \tag{6.29}$$

Using the properties of the dressed spin (6.18) and of the root densities at finite temperature and large $a$ (6.16), one easily realizes that the limit $\mu \to 0$ and the sum over the infinite number of strings do not commute. This can be seen as follows: due to (6.18) the contributions from the first strings vanishes in this limit and only large strings are relevant, namely the ones with $a \gtrsim 1/\mu$. Note that we cannot simply substitute $h_{0;a}^{dr}(\theta) \to a$, following (6.18), and use $\lim_{\mu \to 0} \rho_{p,a} \sim a^{-3}$, the root distributions at half-filling, as the sum would reduce to

$$\lim_{\mu \to 0} C_{00} \simeq \sum_{a \gg 1/\mu} \int_{-\pi/2}^{\pi/2} d\theta \, \rho_{p,a}(\theta)(1 - n_a(\theta))a^2, \tag{6.30}$$

which is a logarithmically diverging sum. Instead, for $a \gtrsim 1/\mu$, the root distribution $\rho_{p,a}$ does not take its half-filling form, and still is exponentially convergent as $\rho_{p,a} \sim e^{-\mu a}$. The clearest way is to perform the sum over the string types in (6.29) with a finite chemical potential $\mu$, and only then the limit $\mu \to 0$ can be taken. In the infinite temperature limit $\beta \to 0$ one finds this way the expected finite result for the half-filling state

$$\lim_{\mu \to 0} \lim_{\beta \to 0} C_{00} = \lim_{\mu \to 0} \lim_{\beta \to 0} \sum_{a \geq 1} \int_{-\pi/2}^{\pi/2} d\theta \, \rho_{p,a}(\theta)(1 - n_a(\theta))(h_{0;a}^{dr}(\theta))^2 = 1/4. \tag{6.31}$$

Notice also that since $n_a \to 0$ at large $a$ the contribution of the statistical factor $1 - n_a$ is irrelevant for such computation and it could be set to 1 from the start (clearly only when we are interested in the half-filling limit at $\mu = 0$).

Let us now compute the spin diffusion constant by computing $\lim_{\mu \to 0} (\mathfrak{D}C)_{00}$. It is useful to decompose the computation between the contribution to the diffusion constant given by the diagonal part of the diffusion kernel and the one given by the off-diagonal one, see eq. (4.24) and (4.27). Then we have $(\mathfrak{D}C)_{00} = (\mathfrak{D}C)_{00}^{diag} + (\mathfrak{D}C)_{00}^{off-diag}$

$$\lim_{\mu \to 0} (\mathfrak{D}C)_{00}^{diag} = \lim_{\mu \to 0} 2 \sum_{a \geq 1} \int_{-\pi/2}^{\pi/2} d\theta \, \tilde{w}_a(\theta) \rho_{p,a}(\theta)(1 - n_a(\theta)) \left( h_{0;a}^{dr}(\theta) \right)^2, \tag{6.32}$$

with the functions $\tilde{w}_a(\theta)$ defined previously in (4.29), should be indeed seen as the variance for the fluctuations of the trajectory of each quasiparticle with rapidity $\theta$ and string type $a$, due to the scattering processes with all the quasiparticles present inside the reference state

$$\tilde{w}_a(\theta) = \frac{1}{2} \sum_{b=1}^{\infty} \int d\alpha \, \rho_{p,b}(\alpha)(1 - n_b(\alpha)) \left( \frac{T_{a,b}^{dr}(\theta, \alpha)}{\rho_{s,a}(\theta)} \right)^2 |v_a^{eff}(\theta) - v_b^{eff}(\alpha)|. \tag{6.33}$$

One finds that the limit of large string number, this function is a finite constant, namely

$$\lim_{a \to \infty} \tilde{w}_a(\theta) = \tilde{w}_\infty, \tag{6.34}$$

given in equation (6.6), and it converges to the asymptotic value exponentially fast in the string length $a$, namely $\widetilde{w}_a(\theta) = \widetilde{w}_\infty + O(e^{-a\eta})$. Now, since the sum over $a$ in (6.32) in the limit $\mu \to 0$ is over $a \gtrsim 1/\mu$ one recovers the same summation as in the calculation of the spin susceptibility (6.29)

$$\lim_{\mu \to 0} \sum_{a \geq 1} \int_{-\pi/2}^{\pi/2} d\theta \, \widetilde{w}_a(\theta) \rho_{p,a}(\theta)(1 - n_a(\theta)) \left( h_{0;a}^{dr}(\theta) \right)^2$$

$$= 2\widetilde{w}_\infty \lim_{\mu \to 0} \sum_{a \gg 1/\mu} \int_{-\pi/2}^{\pi/2} d\theta \rho_{p,a}(\theta)(1 - n_a(\theta)) \left( h_{0;a}^{dr}(\theta) \right)^2 = 2\widetilde{w}_\infty \lim_{\mu \to 0} C_{00}, \quad (6.35)$$

where we used that, if $a \gtrsim 1/\mu$ in the limit $\mu \to 0$, we can simply substitute $\widetilde{w}_a \to \widetilde{w}_\infty$ inside the sum. Let us now consider the contribution from the off-diagonal part of the diffusion kernel

$$\lim_{\mu \to 0} (\mathfrak{D}C)_{00}^{\text{off-diag}} = -\lim_{\mu \to 0} \sum_{a,b \geq 1} \int_{-\pi/2}^{\pi/2} d\theta d\alpha \, h_{0;a}^{dr}(\theta) \rho_{p,a}(\theta)(1 - n_a(\theta))$$

$$\times \frac{\left[ T_{a,b}^{dr}(\theta, \alpha) \right]^2}{\rho_{s,a}(\theta) \rho_{s,b}(\alpha)} |v_a^{\text{eff}}(\theta) - v_b^{\text{eff}}(\alpha)| \rho_{p,b}(\alpha)(1 - n_b(\alpha)) h_{0;b}^{dr}(\alpha). \quad (6.36)$$

The contribution can only be finite in the limit $\lim_{\mu \to 0}$ if the sum over the strings types at $\mu = 0$ diverges at large $a, b$, so that the limit and the sum cannot be commuted, as it is the case for the diagonal case (6.32). Let us then analyse the asymptotic $b \to \infty$ behaviour at fixed $a$. The dressed scattering kernel $T_{a,b}^{dr}$ decays at large $b$ with the same power as the density of states

$$T_{a,b}^{dr}(\theta, \alpha) \sim b^{-1}. \quad (6.37)$$

Therefore we find that the sum decays as $1/b^3$ and that the same is true for the sum over $a$ at fixed $b$. The sum also converges if one sums over $a$ and with $b \sim a$ due to the exponential vanishing of the effective velocities. Therefore the sum in (6.36) converges if we bring the limit $\mu \to 0$ inside the sum, which implies that the sum and the limit can be exchanged, giving a vanishing contribution at half-filling from the off-diagonal parts $\lim_{\mu \to 0}(\mathfrak{D}C)_{00}^{\text{off-diag}} = 0$. We can then finally conclude that the diffusion constant of half-filling stationary states is given by

$$\frac{1}{2} \lim_{\mu \to 0} \mathfrak{D}_{00} = \frac{\lim_{\mu \to 0}(\mathfrak{D}C)_{00}^{\text{diag}}}{\lim_{\mu \to 0} C_{00}} = \widetilde{w}_\infty. \quad (6.38)$$

# 7 Conclusion

In this paper we have shown that there exists a large-scale description of the non-equilibrium dynamics of generic integrable models that also accounts for diffusive and dissipative effects. We derived the hydrodynamic theory from a gradient expansion of the expectation values of the local currents, which allows us to obtain hydrodynamic equations of Navier-Stokes type. In order to compute the diffusion matrix for a generic stationary state we employed the so-called form factor expansion to evaluate the

necessary dynamical correlation functions of current operators. Such expansion directly connects the generalised Navier-Stokes equation to the presence of scattering processes among the quasiparticles, which are responsible for the decay of the current-current correlator and therefore for the presence of finite diffusion constants. Moreover we showed that the diffusion constants of the model are entirely given by two-body scatterings among quasiparticles, while higher scattering processes only determine sub-leading time scales. We presented an exact expression for the diffusion matrix which applies to any integrable model with a thermodynamic description in terms of quasiparticles. In particular we computed the spin diffusion constant for a XXZ spin chain at finite temperature, which constituted a long-standing open problem. We believe our expression can give new insights into the timely problem of computing diffusion constants in many-body systems, see for example [145].

This paper provides a comprehensive description of the recently developed generalised hydrodynamics with diffusive terms, and opens the way to a number of future directions. First, our work shows how to compute the dynamical correlation functions in the thermodynamic limit via quasiparticle excitation processes. While in the current work we only employed a restricted part of the whole spectral sum, one may think of extending it to include higher particle-hole numbers and to fully determine the thermodynamic form factors of local conserved densities or current operators. This would give access to the full collision integrals for the quasiparticles, therefore, in a sense, leading to a new "generalised Boltzmann equation" for interacting integrable models.

The exact results for the diffusion constant of a XXZ chain, and the possible extension to several other model such as the Fermi-Hubbard model, can now provide a perfect playground to test numerical methods. The state of the art in numerical techniques, at the present, seem unable to reach the time scales necessary to fully reconstruct the diffusive dynamics of the system, although some recent developments are encouraging [70]. This will constitute a future challenge for the community working on numerical algorithms for many-body systems.

The divergence of the spin diffusion constant in a XXZ chain in the Heisenberg limit motivates the study of possible super-diffusive transport dynamics in integrable models with isotropic interactions [99]. The origins of such super-diffusive behaviour, and the connections with integrability, are at the present not understood and their quest represents a clear direction to be taken in future researches.

The quasiparticle scattering processes introduced in this work can also, in principle, be employed to characterize other physical phenomena, such as the presence of integrability breaking terms in the Hamiltonian giving the time evolution, see for example [178, 179].

The small temperature limit expansion of the diffusion matrix can be studied and possibly compared with field theoretical techniques that can access the low energy spectrum of microscopic model, see for example [165, 180]. This would give new insights into which types of interactions one need to include in the low-energy field theoretical description in order to describe the diffusive dynamics.

Finally it would be necessary to rule out the possible existence of slow decoherence modes which are not included in the hydrodynamic theory. While some recent works seem to point out that under some mild assumption in quantum many-body systems these modes are absent at diffusive scales [145], a rigorous proof of such statement in interacting integrable model is still lacking, with some results only available in free fermionic theories [86].

**Acknowledgments:** The authors are indebted to Christoph Karrasch for kindly providing tDMRG data in this and in previous related works. They kindly acknowledge Subir Sachdev for illuminating discussions on the low temperature limit of the spin diffusion constant. Moreover they thank Romain Vasseur, Sarang Gopalakrishnan, Enej Ilievski, Tomaž Prosen, Bruno Bertini, Jerôme Dubail, Pasquale Calabrese, Márton Mestyán, Tomohiro Sasamoto, Herbert Spohn and Vir Bulchandani for useful and insightful discussions. D.B. acknowledges support from the CNRS and the ANR via the project under contract number ANR-14-CE25-0003. J.D.N. is supported by the Research Foundation Flanders (FWO) and in the early stage of the work by the LabEx ENS-ICFP:ANR-10-LABX-0010/ANR-10-IDEX-0001-02 PSL*. J.D.N. acknowledges SISSA for hospitality. B.D.'s research was supported in part by Perimeter Institute for Theoretical Physics. Research at Perimeter Institute is supported by the Government of Canada through the Department of Innovation, Science and Economic Development and by the Province of Ontario through the Ministry of Research, Innovation and Science. B.D is also grateful to École Normale Supérieure de Paris for an invited professorship from February 19th to March 20th 2018, where a large part of this work was carried out, as well as hospitality and funding from the Erwin Schrödinger Institute in Vienna (Austria) during the program "Quantum Paths" from April 9th to June 8th 2018 and the GGI in Florence (Italy) during the program "Entanglement in Quantum Systems". B.D. thank the Centre for Non-Equilibrium Science (CNES) and the Thomas Young Centre (TYC).

## A  A direct proof of the sum rule (2.13)

We start with the right-hand side of (2.13) and simply use the conservation laws and space and time translation invariance:

$$
\begin{aligned}
&\int_0^t ds \int_0^t ds' \int dx \, \langle j_i(x,s) j_j(0,s') \rangle^c \\
&= -\int_0^t ds \int_0^t ds' \int dx \, x \partial_x \langle j_i(x,s) j_j(0,s') \rangle^c \\
&= \int_0^t ds \int_0^t ds' \int dx \, x \partial_s \langle q_i(x,s) j_j(0,s') \rangle^c \\
&= \int_0^t ds' \int dx \, x \langle (q_i(x,t) - q_i(x,0)) j_j(0,s') \rangle^c \\
&= -\int_0^t ds' \int dx \, x \langle (q_i(0,t) - q_i(0,0)) j_j(x,s') \rangle^c \\
&= \int_0^t ds' \int dx \, \frac{x^2}{2} \partial_x \langle (q_i(0,t) - q_i(0,0)) j_j(x,s') \rangle^c \\
&= -\int_0^t ds' \int dx \, \frac{x^2}{2} \partial_{s'} \langle (q_i(0,t) - q_i(0,0)) q_j(x,s') \rangle^c \\
&= -\int dx \, \frac{x^2}{2} \langle (q_i(0,t) - q_i(0,0))(q_j(x,t) - q_j(x,0)) \rangle^c \\
&= -\int dx \, \frac{x^2}{2} \langle (q_i(x,t) - q_i(x,0))(q_j(0,t) - q_j(0,0)) \rangle^c \\
&= \int dx \, \frac{x^2}{2} \big( S_{ij}(x,t) + S_{ij}(x,-t) - 2 S_{ij}(x,0) \big).
\end{aligned}
\tag{A.1}
$$

## B  A proof of the equation of motion (2.19)

By the conservation laws, it is immediate that

$$
\partial_t S_{ij}(x,t) + \partial_x \langle j_i(x,t) q_j(0,0) \rangle^c = 0.
\tag{B.1}
$$

Let us now evaluate the two-point function $\langle j_i(x,t) q_j(0,0) \rangle^c$ using the hydrodynamic expansion (2.9). For this purpose, we combine two assumptions. First, the main assumption of hydrodynamics, that all local averages at time $t$ are completely determined by the knowledge of $\{\bar{q}(x,t) : x \in \mathbb{R}, i \in I\}$, that is,

$$
\langle o(x,t) \rangle = \mathcal{O}[\bar{q}.(\cdot,t)](x,t).
\tag{B.2}
$$

Second, *causality*, that the state at time $t$ is completely determined by that at any given time $s < t$. This means that, for any given $s$, the average current $\langle j_i(x,t) \rangle$ at a later time $t > s$ is a functional of $\{\bar{q}(x,s) : x \in \mathbb{R}, i \in I\}$. In what follows, we first assume $t > 0$ and take $s = 0$.

By standard linear response arguments, small perturbations of the state at time $0$ will introduce, in the average $\langle j_i(x,t) \rangle$, local observables at time $0$. Since the state is

determined by all conserved densities, it is expected that there be perturbations that insert conserved densities. Let us define parameters $\beta_j(x)$ which exactly play this role:

$$\frac{\delta\langle\mathfrak{o}(y,t)\rangle}{\delta\beta_j(x)} = \langle\mathfrak{o}(y,t)\mathfrak{q}_j(x,0)\rangle^c. \tag{B.3}$$

At the Euler scale, we may understand the state at time 0, where each fluid cell has maximised entropy with respect to the available conserved charges, to be of the form $\exp\left[\int dx \sum_j \beta_j(x)\mathfrak{q}_j(x)\right]$. This reproduces (B.3). We assume that at the diffusive scale, there also exist perturbations of homogeneous states described by $\beta_j(x)$ such that (B.3) holds.

Applying (B.3) and the principles of hydrodynamics, we can simply use the chain rule in order to obtain (2.19):

$$
\begin{aligned}
\langle\mathfrak{j}_i(x,t)\mathfrak{q}_j(0,0)\rangle^c &= \frac{\delta\bar{\mathfrak{j}}_i(x,t)}{\delta\beta_j(0)} \\
&= \int dy \sum_k \frac{\delta\bar{\mathfrak{j}}_i(x,t)}{\delta\bar{\mathfrak{q}}_k(y,t)} \frac{\delta\bar{\mathfrak{q}}_k(y,t)}{\delta\beta_j(0)} \\
&= \int dy \sum_k \left(A_i^{\ k}\delta(x-y) - \frac{1}{2}\mathfrak{D}_i^{\ k}\partial_x\delta(x-y)\right)\langle\mathfrak{q}_k(y,t)\mathfrak{q}_j(0,0)\rangle^c \\
&= \sum_k \left(A_i^{\ k} - \frac{1}{2}\mathfrak{D}_i^{\ k}\partial_x\right)\langle\mathfrak{q}_k(x,t)\mathfrak{q}_j(0,0)\rangle^c,
\end{aligned}
\tag{B.4}
$$

where in the third line we used (2.9) and (2.20), and homogeneity of the state. Combining with (B.1), we find (2.19).

Let us now consider negative times $t < 0$. In this case the above proof does not hold, because we cannot assume that averages $\langle\mathfrak{o}(y,t)\rangle$ are completely determined by $\{\bar{\mathfrak{q}}(x,s) : x \in \mathbb{R}, i \in I\}$ for $s > t$ (including $s = 0$). This is because beyond the Euler scale, the hydrodynamic approximation of the time evolution is generically *not reversible*: in particular, information is lost as time goes on, and later configurations do not determine earlier configurations.

However, we can establish the following general symmetry relation:

$$\langle\mathfrak{j}_i(x,t)\mathfrak{q}_j(0,0)\rangle^c = \langle\mathfrak{q}_i(x,t)\mathfrak{j}_j(0,0)\rangle^c. \tag{B.5}$$

This is shown by evaluating the space derivative using the conservation laws and using homogeneity and stationarity:

$$
\begin{aligned}
\partial_x\langle\mathfrak{j}_i(x,t)\mathfrak{q}_j(0,0)\rangle^c &= -\partial_t\langle\mathfrak{q}_i(x,t)\mathfrak{q}_j(0,0)\rangle^c \\
&= -\partial_t\langle\mathfrak{q}_i(0,0)\mathfrak{q}_j(-x,-t)\rangle^c \\
&= \partial_x\langle\mathfrak{q}_i(0,0)\mathfrak{j}_j(-x,-t)\rangle^c \\
&= \partial_x\langle\mathfrak{q}_i(x,t)\mathfrak{j}_j(0,0)\rangle^c.
\end{aligned}
$$

Therefore, the left- and right-hand side of (B.5) can only differ by a function of $t$. Taking $x \to \infty$ and using clustering, this function must be zero.

We can now use (B.5) in order to perform a symmetric version of the derivation (B.4), obtaining, for $t < 0$,

$$
\begin{aligned}
\langle j_i(x,t) q_j(0,0) \rangle^c &= \langle q_i(0,0) j_j(-x,-t) \rangle^c \\
&= \sum_k \left( A_j^{\ k} + \frac{1}{2} \mathfrak{D}_j^{\ k} \partial_x \right) \langle q_i(x,t) q_j(0,0) \rangle^c .
\end{aligned}
$$

Note how the summation is over the index of the rightmost conserved density, instead of the leftmost one, in $\langle q_i(x,t) q_j(0,0) \rangle^c$.

## C  Gauge covariance and gauge fixing by $\mathcal{PT}$-symmetry

### C.1  Gauge covariance

Let us first discuss the covariance of the hydrodynamics data under the redefinition of the local charges via

$$
q_i(x,t) \to q_i'(x,t) = q_i(x,t) + \partial_x \mathfrak{o}_i(x,t). \tag{C.1}
$$

The Drude coefficients $D_{ij}$ and the Onsager matrix $\mathfrak{L}_{ij}$ are both invariant under this gauge transformation, which is physically sound as they coded for the microscopic ballistic and diffusive spreading of the correlation functions. Indeed, under the gauge transformation $q_i(x,t) \to q_i'(x,t) = q_i(x,t) + \partial_x \mathfrak{o}_i(x,t)$, the current transform as

$$
j_i(x,t) \to j_i'(x,t) = j_i(x,t) - \partial_t \mathfrak{o}_i(x,t). \tag{C.2}
$$

Hence, the integrand in the double integral $\int_0^t ds \int_0^t ds' \int dx \langle j_i(x,s) j_j(0,s') \rangle^c$ in equation (2.13) is modified by total derivative only, so that this double integral only acquires boundary under gauge transformation. That is we get:

$$
\begin{aligned}
&\int_0^t ds' \int dx \left[ \langle \mathfrak{o}_i(x,t) j_j(0,s') \rangle^c - \langle \mathfrak{o}_i(x,0) j_j(0,s') \rangle^c \right] \\
&+ \int_0^t ds \int dx \left[ \langle j_i(x,s) \mathfrak{o}_j(0,t) \rangle^c - \langle j_i(x,s) \mathfrak{o}_j(0,0) \rangle^c \right] + O(t^0).
\end{aligned}
$$

If $\mathfrak{o}_i(x,t)$ is a local conserved charge, then these terms vanish by global conservation law. If not, by the hydrodynamic projection mechanism [72,101] only the part of the operator projecting non-trivially on the conserved charges contribute at large time. Hence, these boundary terms produce $O(t^0)$ contribution and thus do not contribute to the leading terms in (2.14).

Of course the coefficients $\mathfrak{D}_j^{\ k}$ are not invariant under this gauge transformation because the local charges are modified hence their dynamical equations. Under the gauge transformations (C.1,C.2), the charge and current expectations are modified according to

$$
\bar{q}_i(x,t) \to \bar{q}_i'(x,t) = \bar{q}_i(x,t) + \partial_x \bar{\mathfrak{o}}_i(x,t), \quad \bar{j}_i(x,t) \to \bar{j}_i'(x,t) = \bar{j}_i(x,t) - \partial_t \bar{\mathfrak{o}}_i(x,t).
$$

In the hydrodynamic approximation, $\bar{\mathfrak{o}}_i(x,t) = \mathcal{O}_i[\bar{q}.(x,t)]$ at the Euler scale (which is sufficient at the order of the derivative expansion we are dealing with). Using the

chain rule to compute $\partial_t \bar{\mathfrak{o}}_i(x,t)$ and $\partial_x \bar{\mathfrak{o}}_i(x,t)$, one then gets that $\mathcal{F}_i$ is invariant and $\mathfrak{D}_i^{\ j}$ transforms as

$$\mathfrak{D}_i^{\ j} \to \mathfrak{D}'^{\ j}_i = \mathfrak{D}_i - 2\sum_k \left[ A_i^{\ k} \frac{\partial \mathcal{O}_k}{\partial \bar{\mathfrak{q}}_j} - \frac{\partial \mathcal{O}_i}{\partial \bar{\mathfrak{q}}_k} A_k^{\ j} \right].$$

## C.2 Gauge fixing

Consider $\mathcal{PT}$-symmetry as defined in subsection 2.4.

We first show that there is a unique choice of a "proper" gauge, under the gauge transformation (2.28), such that (2.30) holds. The proper gauge transformations are those which preserve reality of the local conserved densities: we will simply ask that the local observables $\mathfrak{o}_i(x,t)$ in (2.28) have real averages in homogeneous, stationary, maximal entropy states.

First, assuming $\mathcal{PT}$-symmetry, we have (2.29):

$$\mathsf{T}\mathfrak{q}_i(x,t)\mathsf{T}^{-1} = \mathfrak{q}_i(-x,-t) + \partial_x \mathfrak{a}_i(-x,-t). \tag{C.3}$$

By the fact that $\mathsf{T}$ is an involution, applying (C.3) twice we obtain

$$\partial_x \left( \mathsf{T}\mathfrak{a}_i(x,t)\mathsf{T}^{-1} - \mathfrak{a}_i(-x,-t) \right) = 0. \tag{C.4}$$

Let us define $\mathfrak{q}'_i(x,t) = \mathfrak{q}_i(x,t) + \partial_x \mathfrak{o}_i(x,t)$. Then

$$\mathsf{T}\mathfrak{q}'_i(x,t)\mathsf{T}^{-1} = \mathfrak{q}'_i(-x,-t) + \partial_x \left( \mathsf{T}\mathfrak{o}_i(x,t)\mathsf{T}^{-1} + \mathfrak{o}_i(-x,-t) + \mathfrak{a}_i(-x,-t) \right). \tag{C.5}$$

Choosing

$$\mathfrak{o}_i(x,t) = -\frac{\mathfrak{a}_i(x,t)}{2} \tag{C.6}$$

and using (C.4), this shows (2.30).

Second, we show that $\mathfrak{o}_i(x,t)$ has real averages. The only local observables that are independent of space are those proportional to the identity operator, thus eq.(C.4) implies $\mathsf{T}\mathfrak{a}_i(x,t)\mathsf{T}^{-1} = \mathfrak{a}_i(-x,-t) + c_i \mathbf{1}$. On the one hand, the involution property and anti-unitarity of $\mathsf{T}$ shows that $c_i$ must be purely imaginary. On the other hand, by shifting $\mathfrak{a}_i(x,t)$ by a pure imaginary constant times $\mathbf{1}$, the imaginary part of $c_i$ can be made to vanish. Hence

$$\mathsf{T}\mathfrak{a}_i(x,t)\mathsf{T}^{-1} = \mathfrak{a}_i(-x,-t). \tag{C.7}$$

Taking into account the fact that the anti-unitary transformation $\mathsf{T}$ changes averages to their complex conjugate, we therefore obtain $\mathrm{Im}(\langle \mathfrak{a}_i(x,t) \rangle) = 0$ in maximal entropy states, and thus $\mathrm{Im}(\langle \mathfrak{o}_i(x,t) \rangle) = 0$. This shows that there is a proper gauge choice leading to (2.30).

Finally, in order to show uniqueness, assume $\mathsf{T}\mathfrak{q}_i(x,t)\mathsf{T}^{-1} = \mathfrak{q}_i(-x,-t)$ and $\mathsf{T}(\mathfrak{q}_i(x,t) + \partial_x \tilde{\mathfrak{o}}_i(x,t))\mathsf{T}^{-1} = \mathfrak{q}_i(-x,-t) - \partial_x \tilde{\mathfrak{o}}_i(-x,-t)$. This implies that there exists $\tilde{c}_i$ such that $\mathsf{T}\tilde{\mathfrak{o}}_i(x,t)\mathsf{T}^{-1} = -\tilde{\mathfrak{o}}_i(-x,-t) + \tilde{c}_i \mathbf{1}$. By a shift, $\tilde{c}_i$ can be made purely imaginary, and we find $\mathrm{Re}(\langle \tilde{\mathfrak{o}}_i(x,t) \rangle) = 0$. Thus this is not a proper gauge transformation. Hence the proper gauge choice is unique.

We second show that we can choose the currents to be $\mathcal{PT}$-invariant, (2.31). Indeed, the choice of new currents given by $\mathfrak{j}'_i(x,t) = \mathfrak{j}_i(x,t) - \partial_t \mathfrak{o}_i(x,t)$ satisfies the conservation law $\partial_t \mathfrak{q}'_i(x,t) + \partial_x \mathfrak{j}'_i(x,t) = 0$, which implies $\partial_x \left( \mathsf{T}\mathfrak{j}'_i(x,t)\mathsf{T}^{-1} - \mathfrak{j}'_i(-x,-t) \right) = 0$. As a

consequence, there exists $c_i'$ such that $\mathsf{T}\mathsf{j}_i'(x,t)\mathsf{T}^{-1} = \mathsf{j}_i'(-x,-t) + c_i'\mathbf{1}$. By a similar argument as that leading to (C.7), we conclude that we can shift the currents by appropriate constants so that $c_i' = 0$.

## D  An alternative derivation of the current formula

Our analysis of the single-particle-hole form factors also gives a new proof of equation (3.18) for the expectation values of the currents on a GGE.

Consider

$$B_{ij} = \int \mathrm{d}x \, \langle \mathsf{j}_i(x,t)\mathsf{q}_j(0,0)\rangle^c = -\frac{\partial \langle \mathsf{j}_i\rangle}{\partial \beta^j}, \tag{D.1}$$

where $\langle \mathsf{j}_i\rangle$ is the average current in a GGE. Clearly, the conserved densities $\mathsf{q}_i(x,t)$ are operators that are linear functionals of the one-particle eigenvalues $h_i(\theta)$. As a consequence of the conservation laws, so are the currents $\mathsf{j}_i(x,t)$, hence so are their averages in GGEs, $\langle \mathsf{j}_i\rangle$. Therefore, we may write $\langle \mathsf{j}_i\rangle = \int \mathrm{d}\theta \, \rho_\mathrm{p}(\theta)\mathfrak{w}(\theta)h_i(\theta)$ for some state-dependent function $\mathfrak{w}(\theta)$. Hence we have

$$\partial_{\beta^j}\langle \mathsf{j}_i\rangle = \int \mathrm{d}\theta \, \partial_{\beta^j}\big(\rho_\mathrm{p}(\theta)\mathfrak{w}(\theta)\big)h_i(\theta). \tag{D.2}$$

On the other hand we can insert a resolution of particle-hole identity inside the correlator in (D.1). Only the one particle-hole states contribute, because of the integration over $x$ and conservation law for the charges (3.40) and (3.41), by the same reasons as in section 3.4. In the one particle-hole sector, the integration on $x$ forces the rapidities of the hole and the particle to be equal. Using (3.36), the single particle-hole contribution then yields

$$B_{ij} = \int \mathrm{d}\theta \rho_\mathrm{p}(\theta)(1-n(\theta))v^{\mathrm{eff}}(\theta)h_i^{\mathrm{dr}}(\theta)h_j^{\mathrm{dr}}(\theta), \tag{D.3}$$

where $v^{\mathrm{eff}}(\theta)$ is given by the ratio (3.20), and occurs by evaluating $[\varepsilon(\theta_\mathrm{p}) - \varepsilon(\theta_\mathrm{h})]/[k(\theta_\mathrm{p} - \theta_\mathrm{h}]$ at $\theta_\mathrm{p} = \theta_\mathrm{h}$. Finally, completeness of the set of functions $h_i(\theta)$ and equality of (D.2) and (D.3) gives a set of first-order $\beta_j$-differential equations for $\mathfrak{w}(\theta)$. One can check that one solution is indeed the effective velocity,

$$\mathfrak{w}(\theta) = v^{\mathrm{eff}}(\theta). \tag{D.4}$$

Assuming that this solution is unique, this proves the equation (3.18) for the expectation values of the currents on a generic GGE state. Recall that the values used for the one particle-hole form factors at equal particle-hole rapidities are only a consequence of the thermodynamic Bethe ansatz (see subsection 3.3). Hence the present derivation gives a proof that is independent from assumptions on form factors.

## E  Solving $T^{\mathrm{dr}}$ for an XXZ chain in the gapped regime

The equation for the dressed scattering - in the presence of strings - is

$$T_{a,b}^{\mathrm{dr}}(\theta,\theta') - \sum_c T_{a,c}(\theta,\alpha)n_c(\alpha)T_{c,b}^{\mathrm{dr}}(\alpha,\theta') = T_{a,b}(\theta,\theta'), \tag{E.1}$$

which can be rewritten in the usual factorized form as

$$T_{ab}^{\mathrm{dr}} - s \star (T_{a-1,b}^{\mathrm{dr}}(1-n_{a-1}) + T_{a+1,b}^{\mathrm{dr}}(1-n_{a+1})) = s(\delta_{a-1,b} + \delta_{a+1,b}), \qquad \text{(E.2)}$$

with the kernel

$$s(\theta) = \frac{1}{2\cosh \eta \theta}, \qquad \text{(E.3)}$$

with $\Delta = \cosh \eta$. Let us denote the generalized coefficients $f_a^{(b)}$

$$f_a^{(b)} - s \star (f_{a-1}^{(b)}(1-n_{a-1}) + f_{a+1}^{(b)}(1-n_{a+1})) = s\delta_{ab}. \qquad \text{(E.4)}$$

Then

$$T_{ab}^{\mathrm{dr}} = f_a^{(b-1)} + f_a^{(b+1)} \text{ with } f_0^{(b-1)} = 0. \qquad \text{(E.5)}$$

For $b = 1$ equation (E.4) is the equation for the density of states $\rho_{s,a}$. Higher $b$ correspond to higher-spin generalizations of the density of states. Solving numerically the equations for $f_a^{(b)}$ is a very non-trivial task. One indeed needs to truncate the number of string to a finite number $a_{\mathrm{max}}$ to solve them numerically. While for any fixed $b$ the equations (E.4) can be solved similarly to the equations for the density of states, at larger $b$ the solution become less and less accurate, as the driving term $s\delta_{ab}$ gets closer to the largest string. At the moment we have not found an efficient way to truncate the equations to obtain a precise result. The only case where it was possible to exactly solve it is at infinite temperature $\beta = 0$, where the recursive relation can be solved analitically for all $b$ and also for all $a$ (although it becomes increasingly expensive at larger $a$).

## F Thermodynamic limit of sums over excitations

In order to compute dynamical correlation functions in the thermodynamic limit, one needs to perform a multiple integration over all possible values of the rapidites of the particle-hole excitations. The form factors have however a pole singularity whenever $\theta_{\mathrm{p}}^i = \theta_{\mathrm{h}}^j$ and they are finite only when we consider only one single particle-hole $n = 1$ with $\theta_{\mathrm{p}} \to \theta_{\mathrm{h}}$, when the form factor becomes indeed diagonal. Therefore we need to be careful while rewriting the sums as integrals. The aim of this section is to show how this can be done. Let us start with the finite size form of the correlation function where we already neglect sub-leading corrections

$$\langle \mathfrak{o}(x,t)\mathfrak{o}(0,0)\rangle = \sum_{m=0}^{\infty} \frac{1}{m!^2} \prod_{j=1}^{m} \left[ \frac{1}{L} \sum_{\theta_{\mathrm{p}}^j} \frac{1}{L} \sum_{\theta_{\mathrm{h}}^j} \right]$$
$$\langle \rho_{\mathrm{p}}|\mathfrak{o}_i|\{\theta_{\mathrm{p}}^\bullet, \theta_{\mathrm{h}}^\bullet\}\rangle \langle \{\theta_{\mathrm{p}}^\bullet, \theta_{\mathrm{h}}^\bullet\}|\mathfrak{o}_j|\rho_{\mathrm{p}}\rangle e^{ixk[\theta_{\mathrm{p}}^\bullet, \theta_{\mathrm{h}}^\bullet] - it\varepsilon[\theta_{\mathrm{p}}^\bullet, \theta_{\mathrm{h}}^\bullet]}. \qquad \text{(F.1)}$$

The sum over particle and holes rapidites transforms into a product of integrals under a proper regularization. The idea is to write the sum over the holes as a complex integral over all the values that the holes rapidites can take for a finite (but large) $L$ using the following counting function for each hole

$$Q(\theta_{\mathrm{h}}) = L \int_{-\infty}^{\theta_{\mathrm{h}}} \rho_{\mathrm{p}}(u)du. \qquad \text{(F.2)}$$

Notice that in the thermodynamic limit all the correlation functions computed on any discretization of the state are all equivalent. Let us focus first on the sum over one of the holes, that we shall denote with $\theta_{\mathrm{h}}^1 \equiv \theta_{\mathrm{h}}$. We denote

$$
\begin{aligned}
F(z) &\equiv F_{\theta_{\mathrm{p}}^1,\dots,\theta_{\mathrm{p}}^m}(z, \theta_{\mathrm{h}}^2,\dots,\theta_{\mathrm{h}}^m) \\
&= \langle \rho_{\mathrm{p}} | \mathfrak{o}_i | \{\theta_{\mathrm{p}}^\bullet, \theta_{\mathrm{h}}^\bullet\} \rangle \langle \{\theta_{\mathrm{p}}^\bullet, \theta_{\mathrm{h}}^\bullet\} | \mathfrak{o}_j | \rho_{\mathrm{p}} \rangle e^{\mathrm{i}x k[\theta_{\mathrm{p}}^\bullet, \theta_{\mathrm{h}}^\bullet] - \mathrm{i}t\varepsilon[\theta_{\mathrm{p}}^\bullet, \theta_{\mathrm{h}}^\bullet]} \Big|_{\theta_{\mathrm{h}}^1 = z},
\end{aligned}
\tag{F.3}
$$

and with a help of $Q(\theta_{\mathrm{h}}^1)$ we can write the sum over all the values of hole rapidity $\theta_{\mathrm{h}}^1$ as

$$
\begin{aligned}
\frac{1}{L} \sum_{\theta_{\mathrm{h}}^1} F(\theta_{\mathrm{h}}^1) &= \sum_{I_j} \oint_{I_j} dz \, \frac{F(z)Q'(z)}{e^{2\pi \mathrm{i} Q(z)} - 1} \\
&= \left( \int_{\mathbb{R} - \mathrm{i}\epsilon} - \int_{\mathbb{R} + \mathrm{i}\epsilon} \right) \frac{F(z)Q'(z)}{e^{2\pi \mathrm{i} Q(z)} - 1} dz - 2\pi \mathrm{i} \sum_j \mathrm{Residue} \Big|_{z = \theta_{\mathrm{p}}^j} \frac{F(z)Q'(z)}{e^{2\pi \mathrm{i} Q(z)} - 1},
\end{aligned}
\tag{F.4}
$$

where the first integrals are taken on a single contour including the poles in $Q(z) =$ integers where integers are all the possible quantum numbers of the hole (a valid discretization of the state at finite $L$). In the second step we modified the sum over all these contours in the integral over the line above and below the real axes. In order to do that we need to subtract the poles of the form factors that we do not want to include in the sum, namely the sum over the residues of $F$ at the positions of the particles. Let us now consider the thermodynamic limit $L \to \infty$ of the first contribution. The real part of $2\pi \mathrm{i} Q(z - \mathrm{i}\epsilon)$ is positive and proportional to $L$, and therefore

$$
\int_{\mathbb{R} - \mathrm{i}\epsilon} \frac{F(z)Q'(z)}{e^{2\pi \mathrm{i} Q(z)} - 1} dz \to 0.
\tag{F.5}
$$

Using that $Q' = L\rho_{\mathrm{p}} + O(1)$ we then arrive in the thermodynamic limit to

$$
\frac{1}{L} \sum_{\theta_{\mathrm{h}}^1} F(\theta_{\mathrm{h}}^1) \to \int_{\mathbb{R} + \mathrm{i}\epsilon} F(z)\rho_{\mathrm{p}}(z) dz - \lim_{L \to \infty} 2\pi \mathrm{i} \sum_j \mathrm{Residue} \Big|_{z = \theta_{\mathrm{p}}^j} \frac{F(z)\rho_{\mathrm{p}}(z)}{e^{2\pi \mathrm{i} Q(z)} - 1}.
\tag{F.6}
$$

The final regularized integral can be written as the Hadamard finite part of the integral

$$
\int_{\mathbb{R} + \mathrm{i}\epsilon} F(z)\rho_{\mathrm{p}}(z) dz = ⨏_{\mathbb{R}} F(z)\rho_{\mathrm{p}}(z) dz - \mathrm{i}\pi \sum_j \mathrm{Residue} \Big|_{z = \theta_{\mathrm{p}}^j} \big[ F(z)\rho_{\mathrm{p}}(z) \big],
\tag{F.7}
$$

with the finite part taken with respect to each pole at any particle position $\theta_{\mathrm{p}}^j$ and defined as

$$
⨏ \mathrm{d}\theta \, \frac{f(\theta)}{(\theta - \alpha)^2} = \lim_{\epsilon \to 0^+} \left( \int_{-\infty}^{\alpha - \epsilon} \mathrm{d}\theta \, \frac{f(\theta)}{(\theta - \alpha)^2} + \int_{\alpha + \epsilon}^{+\infty} \mathrm{d}\theta \, \frac{f(\theta)}{(\theta - \alpha)^2} - \frac{2f(\alpha)}{\epsilon} \right).
\tag{F.8}
$$

It remains to discuss the contributions from the residues at the positions of the particles. After having summed also on the particle positions these will be some regular contributions to add to the form factors expansion. Therefore we can always define shifted form factors to include also those contributions. For example for the case of

two particle-hole excitations we can always shift the form factor with an analytic function $\widetilde{F}_{\theta_{\mathrm{p}}^1,\theta_{\mathrm{p}}^2}(\theta_{\mathrm{h}}^1,\theta_{\mathrm{h}}^2) = F_{\theta_{\mathrm{p}}^1,\theta_{\mathrm{p}}^2}(\theta_{\mathrm{h}}^1,\theta_{\mathrm{h}}^2) + R_{\theta_{\mathrm{p}}^1,\theta_{\mathrm{p}}^2}(\theta_{\mathrm{h}}^1,\theta_{\mathrm{h}}^2)$ such that the integrated function $R_{\theta_{\mathrm{p}}^1,\theta_{\mathrm{p}}^2}(\theta_{\mathrm{h}}^1,\theta_{\mathrm{h}}^2)$ cancels exactly the contributions from the residues in (F.6) and (F.7) (and does not affect the kinematic poles of $F_{\theta_{\mathrm{p}}^1,\theta_{\mathrm{p}}^2}(\theta_{\mathrm{h}}^1,\theta_{\mathrm{h}}^2)$). This brings us to conclude that there is always a proper choice of form factors such that the sum over excitations can be regularized simply via the Hadamard integral

$$
\frac{1}{L}\sum_{\theta_{\mathrm{p}}^1,\theta_{\mathrm{p}}^2}\frac{1}{L}\sum_{\theta_{\mathrm{h}}^1,\theta_{\mathrm{h}}^2}F_{\theta_{\mathrm{p}}^1,\theta_{\mathrm{p}}^2}(\theta_{\mathrm{h}}^1,\theta_{\mathrm{h}}^2)
$$

$$
\to \int \mathrm{d}\theta_{\mathrm{p}}^1\mathrm{d}\theta_{\mathrm{p}}^2\rho_{\mathrm{h}}(\theta_{\mathrm{p}}^1)\rho_{\mathrm{h}}(\theta_{\mathrm{p}}^2)\fint \mathrm{d}\theta_{\mathrm{h}}^1\mathrm{d}\theta_{\mathrm{h}}^2\rho_{\mathrm{p}}(\theta_{\mathrm{h}}^1)\rho_{\mathrm{p}}(\theta_{\mathrm{h}}^2)\widetilde{F}_{\theta_{\mathrm{p}}^1,\theta_{\mathrm{p}}^2}(\theta_{\mathrm{h}}^1,\theta_{\mathrm{h}}^2), \tag{F.9}
$$

with the same logic applying to higher particle-hole excitations. A similar result can be found in [137].

# G   Diffusion matrix with different particle types

Result (4.15) has been derived using the particle-hole form factor expansion, with the understanding that there is a single quasiparticle type (a single string length). We now explain how to extend the derivation result to integrable models with arbitrary number of quasiparticle types, in agreement with the proposed general result (4.19).

In general the rapidites describing a generic state in integrable models are not only real and, in the thermodynamic limit, they form patterns on the complex plane called strings, which can be interpreted as bound states. Strings are characterised by their centre of mass rapidity $\theta_\ell^{(a)}$ with $\theta_\ell^{(a)} \in \mathbb{R}$, and their length $m_a$, such that the rapidities belonging to a string are given by

$$
\theta_{i,\ell}^{(a)} = \theta_\ell^{(a)} + \mathrm{i}\kappa(m_a + 1 - 2i) \tag{G.1}
$$

with $i = 1,\ldots,m_a$, for some $\kappa$ that depends on the model [7]. In some system each string can also carry a sign or parity $\sigma$ associated to it, expressing the sign of the derivative of its dressed momentum, see for example the gapless XXZ chain or the Hubbard chain [113,181]. For each string length, the centre of mass rapidities becomes dense on the real line, and therefore eigenstates can be described by densities of string-centre rapidities, $\rho_{\mathrm{p},a}(\theta)$, where $a$ runs over all the allowed types of strings (allowed string lengths). The string centres and types are interpreted as quasiparticles' rapidities and types. In this description, the quasiparticles scatter diagonally and elastically, with two-body scattering amplitude obtained by summing the scattering amplitudes between each elements of the two strings

$$
\frac{\log S_{a,b}(\theta_1^{(a)},\theta_2^{(b)})}{2\pi\mathrm{i}} = \frac{1}{2\pi\mathrm{i}}\sum_{i=1}^{m_a}\sum_{j=1}^{n_b}\log S\left(\theta_{i,1}^{(a)},\theta_{i,2}^{(a)}\right). \tag{G.2}
$$

---

[7]In the gapless XXZ spin chain, when the anisotropy $\Delta$ is chosen to be at the (so-called) roots of unity values $\Delta = \cos \pi n/m$, due to the periodicity on the imaginary line of the scattering kernel, the string expression should be generalized to $\theta_\ell^{(a)} + \mathrm{i}\kappa(m_a + 1 - 2i) + i\pi(1 - v_a)/4$ with the additional parameters $v_a = \pm 1$ (to not be confused with the parity $\sigma_a$) determined by the choice of the integers $n, m$, see for example [112].

In these cases, the natural generalisation of (4.15) is to replace each rapidity integral by the combination of a rapidity integral and a sum over quasiparticle types as it follows

$$(\mathfrak{D}C)_{ij} = \sum_{a,b} \int \frac{d\theta_1 d\theta_2}{2} \rho_{p;a}(\theta_1) f_a(\theta_1) \rho_{p;b}(\theta_2) f_b(\theta_2) |v_a^{\text{eff}}(\theta_2) - v_b^{\text{eff}}(\theta_1)|$$

$$\times \left( T_{a,b}^{\text{dr}}(\theta_2,\theta_1) \right)^2 \left( \frac{h_{i;b}^{\text{dr}}(\theta_2)}{\sigma_b \rho_{s;b}(\theta_2)} - \frac{h_{i;a}^{\text{dr}}(\theta_1)}{\sigma_a \rho_{s;a}(\theta_1)} \right) \left( \frac{h_{j;b}^{\text{dr}}(\theta_2)}{\sigma_b \rho_{s;b}(\theta_2)} - \frac{h_{j;a}^{\text{dr}}(\theta_1)}{\sigma_a \rho_{s;a}(\theta_1)} \right), \quad \text{(G.3)}$$

with $f_a(\theta) = 1 - n_a(\theta)$. here the only non-trivial generalization is the presence of the parity $\sigma_a$ associated to the particle $a$, defined as $k_a'(\theta) = 2\pi\sigma_a \rho_{s,a}(\theta)$, with the momentum of the string given in (G.5) and with the dressing of the scattering kernel, and all of the other thermodynamic functions, provided by

$$T_{a,b}^{\text{dr}} = [(1 - Tn\sigma)^{-1}]_{a,c} T_{c,b}, \quad \text{(G.4)}$$

with $\sigma$ the vector of signs $\sigma_a$ [8]. We provide below an argument for the validity of this generalisation for the diffusion operator in quantum integrable models.

In order to confirm the validity of formula (G.3) one should notice that the scattering kernel $T_{a,b}(\theta,\alpha)$, analogously to the scattering amplitude (G.2), is additive on the string components and similarly, the dressed (and bare) energy and momentum functions for the strings are also given by the sum of all the string components. Given the string $\theta_i^{(a)} = \theta^{(a)} + i\kappa(m_a + 1 - 2i)$ indeed one has for the string momentum and energy

$$k_a(\theta^{(a)}) = \sum_{i=1}^{m_a} k(\theta_i^{(a)}) \quad \text{(G.5)}$$

$$\varepsilon_a(\theta^{(a)}) = \sum_{i=1}^{m_a} \varepsilon(\theta_i^{(a)}). \quad \text{(G.6)}$$

This directly implies that all formulae of subsections 3.1 and 3.2, as well as (3.35), hold by replacing the integral over rapidities into the sum over particle types $a$ and integral on the real parts

$$\int d\theta \rightarrow \sum_a \int d\theta^{(a)}. \quad \text{(G.7)}$$

This agrees with the general TBA structure used in GHD, see [49, 50]. Further, it is clear that in the form factor expansion, one must also sum over string lengths, as particle and hole excitations can be created for any string states: the presence of strings introduces extra types of excitations on top of the length-1 particle-hole excitations in the resolution of identity (3.30). In fact, in general, not only particle-hole pairs with the same string length contribute, but also strings can be destroyed and larger or smaller strings created (for example, two strings of size 1 can make a string of size 2 or vice versa). Thus, the replacement (G.7) is to be applied also in the form factor expansion (with the condition that total string lengths of particles agree with that of holes for given form factor to be nonzero). As for the kinematic pole conditions in subsection 3.3, we need to assume

---

[8]In the gapped spin XXZ chain all $\sigma_a$ are equal to 1, however this is not the case in the gapless regime at roots of unity [112] or in fermionic models like the Fermi-Hubbard chain [113]. These signs also enter the description of the local stationary state also at Euler scale, see [50, 114].

that formulae (3.40) and (3.41) hold, and that poles can only occur, again, at coinciding particle and hole rapidities. The explicit residues, when particle and hole types agree, are assumed to be given by the natural generalisation of (3.49):

$$
\begin{aligned}
f_{i;a,b}(\theta_{\mathrm{p}}^1,\theta_{\mathrm{h}}^1,\theta_{\mathrm{p}}^2,\theta_{\mathrm{h}}^2) =& \frac{T_{b,a}^{\mathrm{dr}}(\theta_{\mathrm{h}}^2,\theta_{\mathrm{h}}^1)h_{i;b}^{\mathrm{dr}}(\theta_{\mathrm{h}}^2)}{\rho_{\mathrm{s};a}\sigma_a(\theta_{\mathrm{h}}^1)k_b'(\theta_{\mathrm{h}}^2)(\theta_{\mathrm{p}}^1-\theta_{\mathrm{h}}^1)} + \frac{T_{a,b}^{\mathrm{dr}}(\theta_{\mathrm{h}}^1,\theta_{\mathrm{h}}^2)h_{i;a}^{\mathrm{dr}}(\theta_{\mathrm{h}}^1)}{\rho_{\mathrm{s};b}(\theta_{\mathrm{h}}^2)\sigma_b k_a'(\theta_{\mathrm{h}}^1)(\theta_{\mathrm{p}}^2-\theta_{\mathrm{h}}^2)} \\
&+ \frac{T_{b,a}^{\mathrm{dr}}(\theta_{\mathrm{h}}^2,\theta_{\mathrm{h}}^1)h_{i;b}^{\mathrm{dr}}(\theta_{\mathrm{h}}^2)}{\rho_{\mathrm{s};a}(\theta_{\mathrm{h}}^1)\sigma_a k_b'(\theta_{\mathrm{h}}^2)(\theta_{\mathrm{p}}^2-\theta_{\mathrm{h}}^1)} + \frac{T_{a,b}^{\mathrm{dr}}(\theta_{\mathrm{h}}^1,\theta_{\mathrm{h}}^2)h_{i;a}^{\mathrm{dr}}(\theta_{\mathrm{h}}^1)}{\rho_{\mathrm{s};b}(\theta_{\mathrm{h}}^2)\sigma_b k_a'(\theta_{\mathrm{h}}^1)(\theta_{\mathrm{p}}^1-\theta_{\mathrm{h}}^2)} \\
&+ \text{regular}, \qquad\qquad\qquad\qquad\qquad\qquad\qquad\qquad (\mathrm{G.8})
\end{aligned}
$$

where the presence of the parities $\sigma_a, \sigma_b$ comes from the proper definition of the back-flow function $F$, related to the dressed scattering kernel by (3.48) when parities are non-trivial (see for example the supplementary material of [50]). With these, the derivation of (4.15) carried out in section 4.1 can be done in the presence of different types of strings. The kinematic constraints (4.5) for generic two-body scattering processes then would read

$$
k_a(\theta_{\mathrm{p}}^1) + k_b(\theta_{\mathrm{p}}^2) = k_c(\theta_{\mathrm{h}}^1) + k_d(\theta_{\mathrm{h}}^2), \quad \varepsilon_a(\theta_{\mathrm{p}}^1) + \varepsilon_b(\theta_{\mathrm{p}}^2) = \varepsilon_c(\theta_{\mathrm{h}}^1) + \varepsilon_d(\theta_{\mathrm{h}}^2), \qquad (\mathrm{G.9})
$$

with the extra condition for the conservation of the total particle number

$$
m_a + m_b = m_c + m_d. \qquad\qquad (\mathrm{G.10})
$$

Due to the vanishing of the form factor with the total energy difference, as shown in section 4.1, the only finite contributions to diffusion are the solutions of the kinematic constrains that collapse the integral on the kinematic poles of the form factor, namely when $\theta_{\mathrm{p}}^i \to \theta_{\mathrm{h}}^j$. Since generically dressed momenta and energies at a given rapidity are different for different string types, it is not hard to check that the only solutions are $c = a$ and $d = b$, or $c = b$ and $d = a$, that is, for coinciding-rapidity particle-hole pairs with the same string type. The expression for the matrix $(\mathfrak{D}C)_{ij}$ in presence the of generic types of particles generalises to eq. (G.3).

## H Comparison with diffusion in the hard-rod gas

Hard rod gases are a special case of classical integrable systems [64, 72, 78, 80, 142, 156, 182–186]. They are characterized by an ensemble of hard rod moving in one dimension with velocities $v$. Whenever two rods collide, they exchange their velocities. In the language of TBA, the quasiparticles are identified with the velocity tracers, following the centres of rods with a given velocity. When quasiparticles scatter, their positions are displaced by a constant shift $a$, the length of the rod. The velocity $v$ plays the role of the rapidity $\theta$ and one can analogously define the local stationary state via the local density of velocities

$$
\rho_{\mathrm{p}}(v;x,t). \qquad\qquad (\mathrm{H.1})
$$

The scattering kernel of the gas is given by

$$
T(v,v') = -\frac{a}{2\pi} \qquad\qquad (\mathrm{H.2})
$$

(the factor of $2\pi$ is related to the choice of the phase-space integration measure defining the ensemble, $\mathrm{d}p\mathrm{d}x/(2\pi)$). As the quasiparticles are classical, the statistical factor is simply $f = 1$ (the free energy function is $\mathsf{F}(\epsilon) = -e^{-\epsilon}$). Given the form of the scattering kernel and the classical statistics, the dressing of single-particle functions take the form

$$h^{\mathrm{dr}}(v) = h(v) - a\bar{\rho}\langle h\rangle, \tag{H.3}$$

where $\bar{\rho} = \int \mathrm{d}v\,\rho_{\mathrm{p}}(v)$ and $\langle h\rangle = \int \mathrm{d}v\,h(v)\rho_{\mathrm{p}}(v)/\bar{\rho}$. Therefore,

$$T^{\mathrm{dr}}(v, v') = -\frac{a}{2\pi}(1 - a\bar{\rho}). \tag{H.4}$$

Also, one has

$$2\pi\rho_{\mathrm{s}}(v) = 1 - a\bar{\rho}, \tag{H.5}$$

and

$$v^{\mathrm{eff}}(v) = \frac{v - a\bar{\rho}\langle h_1\rangle}{1 - a\bar{\rho}}, \qquad h_1(v) = v. \tag{H.6}$$

We then consider our final general result (4.19) and we specialize to the hard rod gas by using a single quasiparticle type (hence no type index), and setting $f(v) = 1$ and $T^{\mathrm{dr}}(v, v') = -\frac{a}{2\pi}(1 - a\bar{\rho})$. By Galilean invariance it is sufficient to consider states with zero average velocities, $\langle h_1\rangle = 0$, and putting everything together, we find the the operator $\mathfrak{D}C(v, v')$ in velocities space given by

$$\mathfrak{D}C(v, v') = a^2(1 - a\bar{\rho})^{-1}\big(\delta(v - v')r(v)\rho_{\mathrm{p}}(v) - \rho_{\mathrm{p}}(v')\rho_{\mathrm{p}}(v)|v - v'|\big) \tag{H.7}$$

with

$$r(v) = \int dv'\rho_{\mathrm{p}}(v')|v' - v|. \tag{H.8}$$

This expression agrees with the one previously found in [64, 72, 80]. This confirms that although derived in the quantum models, the expression (4.15) generalises to (4.19), expressed in complete generality as a function of the differential scattering kernel $T(\theta, \alpha)$ and the statistical factor $f(\theta)$. We stress that, differently from the expression of the effective velocity (3.19), the diffusion matrix (4.19) depends explicitly on the statistical factor $f$ and therefore classical and quantum models are expected to have in general different diffusive dynamics.

# I  Numerical evaluations of the spin diffusion constant and tDMRG predictions

In Figure 3 we have plotted the diffusion constant of a XXZ chain at infinite temperature $\beta = 0$ and for different values of $\Delta$. This is indeed the only case where we were able to solve numerically the equations for the dressed scattering kernel $T^{\mathrm{dr}}_{a,b}$, see E. We find a finite value at $\Delta \to \infty$ given by $\frac{1}{2}\lim_{\Delta\to\infty}\mathfrak{D}_{00} \simeq 0.424$, close to the value first numerically predicted in [91]. We compare the solution with numerical data obtained by simulating the dynamical correlator $\langle \mathsf{j}^z_0(j, t)\mathsf{j}^z_0(0, 0)\rangle^c$ with tDMRG up to some maximal time $t_{\max}$ in [35, 91, 92] and with the spin current $\mathsf{j}^z_0(j, 0) = \frac{i}{2}\,(S^+_j S^-_{j+1} - S^-_j S^+_{j+1})$. We find

that these data only constitute a lower bound to the exact spin diffusion constant as they are slightly smaller than our theoretical prediction. The discrepancy is due to the time truncation, namely by defining the finite time diffusion constant as

$$\mathfrak{D}_{00}(t_{\max})/2 = \sum_j \int_0^{t_{\max}} dt \, \langle j_0^z(j,t) j_0^z(0,0) \rangle^c, \tag{I.1}$$

we have

$$\mathfrak{D}_{00}(t_{\max}) \le \mathfrak{D}_{00}(\infty), \tag{I.2}$$

since the correlator appears to be positive for any $t \ge 0$ in this regime. In Figure 4 we plot the diffusion constant obtained by integrating over a maximal time $t_{\max}$ as function of this. Since the correlator $\sum_j \langle j_0^z(j,t) j_0^z(0,0) \rangle^c$ is expected to decay at large times $t$ as a power law

$$\sum_j \langle j_0^z(j,t) j_0^z(0,0) \rangle^c \sim t^{-3/2}, \tag{I.3}$$

(compatible with the numerical data) the integrated one converges to the infinite $t_{\max}$ value with corrections of order $t_{\max}^{-1/2}$. We use the fitting function $a + b t_{\max}^{-1/2}$ to extrapolate the numerical value of $a$ and we find that this is in much better agreement with our theoretical prediction (6.38). Finally we find that the diffusion constant diverges in the limit $\Delta \to 1$ as $\sim (\Delta - 1)^{-1/2}$, signaling super-diffusive behaviour, in accord with recent predictions [99].

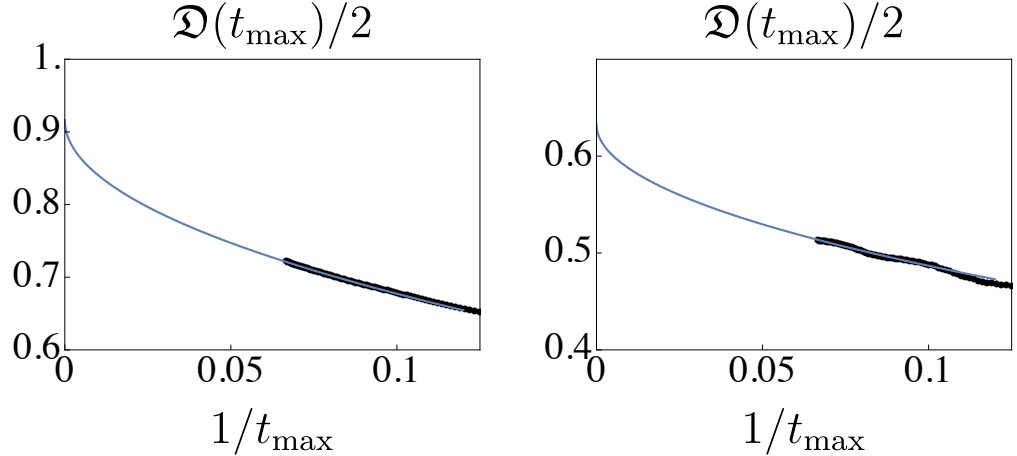

Figure 4: Plot of the infinite temperature diffusion constant at finite time, defined in equation (I.1) for a gapped XXZ chain at infinite temperature and half-filling at $\Delta = 1.5$ (Left) and $\Delta = 3$ (Right). Dots are obtained from tDMRG simulations from [35, 91, 92] and the continuous line is a fitting function as $a + t_{\max}^{-1/2} b$ with $a, b$ fitting parameters.

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
