# Peer review of "Diffusion in generalized hydrodynamics and quasiparticle scattering"

_SciPost Physics, doi:SciPost Phys. 6, 049 (2019)_

## Round 2 · Referee Report · Anonymous (Referee 2) · 2019-2-13

Strengths

See report

Weaknesses

See report

Report

In this article the authors review and expand upon recent attempts to study diffusive corrections to the generalized hydrodynamic (GHD) theory for integrable models.

The discussed theory and the obtained results are certainly of interest and worth to be published. I do, however, see some problems with the manuscript in its current form. My main criticisms are:

1) There is often no clear distinction made between phenomenological theory, conjectures for exact or asymptotic formulas, and formulas which are exact and proven.

The hydrodynamic equation (2.5) and the Navier-Stokes corrections (2.8) are a phenomenological ansatz which might or might not provide good approximations of the true dynamics in the appropriate limit. In this regard one should perhaps mention that even the seemingly simplest case of non-interacting particles (where according to the manuscript the hydrodynamic description should be exact in the appropriate limit without any corrections) does not appear to be completely settled, see e.g. Ref. [86].

Accepting GHD+corrections as the phenomenological framework, the paper also makes several claims of a proof for certain formulas (e.g. for the current (3.18), see also Appendix D) which in my view are instead rather limited consistency checks.

2) A problematic point when considering corrections to hydrodynamics is that the corrections terms are not uniquely defined. This point is discussed in the paper and a solution is proposed choosing a particular 'gauge' by PT symmetry. It has not at all become clear to me why this is the proper choice leading, for example, to the same diffusion constant obtained from a microscopic calculation.

3) I do find some of the notation very cumbersome and also used inconsistently. In chapter 2, for example the authors define \bar{q}_i(x,t)=:<q_i(x,t)> and <j_i(x,t)>=:\bar{j}_i\bar{q}(.,t). Why are there two different notations for the same quantity? The notation with the 'bars' is then used in (2.7) and (2.8) but then almost completely disappears for the rest of the paper.

Further points:

i) Large parts of chapters 1-4 have already been published elsewhere. For a regular article the percentage of material which is a review appears quite high.

ii) Page 4: 'no diffusive dynamics in non-interacting systems'. This is too general. Diffusive dynamics can, for example, arise in non-interacting systems in the presence of time-dependent potentials, see e.g. PRL 39, 1424 (77).

iii) Eq. (2.7): F_i does not seem to be defined.

iv) Eq. (3.18-3.20) are simply stated. Below it says that Appendix D provides an alternative proof implying that a proof has already been provided. I also fail to see why App. D is a proof.

v) There are several typos which make it unclear what the authors are refering to. On pages 21, 22 the authors refer for example to points (i) and point (ii) but there are no such points.

vi) There has been a long discussion whether or not there is spin diffusion in the XXZ chain. I find it misleading to state that a standard diffusion equation for the local magnetization is obtained, see (6.5). This is simply a consequence of the phenomenological ansatz made which is purely diffusive in the absence of a ballistic part. This is not related to the particular model considered and does not provide any information about that model.

vii) Below (6.7) a figure is not properly referenced.

xiii) The authors only mention that the phenomenological GHD+corrections theory might possibly fail in the last paragraph of the conclusions. This point does deserve a broader discussion and should already be made in the introduction.

ix) I wonder how much formulas like (3.51) help in explaining the final formulas for the diffusion matrix. From the information given I find it impossible to judge whether the conjectured form for the two particle-hole form factors is valid. The formulas which are finally evaluated (6.32-6.33) are perhaps easier to understand directly as a conjecture.

Requested changes

see report

  • validity: good
  • significance: high
  • originality: good
  • clarity: ok
  • formatting: excellent
  • grammar: excellent

Author:  Jacopo De Nardis  on 2019-03-05  [id 458]

(in reply to Report 2 on 2019-02-13)

1) We believe that we have spelled out in the introduction which are the assumptions underlying our derivation, especially now in the resubmitted version. Please notice that ref.[86], which deals with free fermionic theories, is not in contradiction with our results as this reference pointed the existence of higher order corrections to Euler hydrodynamics that start at third order in derivative expansion for free theories (while the diffusion processes we are dealing here are at the second order and are absent in free theories)

2) See answer to referee 1. We have added a detailed discussion on gauge invariance and gauge covariance in Appendix C. We also improved clarity in the main text. Please do not confuse the microscopic diffusion, defined via the diffusive spreading of correlation functions, with the diffusion matrix.

3) Thank you for pointing this possibly confusing notation. We have optimised and uniformised our notation.

i) Even if some part of those chapters is partially spread out in the literature, we believe it is useful for the scientific community to have a consistent presentation as we have proposed. Chapters 1-4 contain a large quantity of unpublished material.

ii) By this statement we mean that the diffusion matrix is zero in free systems. This was recently put forward in ref.[69] and it is confirmed by our analysis. The example mentioned by referee 2 deals with open systems (which of course may show diffusive behaviours) while we are here dealing with closed systems. We actually wrote explicitly in the introduction that « confirming the general intuition that there is no diffusive dynamics in non-interacting systems [69,86] (except, potentially, with external disorder [87–90]) ».

iii) Thank you. We corrected it.

iv) We have added references to the previous proof, and give more detailed steps in the Appendix D. But since both referees are not happy with calling this a proof we changed the title of the appendix to “derivation” instead of proof.

v) Thank you for pointing out these misprints.

vi) It is true that the simple diffusion equation comes directly from the hydrodynamic assumption with the extra information that $v_\infty=0$ (by symmetry) in this particular case. However, we do provide a computation of the diffusion constant as a function of all external parameters (temperature, etc.) in the XXZ chain from the microscopic model and checked it numerically. This prediction contains new information: We show that in the limit of half filling (states with zero total magnetisation) the spin ballistic current (or alternatively, the spin Drude weight) is zero and the diffusion constant is not. We analytically and numerically evaluate the latter, and we compare with state-of-the-art DRMG simulations in that specific model.

vii) Thank you. We corrected it.

viii) Actually we already mentioned in Section 2 and 6 the possibility of super-diffusion, which will manifest itself through divergences in the Onsager matrix (for example in the Heisenberg XXX chain at finite temperature). We pointed this again in the introduction as this is an interesting point to investigate further. This is a subject of current research.

ix) Formulas 6.34 for the spins diffusion constant at half-filling (zero magnetic field) is a direct consequences of our main result, formula 4.22 for the Onsager matrix and its derivation from the latter is the main scope of section 6. Formula 4.22 is based on a conjecture on the poles of the form factors. We believe that this is very different from stating that 6.34 is a conjectured equation.

---

## Round 2 · Referee Report · Anonymous (Referee 1) · 2019-2-13

Strengths

1) The manuscript deals with the interesting topic of extending the hydrodynamic theory for integrable systems beyond the Euler scale, namely to account for diffusive dynamics.

2) The manuscript is well-written and provides many steps that are necessary for the treatment of the subject.

Weaknesses

1) Unfortunately, many of the necessary techniques, e.g. TBA and "backflow" for excited states, are quoted from the literature without attempting a derivation from just few principles. This observation would not be a critique if the manuscript was compact, but it comprises more than 60 pages. Many of the comments on equations should be replaced by derivations, even if they were brief.

2) There are many instances where assumptions are used that are not spelled out.

Report

The manuscript deals with the interesting topic of extending the hydrodynamic
theory for integrable systems beyond the Euler scale, namely to account for
diffusive dynamics. The manuscript is well-written and provides many steps
that are necessary for the treatment of the subject. Unfortunately, many of
the necessary techniques, e.g. TBA and "backflow" for excited states, are
quoted from the literature without attempting a derivation from just few
principles. This observation would not be a critique if the manuscript was
compact, but it comprises more than 60 pages. Many of the comments on
equations should be replaced by derivations, even if they were brief.

In (2.6) and (2.7) functions \cal F and \cal D are postulated allowing to
express the expectation value of j from the expectation value of q and its
derivative in case of inhomogeneous systems. From (2.7) the relation (B.3)
between the <jq> and <qq> correlations is "derived". I do not trust this
derivation as (B.2) is used with a generating function depending on some
generating field beta(x) which may or may not depend on time. If it does not
depend on time, then a time integral on the r.h.s. of (B.2) should appear. If
the field beta(x) is time dependent, then relation (2.7) must have built in a
dependence on this field and then additional terms appear in (B.3).

The next step in the computation of the diffusion matrix makes use of the
partial differential equation for the <qq> correlator and explicit results for
this correlator.

It is, however, quite disturbing that the results for the diffusion matrix
depend on the "gauge" of the local conserved operator. Adding a divergence
field preserves the continuity equation, but changes the diffusion matrix. The
authors state this honestly by "One must therefore choose a gauge in order to
fix the diffusion matrix". And they do it by choosing a gauge rendering q and
j invariant under PT transformation. They even prove that there is a unique
gauge to this end. But they do not prove why this yields the physical
diffusion matrix. I would find the procedure presented by the authors
convincing if they could prove that the two differing diffusion matrices for
the left hand side and the right hand side of (2.26) would result in
compatible L-coefficients (by use of (2.14)).

The authors use the current formula (3.18) and comment after (3.20) that "in
Appendix D we also provide an alternative proof for this formula". I do not
see any proof of (3.18) in the main body of the manuscript. So it is not
appropriate to call Appendix D an "alternative" proof. (In addition I have to
note that the "proof" of Appendix D is full of assumptions.)

Section 4 is very technical and hard to digest. Many calculations depend on
regularization schemes like (3.37), but I trust that the authors here apply
(their own) state of the art results for the Lieb-Liniger gas. Section 5 reaps
nice physical results.

In summary, I like to ask the authors to sharpen their presentation. There are
many instances (see above) where assumptions are used that are not spelled
out.

Requested changes

1) I like to ask the authors to generally sharpen their presentation.

2) Please replace general comments on equations by deductions.

3) Please spell out assumptions and keep them separate from proper proofs.

  • validity: ok
  • significance: high
  • originality: high
  • clarity: ok
  • formatting: reasonable
  • grammar: excellent

Author:  Jacopo De Nardis  on 2019-03-05  [id 459]

(in reply to Report 1 on 2019-02-13)

We thank the referee for his.her insightful comments which led, we believe, to clarifications of important aspects of the discussion that are now in the resubmitted version.

We have spelled out in the main text which are the assumptions underlying our derivation. In the introduction, we have clarified the main hypotheses for the paper: those underlying the hydrodynamic approach, and those used for the form factors. The former is a standard set of hypotheses dating from more than one century. The latter is grounded on many examples, but a general theory is still missing and worth investigating further (see the recent paper [137]).

“In (2.6) and (2.7) functions \cal F and \cal D are postulated allowing to express the expectation value of j from the expectation value of q and its derivative in case of inhomogeneous systems.”

it maybe worth clarifying that we actually compute the Onsager matrix, which code for the microscopic diffusive spreading of the charges correlation functions. The diffusion coefficient \mathfrak{D} is then reconstructed using the hydrodynamics assumptions. The validity of the hydrodynamic assumptions have been proved in mathematically rigorous way only for a very restricted set of models (which however include the classical hard rod model, see ref.[80]).

“From (2.7) the relation (B.3) between the <jq> and <qq> correlations is "derived". I do not trust this derivation as (B.2) is used with a generating function depending on some generating field beta(x) which may or may not depend on time. If it does not depend on time, then a time integral on the r.h.s. of (B.2) should appear. If the field beta(x) is time dependent, then relation (2.7) must have built in a dependence on this field and then additional terms appear in (B.3).
The next step in the computation of the diffusion matrix makes use of the partial differential equation for the <qq> correlator and explicit results for this correlator.”

Thank you for pointing out the lack of clarity here. The question of the referee concerns the dynamical variables used in hydro. We had discussed aspects of this in section 2, however we have tried to further clarify and added more explanations: sentences around the new equation 2.4, and explanations in the appendix B. The main point is that the dynamical variables of hydro are the conserved densities evaluated on a given time slice. The choice of reference time slice is arbitrary (similarly to, for instance, classical field theory). These dynamical variables evolve in time according to the hydrodynamic equations. Therefore, local changes of initial conditions of this dynamical system - local changes of $\bar{\mathfrak{q}}_i(x,0)$ - produce changes of averages evaluated at any positive time, because such averages are evaluated by time evolving from a perturbed initial condition. By statistical mechanics principle, a change of initial condition at some point $x$ is equivalent to the insertion of a local observable at $x$ and $t=0$. There is no time integration, as only the initial condition is changed; the time evolution is unchanged. The effect is of course nevertheless nontrivial on observables evaluated at all later times, and this is described by a correlation, as on the right hand side of B.3. We assume there is a change that exactly inserts local conserved densities; the associated variables are the local effective temperatures $\beta_i(x)$.

“It is, however, quite disturbing that the results for the diffusion matrix depend on the "gauge" of the local conserved operator. Adding a divergence field preserves the continuity equation, but changes the diffusion matrix. The authors state this honestly by "One must therefore choose a gauge in order to fix the diffusion matrix". And they do it by choosing a gauge rendering q and j invariant under PT transformation. They even prove that there is a unique gauge to this end. But they do not prove why this yields the physical diffusion matrix. I would find the procedure presented by the authors convincing if they could prove that the two differing diffusion matrices for the left hand side and the right hand side of (2.26) would result in compatible L-coefficients (by use of (2.14)).”

Thank you again for this good question. There was already a comment in particular about the independence of the L matrix upon choice of gauge, but it was not clear and not well stated. We have clarified the discussion in the main text, see the first paragraph of section 2.4, and we have added further details in appendix C, sub appendix C.1. The Drude weight $D_{ij}$ and the Onsager coefficients ${\cal L}_{ij}$ are indeed gauge invariant (and the latter code for the microscopic diffusive spreading of the correlation function). The coefficients ${\cal D}_{i}^{~j}$ is on the hand not invariant, but covariant, under gauge transformation, as it depends on the specific gauge for local conserved charges.

“The authors use the current formula (3.18) and comment after (3.20) that in Appendix D we also provide an alternative proof for this formula. I do not see any proof of (3.18) in the main body of the manuscript. So it is not appropriate to call Appendix D an "alternative" proof. (In addition I have to note that the "proof" of Appendix D is full of assumptions.)”

We are sorry about this confusion. What we meant was that there already are proofs of this formula in the literature, we have added references to these (Refs 49,105, 106). As for the proof in appendix D, too many steps were implicit. The main points are as follows. (1) Given the existence of a form factor expansion, the results from thermodynamic Bethe ansatz give us the one-particle form factors of densities and currents at equal particle and hole rapidities. Indeed, TBA gives eq 3.39, while the form factor expansion gives 3.40. Comparing, we get the density form factors 3.37. The continuity relation then gives the result for the currents. (2) We can then evaluate $B_{ij}$ using form factor expansion. At the same time, the average currents in homogeneous stationary states are from general principle linear functionals of the $h_i(\theta)$, thus expressible as integrals over $h_i(\theta)$ times some state-dependent function of rapidity. Thus the form-factor expression for $B_{ij}$ gives a first-order differential equation for this state-dependent function. The effective velocity times particle density is a solution to this equation. We assume uniqueness of the solution. This is the single nontrivial assumption.

"Many calculations depend on regularization schemes like (3.37), but I trust that the authors here apply (their own) state of the art results for the Lieb-Liniger gas"

We are sorry for the confusion but we stated that the final result for the diffusion matrix does not depend on the details of the regularisation scheme.
We stressed now below eq 3.30 that the only requirement is that the form factor expansion eq 3.30 exist with regularised integrations on the real axis and below eq 4.15 that eventually the final result does not depend on the regularisation.

---

## Round 3 · Referee Report · Anonymous (Referee 1) · 2019-3-18

Strengths

as before

Weaknesses

as before

Report

I thank the authors for their in general adequate response and for the amendments of their manuscript.

I made a few observations.

1) In the new final paragraph of section 1 the following sentence needs a reformulation:

"That is, we suppose that, at large times, the relevent of degrees of freedom is reduced to the local mean charge densities..."

2) Thank you for extending appendix for explanations on "gauge covariance".

(In appendix C the word "gauge" is mispelled as "gaude".)

It is good to know that (top of page 11):

"It is possible to show, assuming the validity of the hydrodynamic projection [72, 101], that the Onsager coefficients Lij are invariant under (2.28)."

and

"The hydrodynamic approximation of the currents (2.9) is explicitly dependent on the choice of densities. See Appendix C."

but why do you say

"One must therefore choose a gauge in order to fix the diffusion matrix itself."

Why not "Use any gauge and stick to it"? Maybe only very special gauges allow for a hydrodynamical approach?

3) I am still having problems with appendix B. I find (B.3) problematic. The derivation uses a generating functional with time-independent fields \beta_j(x). Hence on the RHS of (B.3) instead of a two-point correlator with second time variable identical to 0 an integral over a two-point correlator with the second time variable as variable of integration should appear.

If (B.3) in the literal form is to be derived, the generating functional should involve fields like \beta_j(x,tau).

Requested changes

see report

  • validity: high
  • significance: high
  • originality: high
  • clarity: ok
  • formatting: reasonable
  • grammar: excellent

Author:  Jacopo De Nardis  on 2019-04-04  [id 484]

(in reply to Report 2 on 2019-03-18)

1) Thank you, we have reformulated the sentence.

2) Thank you very much for pointing out the misprint.

The hydrodynamical approach holds for any choice of densities of charges. That is, the form of eq 2.10 is valid for any choice of gauge. However, the explicit function $\mathfrak{D}_i^{~j}(\bar{\mathfrak{q}})$ (and the form factors we have used to compute correlation functions) depends on the choice of gauge. Thus, one needs to fix one specific gauge in order to obtain a specific function (any gauge will do, but let us fix one). Under change of gauge, the function transforms according to specific rules, as expressed in appendix C.1. We here choose the PT symmetry for two reasons. The first is that it simplifies the relation between the Diffusion matrix and the Onsager matrix. By imposing PT symmetry we only need to compute the Onsager matrix and the susceptibility matrix in order to have the diffusion matrix. The second reason is that eventually we want to write the densities of the charges in terms of the densities of quasiparticle, eq 3.25. Since the rapidity parametrisation in densities of quasiparticles is usually chosen PT invariant (in analogy to GGE states), in order to do that it is simpler to impose PT symmetry. While this mapping is useful to carry on actual calculations, it does not mean that PT symmetry is fundamental for the hydrodynamical approach. We commented on this in the manuscript, footnote 3.  

3) We do not understand the reason of the referee’s confusion. The expectation value on the right hand side of B.3 is taken with respect to a density matrix of the form $\exp[\int dx \sum_j \beta_j(x) q_j(x)]$. That is, one has $<o(y,t)> = Z^{-1} Tr ( \exp[\int dx \sum_j \beta_j(x) q_j(x)] U_t(o(y)) )$ where $U_t$ is the time-evolution unitary operator (Heisenberg picture). Therefore differentiating the numerator with respect to $\beta_j(x)$ produces $Tr ( \exp[\int dx \sum_j \beta_j(x) q_j(x)] U_t(o(y)) q_j(x) )$, and the contribution of the denominator Z gives the connected average, the right hand side of B.3. One does not differentiate the evolution operator, only the initial distribution. Thus no integral over the time of the second operator appears. This is all commented below eq B.3. The generating functional does not involve integrals over tau of $\beta_j(x, \tau)$, as the perturbation is only in the initial state, there is no $\beta_j(x, \tau)$, there is only $\beta_j(x)$. After taking the derivative, one sets $\beta_j(x) = \beta_j$, independent of $x$, so that the correlation function becomes homogeneous and stationary (a homogeneous GGE): we are considering linear response correlations, namely small perturbation on top a homogenous GGE state. We would also like to mention that eq B.4, consequence of B.3, is relatively standard in the hydrodynamic theory.

---

## Round 3 · Referee Report · Anonymous (Referee 2) · 2019-3-24

Strengths

as before

Weaknesses

as before

Report

The authors have responded to all the points raised in my previous report and I am satisfied with the response in most cases.

I have two comments with regard to the amended version of the manuscript:

1) In my view, a distinction between phenomenological results, conjectures of possibly exact formulas (e.g. for form factors) and proofs is still not always clear. The manuscript, however, allows readers to form their own views so I do not want to insist on further changes.

2) I thank the authors for extending the section about the gauge and the diffusion matrix. Clarifying that the Onsager coefficients L_{ij} are gauge invariant is important. However, I am now wondering why choosing any particular gauge (such as the one using PT symmetry) is of relevance at all. If the L_{ij}'s are invariant: Why can I not choose any arbitrary gauge as long as I stay consistent?

Requested changes

i) The authors should further clarify point 2) above.

  • validity: high
  • significance: high
  • originality: good
  • clarity: good
  • formatting: excellent
  • grammar: excellent

Author:  Jacopo De Nardis  on 2019-04-04  [id 483]

(in reply to Report 3 on 2019-03-24)

1) We think that our claims and assumptions are now well spelled (we have made an explicit list of the two main assumptions in the introduction now). We shall repeat here once again our findings A— We have developed an hydrodynamic theory for the description of large scales interacting integrable models that account for diffusive terms. Hydrodynamics is a theory valid at large scales based on certain assumptions that we have provided here.

B— We have computed exactly the Onsager matrices of interacting integrable systems via a thermodynamic form factor expansion. The poles of the 2 particle-hole thermodynamic form factor have been conjectured via an educated guess that agrees many different checks.

2) One is indeed free to choose an arbitrary Gauge. Please see reply to 2nd referee.

---

## Editorial Decision

published